# *CSBrain*: A *C*ross-scale *S*patiotemporal *B*rain Foundation Model for EEG Decoding

**Yuchen Zhou**[1,2], **Jiamin Wu**[1,3],[*] **Zichen Ren**[1], **Zhouheng Yao**[1], **Weiheng Lu**[1],
**Kunyu Peng**[4], **Qihao Zheng**[1], **Chunfeng Song**[1], **Wanli Ouyang**[1,3], **Chao Gou**[2]
[1]Shanghai Artificial Intelligence Laboratory [2]Sun Yat-sen University
[3]The Chinese University of Hong Kong [4]Karlsruher Institut fur Technologie

## Abstract

Understanding and decoding human brain activity from electroencephalography (EEG) signals is a fundamental problem in neuroscience and artificial intelligence, with applications ranging from cognition and emotion recognition to clinical diagnosis and brain–computer interfaces. While recent EEG foundation models have made progress in generalized brain decoding by leveraging unified architectures and large-scale pretraining, they inherit a scale-agnostic dense modeling paradigm from NLP and vision. This design overlooks an intrinsic property of neural activity—**cross-scale spatiotemporal structure**. Different EEG task patterns span a broad range of temporal and spatial scales, from brief neural activations to slow-varying rhythms, and from localized cortical activations to large-scale distributed interactions. Ignoring this diversity may lead to suboptimal representations and weakened generalization ability. To address these limitations, we propose *CSBrain*, a *C*ross-scale *S*patiotemporal *B*rain foundation model for generalized EEG decoding. CSBrain introduces two key components: **(i) Cross-scale Spatiotemporal Tokenization (CST)**, which aggregates multi-scale features within localized temporal windows and anatomical brain regions into compact scale-aware token representations; and **(ii) Structured Sparse attention (SSA)**, which models cross-window and cross-region dependencies for diverse decoding tasks, further enriching scale diversities while eliminating the spurious dependencies. CST and SSA are alternately stacked to progressively integrate cross-scale spatiotemporal dependencies. Extensive experiments across 11 representative EEG tasks and 16 datasets demonstrate that CSBrain consistently outperforms both task-specific models and strong foundation baselines. These results establish cross-scale modeling as a key inductive bias for generalized EEG decoding and highlight CSBrain as a robust backbone for future brain–AI research. The code and model are available at `https://github.com/yuchen2199/CSBrain`.

## 1 Introduction

Understanding and decoding human brain activity is a long-standing ambition of neuroscience and artificial intelligence, with implications ranging from cognitive science to medical diagnostics and brain-computer interfaces (BCI) [1–6]. Among various neural recording modalities, electroencephalography (EEG) is particularly attractive due to its non-invasive nature, high temporal resolution, and affordability [7–10]. EEG signals are high-dimensional, multichannel time series that reflect the brain's dynamic electrical activity and are widely applied to abnormal identification [11, 12], motor imagery [13, 14], emotion analysis [15, 16], sleep stage analysis [17, 18], seizure detection [19, 20], and various neurological disorders [21–23]. Over the past decade, a variety of task-specific deep

---

[*]Corresponding Author (`wujiamin1@pjlab.org.cn`)

39th Conference on Neural Information Processing Systems (NeurIPS 2025).

learning models—spanning CNNs [24–26], RNNs [27–29], GNNs [30? –35] , and Transformers [36–38] —have been proposed for EEG decoding in the aforementioned applications. Although effective in narrow settings, these models are typically tightly coupled to specific datasets and task formats, limiting their generalizability and scalability [39, 40].

Motivated by the paradigm shift in AI from task-specific to foundation models, recent efforts have explored brain foundation models [40–46] that aim to learn universal EEG representations across diverse brain decoding tasks via unified architectures and large-scale self-supervised pretraining. These models represent a significant step toward generalizable EEG decoding, and most adopt a pipeline transplanted from natural language processing and computer vision [47–52] (as shown in Figure 1(a)): EEG signals are first segmented into fixed-scale tokens, and then dense attention is applied across tokens to model dependencies. However, such scale-agnostic and dense modeling strategies fundamentally mismatch the intrinsic **cross-scale spatiotemporal nature** of EEG signals, which is critical for accurately capturing task-specific neural patterns.

Specifically, **different EEG decoding tasks inherently exhibit distinct spatiotemporal scales.** *From a temporal perspective*, distinct EEG decoding tasks exhibit significant differences in neural dynamics [53, 54]. For instance, motor and speech imagine tasks involve transient bursts of activity over short time windows [55, 56], whereas sleep staging relies on slower oscillatory cycles across longer timescales [57]. *From a spatial perspective*, tasks differ in the activation patterns of brain regions due to the functional specialization of human brains [55, 58]. Motor imagery typically involves localized or focal neural activations [59], while emotion recognition requires coordinated activity across widely distributed brain regions [60], reflecting longer-range spatial dependencies. Such variability in spatiotemporal scales across tasks places strong demands on token representations and modeling capacity. Most existing foundation models face three key limitations [61, 62]: **(1) Scale-agnostic tokenization:** They rely on a one-time fixed-scale tokenization strategy, resulting in token representations that fail to accommodate the diverse neural patterns across tasks. These tokens often suffer from scale mismatch and semantic dilution, which undermines their effectiveness as the most fundamental modeling units. **(2) Structure-agnostic dense attention:** Due to the lack of spatiotemporal scale awareness in tokenization, existing foundation models rely on dense attention across all tokens to uncover task-specific neural patterns underlying noisy signals. However, indiscriminately applying attention to suboptimal tokens, without considering the intrinsic cross-scale structures, tends to introduce spurious dependencies due to the inherently high noise of EEG signals [14, 63], ultimately degrading representation quality and increasing computational overhead. **(3) Limited generalization:** These limitations collectively hinder the generalization and adaptability of unified EEG foundation models across diverse BCI tasks with heterogeneous spatiotemporal scales.

To address these limitations, we argue that EEG foundation models must move beyond scale-agnostic, dense modeling paradigms and embrace cross-scale, structure-aware architectures that respect the intrinsic regional organisation and temporal dynamics of neural activity. In this paper, we propose *CSBrain*, a *C*ross-scale *S*patiotemporal *B*rain foundation Model for generalized EEG decoding, as illustrated in Figure 1(b). Guided by neurophysiological priors, CSBrain decomposes EEG signals along spatial and temporal axes into distinct brain regions and windows, and captures cross-scale dependencies both within and across these structures, yielding robust and highly adaptable representations. Specifically, we propose a **Cross-scale Spatiotemporal Tokenization (CST)** module, which captures EEG features at multiple temporal and spatial scales within localized regions. By integrating these patterns into compact token representations, CST enables adaptable alignment with diverse spatiotemporal requirements of EEG decoding tasks. Building upon these scale-aware tokens, we propose a **Structured Sparse Attention (SSA)** module to further enrich the diversity of modeling scales for EEG. SSA captures long-range dependencies across temporal windows and brain regions in a structured and efficient manner. By replacing costly dense attention, SSA reduces spurious dependencies and computational overhead, yielding more discriminative representations for noisy EEG signals. Finally, CST and SSA are alternately stacked and mutually reinforcing, progressively integrating cross-scale dependencies across both temporal and spatial dimensions. This hierarchical design enables CSBrain to build robust EEG representations that reflect the complex patterns of neural activity, thereby achieving superior generalization across EEG decoding tasks with inherently diverse spatiotemporal demands. CSBrain fundamentally differs from prior EEG foundation models by explicitly addressing and leveraging the intrinsic cross-scale spatiotemporal structure of neural activity, a property often overlooked by previous scale-agnostic dense modeling paradigms. Overall, our contribution can be summarized as follows:

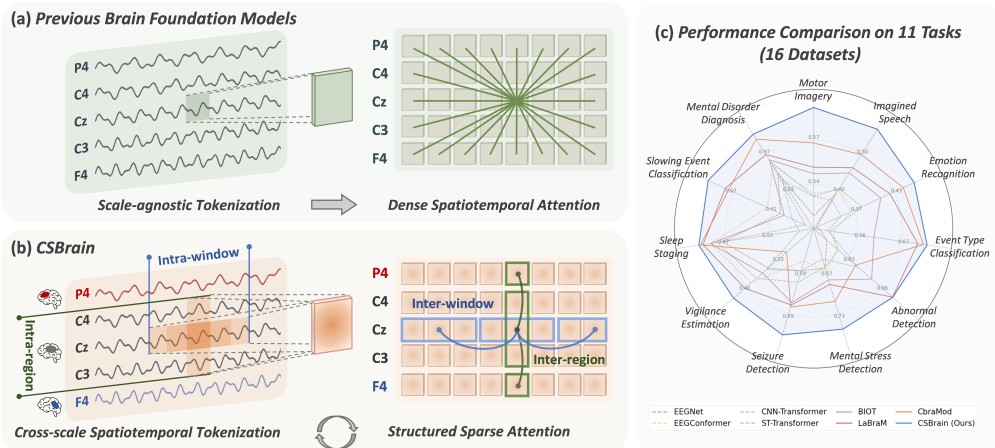

Figure 1: Comparison between previous brain foundation models and our proposed CSBrain. **(a)** Prior models adopt a scale-agnostic and densely connected architecture, which fails to reflect the intrinsic cross-scale spatiotemporal nature of EEG signals. **(b)** To address these limitations, we propose a cross-scale, sparsity-aware architecture that enables efficient and robust EEG representation learning, supporting generalization across brain decoding tasks with diverse spatiotemporal demands. **(c)** CSBrain outperforms both task-specific and foundation baselines on 11 representative tasks across 16 datasets, demonstrating consistent and superior performance.

- We propose CSBrain, a Cross-scale Spatiotemporal Brain foundation model for generalized brain decoding, offering a cross-scale, structure-aware architecture that effectively captures multi-scale spatiotemporal neural patterns across diverse tasks.

- We introduce Cross-scale Spatiotemporal Tokenization (CST) to explicitly integrate multi-scale neural patterns *within* temporal windows and brain regions into compact, scale-aware tokens. Additionally, Structured Sparse Attention (SSA) is designed to efficiently capture long-range dependencies *across* windows and regions, thereby enriching modeling scale diversity. Alternately stacked, CST and SSA progressively integrate cross-scale dependencies, enabling robust EEG representations across tasks.

- Extensive experiments across 11 representative EEG decoding tasks and 16 public datasets demonstrate CSBrain's superiority over both task-specific and foundation models. Further analysis reveals diverse cross-scale patterns across tasks, establishing cross-scale modeling as a key inductive bias for generalized EEG decoding.

## 2 Model Architecture

The overall framework of CSBrain is illustrated in Figure 2. The model first applies a standardized preprocessing module to extract initial feature representations from raw EEG signals. These features are then fed into the **Cross-scale Spatiotemporal Tokenization (CST)** module, which constructs robust cross-scale tokens by aggregating multi-resolution information *within localized temporal windows and anatomically defined brain regions*. Built upon these semantically enriched tokens, the **Structured Sparse Attention (SSA)** module expands the modeling scope, capturing long-range dependencies *across time and brain regions* while avoiding redundant interactions. CST and SSA are alternately stacked for $L$ layers, progressively integrating cross-scale spatiotemporal dependencies. Finally, a lightweight task-specific head is attached to support various objectives, including masked reconstruction during pretraining and classification or regression for downstream tasks.

### 2.1 EEG Signal Preprocessing

Initially, we formalize the input EEG signals as $E \in \mathbb{R}^{|\mathcal{C}_x| \times T}$, where $\mathcal{C}_x \subseteq \mathcal{C}$ denotes the set of electrodes used, and $T$ denotes the number of timestamps. $\mathcal{C}$ corresponds to the standardized

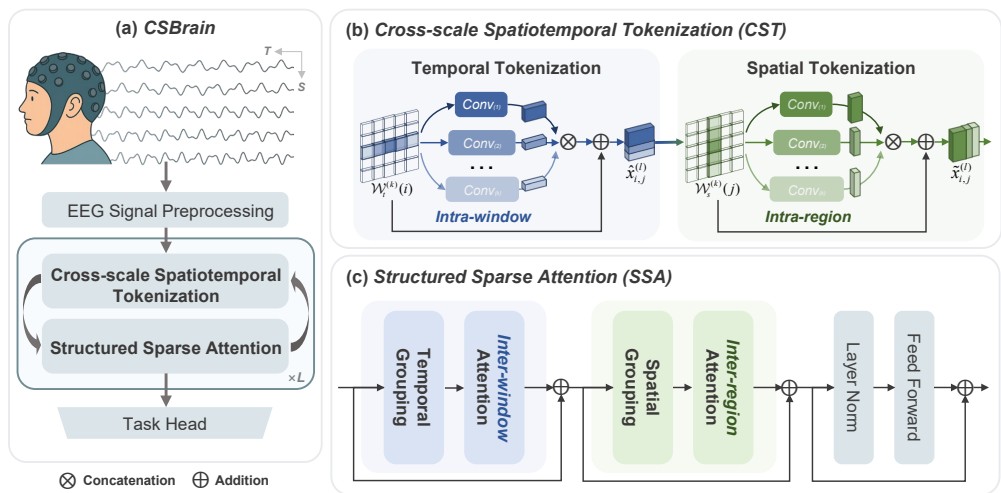

Figure 2: Overview of the CSBrain architecture. After EEG signal preprocessing, the Cross-scale Spatiotemporal Tokenization (CST) module encodes multi-resolution features within localized temporal windows and brain regions to produce robust, scale-aware tokens. The Structured Sparse Attention (SSA) module then captures long-range dependencies across windows and regions in a structured and efficient manner. CST and SSA are alternately stacked for $L$ layers to progressively integrate cross-scale spatiotemporal dependencies to build unified and robust representations for diverse BCI tasks with varying spatiotemporal scales. Finally, a lightweight task head is appended for reconstruction, classification, or regression.

international 10-20 system. To enhance signal quality and ensure feature consistency across datasets, we follow prior works [41, 40] and apply a standardized signal preprocessing pipeline.

**Signal Standardization.** The raw signals $E$ are processed with a band-pass filter to remove low-frequency drifts and high-frequency noise, followed by a notch filter to eliminate power line interference. The signals are then resampled to a uniform rate of 200 Hz and scaled to 100 $\mu$V to normalize the amplitude range to $[-1, 1]$. To enable representation learning, the preprocessed EEG signal is temporally segmented into non-overlapping segments of length $t$, resulting in $E_p \in \mathbb{R}^{C \times n \times t}$, where $n = \lfloor \frac{T}{t} \rfloor$ is the number of segments per electrode.

**Preliminary Feature Encoding.** We extract both temporal and spectral features from each segment in $E_p$. Local temporal dynamics are captured using 1D convolutions with normalization, while spectral information is obtained by applying a Fast Fourier Transform (FFT) followed by a fully connected layer to encode frequency energy distributions. The two feature types are fused to form a unified feature embedding. Finally, we add a learnable positional encoding [46], resulting in the initial EEG representation $x^{(0)} \in \mathbb{R}^{C \times n \times d}$, where $d$ denotes the feature embedding dimension.

## 2.2 Cross-scale Spatiotemporal Tokenization (CST)

Due to the inherent cross-scale spatiotemporal nature of EEG signals, informative neural patterns emerge at heterogeneous temporal and spatial resolutions. Previous EEG decoding methods adopting a fixed token granularity cannot effectively capture such patterns, limiting its generalizability across diverse BCI tasks. To address this, we propose Cross-scale Spatiotemporal Tokenization (CST), which encodes multi-resolution information within localized temporal windows and anatomically defined brain regions into unified token representations. These tokens form the basic modeling units for subsequent attention computation. In the following, we detail the two components of CST: *Temporal Tokenization*, which captures multi-scale temporal patterns within local time windows (*i.e., intra-window*), and *Spatial Tokenization*, which extracts region-specific features within anatomical brain regions (*i.e., intra-region*).

**Temporal Tokenization.** To enable tokenization at various temporal resolutions, we introduce **multi-scale temporal convolution kernels** $C_T = \{\text{Conv}_t^{(k)}\}_{k=1}^K$, where $K$ denotes the number of kernels

and each temporal kernel $\text{Conv}_t^{(k)}$ has a kernel size $s_t^{(k)}$ and output embedding dimension $d_k$. Given the preprocessed EEG feature $x^{(l-1)} \in \mathbb{R}^{C \times n \times d}$, for each location $(i,j)$, where $i \in \{1, \ldots, n\}$ and $j \in \{1, \ldots, C\}$ denotes the temporal and electrode channel indices, respectively, we define a localized temporal window $\mathcal{W}_t^{(k)}(i)$ centered at $i$ for the $k$-th scale with size $s_t^{(k)}$. Subsequently, the multi-scale temporal convolution is applied over these windows to extract temporal patterns at varying resolutions for each location. The process can be formulated as:

$$\hat{x}_{i,j}^{(l)} = \text{Concat}\left(\{\text{Conv}_t^{(k)}(\mathcal{W}_t^{(k)}(i))\}_{k=1}^K\right) + \text{Proj}_t(x_{i,j}^{(l-1)}), \tag{1}$$

where $\text{Proj}_t$ is a residual projection for dimension alignment. The outputs from different temporal kernels are concatenated to form a temporally cross-scale token $\hat{x}_{i,j}^{(l)}$, which captures rich temporal dynamics within each channel. The resulted token representations could enhance model's robustness to signal noise and scalability to diverse decoding tasks with varying temporal patterns.

**Spatial Tokenization.** Similar to Temporal Tokenization, we design **multi-scale spatial convolution kernels** $C_S = \{\text{Conv}_s^{(k)}\}_{k=1}^K$, with each spatial kernel $\text{Conv}_s^{(k)}$ has a kernel size $s_t^{(k)}$ and output embedding dimension $d_k$. To encode spatial context across functionally related electrodes in each brain region at multiple scales, we utilize these convolution kernels to aggregate electrode features within anatomically defined brain regions. Specifically, given EEG feature $\hat{x}^{(l)} \in \mathbb{R}^{C \times n \times d}$ after Temporal Tokenization, we first divide electrodes into a set of brain regions $\mathcal{R} = \{R_r\}_{r=1}^R$ based on the 10–20 system, where each region $R_r$ comprises spatially adjacent electrodes. For each electrode $j \in R_r$, we define localized spatial neighborhoods $\mathcal{W}_s^{(k)}(j) \subseteq R_r$ with size $s_s^{(k)}$ within brain region for the $k$-th scale. Finally, the spatial tokenization is performed by applying multi-scale spatial convolution kernels over their corresponding neighborhoods:

$$\tilde{x}_{i,j}^{(l)} = \text{Concat}\left(\{\text{Conv}_s^{(k)}(\mathcal{W}_s^{(k)}(j))\}_{k=1}^K\right) + \text{Proj}_s(\hat{x}_{i,j}^{(l)}), \tag{2}$$

where $\text{Proj}_s$ is a residual projection for dimension alignment. The outputs from different spatial kernels are concatenated to form the final cross-scale token representation $\tilde{x}_{i,j}^{(l)}$, capturing localized spatial dependencies within each brain region. $\tilde{x}_{i,j}^{(l)}$ encapsulates spatiotemporal characteristics across diverse scales, serving as the final output of CST module and providing robust scale-aware features for downstream attention computation.

To balance representational capacity and computational efficiency, we allocate embedding dimensions across scales in an exponentially decaying scheme: $d_k \propto \frac{1}{2^k}$, $\sum_{k=1}^K d_k = d$, *i.e.*, assigning higher dimensions to smaller kernels to retain fine-grained features, while allocating lower dimensions to larger kernels that summarize coarse context. This strategy aligns representational capacity with information density at each scale for balanced and efficient EEG representation.

## 2.3 Structured Sparse Attention (SSA)

Prior approaches typically apply dense attention uniformly across fixed-scale tokens. While effective in some cases, such indiscriminate attention can introduce spurious dependencies, reduce representational quality, and increases computational costs. Meanwhile, our CST provides structured, cross-scale token representations within localized temporal windows and brain regions, laying a robust representational space for further modeling. Building upon this, we propose Structured Sparse Attention (SSA), which efficiently captures long-range dependencies *across* temporal windows and spatial regions while avoiding redundant interactions. SSA complements CST along both temporal and spatial scales, and together they enable structured cross-scale modeling of EEG representations. SSA consists of two components: *Inter-window Attention*, responsible for capturing long-range temporal dependencies across local windows, and *Inter-region Attention*, which models spatial dependencies across distinct brain regions.

**Inter-window Attention.** Given the cross-scale tokens $\tilde{x}_{i,j}^{(l)} \in \mathbb{R}^{C \times n \times d}$, we first perform a *temporal grouping* operation to form cross-window groups. Specifically, for each relative index $g \in \{1, \ldots, w\}$, we collect all tokens that occupy the same position $g$ in each window to form a temporal group $\mathcal{G}_t^{(g)}$. Self-attention is then computed within each group to model structured long-range dependencies across time:

$$\tilde{x}_{i,j}^{(l,\text{win})} = \text{Attn}(\mathcal{G}_t^{(g)})_{i,j} + \tilde{x}_{i,j}^{(l)}. \tag{3}$$

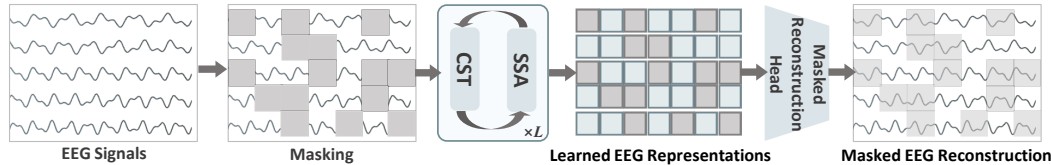

EEG Signals        Masking        Learned EEG Representations        Masked EEG Reconstruction

Figure 3: The overview of CSBrain pre-training process.

This design enables efficient and structured temporal interaction while avoiding redundancy.

**Inter-region Attention.** To ensure structured sparsity while maintaining coverage across all electrodes, we perform a *spatial grouping* operation to construct multiple spatial groups across brain regions. Unlike temporal windows (which have fixed lengths), the number of electrodes varies across brain regions. In regions with more electrodes, CST might not fully capture all information. By introducing the regional descriptor, we ensure complete information flow between regions. For each group, we sequentially sample a token $x_r^{\text{rep}}$ from each region $R_r$, and combine it with the average feature of $R_r$ to form a covering partition of the regional descriptor:

$$\tilde{x}_r = \tilde{x}_r^{\text{rep}} + \phi\left(\mathcal{P}(R_r)\right), \quad r = 1, \ldots, R, \tag{4}$$

where $\mathcal{P}(R_r)$ denotes the mean-pooled feature of region $R_r$, and $\phi(\cdot)$ is a learnable linear transformation. Notably, this ensures that each token is enriched by its region's context while preserving original information before inter-region attention and does not physically rearrange token positions. These descriptors are assembled into a spatial group $\mathcal{G}_s^{(g)} = \{\tilde{x}_1, \tilde{x}_2, \ldots, \tilde{x}_R\}$, over which self-attention is applied to capture structured dependencies across anatomical regions:

$$\tilde{x}_{i,j}^{(l,\text{reg})} = \text{Attn}(\mathcal{G}_s^{(g)})_{i,j} + \tilde{x}_{i,j}^{(l,\text{win})}. \tag{5}$$

Finally, we apply layer normalization and a feedforward network with residual connection to refine the output:

$$x_{i,j}^{(l)} = \text{FFN}\left(\text{LN}\left(\tilde{x}_{i,j}^{(l,\text{reg})}\right)\right) + \tilde{x}_{i,j}^{(l,\text{reg})}. \tag{6}$$

Together, SSA enables CSBrain to efficiently model long-range dependencies across both temporal and spatial dimensions by leveraging the robust cross-scale token layout provided by CST. This design progressively builds global context while preserving scale-awareness and mitigating spurious correlations in noisy signals, thereby enhancing generalization across diverse EEG decoding tasks.

## 2.4 Pretraining with Masked Autoencoding

To learn generalizable EEG representations, we adopt a self-supervised pretraining strategy based on masked autoencoding [51, 64], as shown in Figure 3. This process encourages the model to capture meaningful spatiotemporal dependencies from unlabeled EEG signals, laying the foundation for effective downstream transfer.

**Masking Strategy.** Given a preprocessed EEG signal $E_p \in \mathbb{R}^{C \times n \times t}$, we randomly mask a fixed ratio $r \in (0, 1)$ of segments along the temporal axis using a Bernoulli sampling scheme. This results in two subsets: visible segments $E_v \in \mathbb{R}^{C \times n_v \times t}$ and masked segments $E_m \in \mathbb{R}^{C \times n_m \times t}$, where $n_v + n_m = n$ and $\frac{n_m}{n} = r$. The masked EEG sequence is encoded by alternately stacked CST and SSA modules, which jointly model neural dependencies across multiple spatiotemporal scales. Masked segments are represented by learnable embeddings and integrated alongside visible tokens during encoding.

**Reconstruction Objective.** The reconstruction head consists of a lightweight fully connected layer, which takes as input the encoded visible tokens and learnable masked embeddings, and projects them into the original EEG signal space. Let $\hat{E}_m \in \mathbb{R}^{C \times n_m \times t}$ denote the predicted EEG segments and $E_m$ the corresponding ground truth. The reconstruction loss is defined as the mean squared error (MSE) over all masked positions:

$$\mathcal{L}_{\text{rec}} = \|\hat{E}_m - E_m\|^2. \tag{7}$$

This masked autoencoding objective drives the model to recover meaningful spatiotemporal dependencies from context, yielding robust and transferable EEG representations.

Table 1: Overview of Downstream BCI datasets and tasks used for evaluation.

| Task | Dataset | Rate (Hz) | #Channels | Duration | #Samples | #Subjects | Label |
|---|---|---|---|---|---|---|---|
| Motor Imagery Classification | BCIC-IV-2a [66] | 250 | 22 | 4s | 5,184 | 9 | 4-class |
| | PhysioNet-MI [67] | 160 | 64 | 4s | 9,837 | 109 | 4-class |
| | SHU-MI [68] | 250 | 32 | 4s | 11,988 | 25 | 2-class |
| Emotion Recognition | FACED [69] | 250 | 32 | 10s | 10,332 | 123 | 9-class |
| | SEED-V [70] | 1000 | 62 | 1s | 117,744 | 16 | 5-class |
| Seizure Detection | CHB-MIT [71, 72] | 256 | 16 | 10s | 326,993 | 21 | 2-class |
| | Siena [71, 73] | 512 | 29 | 10s | 51,307 | 14 | 2-class |
| Sleep Staging | ISRUC [74] | 200 | 6 | 30s | 89,240 | 100 | 5-class |
| | HMC [75] | 256 | 4 | 30s | 137,243 | 151 | 4-class |
| Imagined Speech Classification | BCIC2020-3 [56] | 256 | 64 | 3s | 6,000 | 15 | 5-class |
| Vigilance Estimation | SEED-VIG [76] | 200 | 17 | 8s | 20,355 | 21 | regression |
| Mental Stress Detection | MentalArithmetic [71, 77] | 500 | 20 | 5s | 1,707 | 36 | 2-class |
| Mental Disorder Diagnosis | Mumtaz2016 [78] | 256 | 19 | 5s | 7,143 | 64 | 2-class |
| Event Type Classification | TUEV [79] | 256 | 16 | 5s | 112,491 | 370 | 6-class |
| Abnormal Detection | TUAB [79] | 256 | 16 | 10s | 409,455 | 2,383 | 2-class |
| Slowing Event Classification | TUSL [79] | 256 | 23 | 10s | 245 | 28 | 3-class |

# 3 Experiments

## 3.1 Pre-training Setup

**Data Preprocessing.** We pre-train CSBrain on the Temple University Hospital EEG corpus (TUEG) dataset, which has been shown effective for foundation model studies [46, 65]. We apply standard preprocessing steps: signals are band-pass filtered between 0.3–75 Hz to remove low- and high-frequency noise, and a notch filter at 60 Hz is used to eliminate power line interference. All signals are resampled to 200 Hz and segmented into 30-second EEG samples. The amplitude is normalized to 100 $\mu$V to ensure the signal range falls within $[-1, 1]$, consistent with prior work [41, 46]. After cleaning, a total of 1,109,545 EEG segments (over 9,000 hours) are used for pre-training.

**Pre-training Settings.** We adopt a masked autoencoding objective, with a masking ratio of 50% applied to randomly sampled EEG patches. Pre-training is conducted using Python 3.11.11 and PyTorch 2.5.1 with CUDA 12.4. The model is trained with the AdamW optimizer, a learning rate of 5e-4, weight decay of 5e-2, and cosine annealing learning rate scheduling. We use a batch size of 128 and train for 40 epochs on 4 NVIDIA A100 GPUs, with a total pre-training time of approximately 101 hours. More details can be found in the supplementary material.

## 3.2 Experimental Setup of Downstream BCI Tasks

**Tasks & Datasets.** To comprehensively evaluate the generalizability of our model, we conduct experiments on **11 representative BCI tasks** spanning **16 publicly available EEG datasets**, as summarized in Table 1. These tasks include *Motor Imagery Classification* (BCIC-IV-2a, PhysioNet-MI, SHU-MI), *Emotion Recognition* (FACED, SEED-V), *Seizure Detection* (CHB-MIT, Siena), *Sleep Staging* (ISRUC, HMC), *Imagined Speech Classification* (BCIC2020-3), *Vigilance Estimation* (SEED-VIG), *Mental Stress Detection* (MentalArithmetic), *Mental Disorder Diagnosis* (Mumtaz2016), *Event Type Classification* (TUEV), *Abnormal Detection* (TUAB), and *Slowing Event Classification* (TUSL). In all experiments, we strictly follow the training, validation, and test splits to ensure fair and consistent evaluation. **To the best of our knowledge, this is among the most comprehensive evaluations conducted on EEG foundation models to date.** See supplementary for details.

**Baselines & Metrics.** We extensively compare CSBrain with two major categories of baselines: **(1)** *representative task-specific EEG decoding models*, including EEGNet [24], EEGConformer (Conformer) [36], SPaRCNet [25], ContraWR [26], CNN-Transformer (C-Trans) [38], FFCL [27], and ST-Transformer (ST-Trans) [37]; and **(2)** *recent EEG foundation models*, including BIOT [40], LaBraM [41], and CBraMod [46]. Following prior works [40, 41, 46], we report Balanced Accuracy, Cohen's Kappa, and Weighted F1 for multiclass classification tasks; Balanced Accuracy, AUC-PR, and AUROC for binary classification; and Pearson correlation, R2 score, and RMSE for regression tasks. See supplementary for details.

Table 2: Performance comparison on 11 BCI tasks Using 16 datasets.

| Dataset | Metrics | EEGNet | Conformer | SPaRCNet | ContraWR | CNN-Trans | FFCL | ST-Trans | BIOT | LaBraM | CbraMod | CSBrain |
|---|---|---|---|---|---|---|---|---|---|---|---|---|
| BCIC-IV-2a | B-Acc | 0.4482 | 0.4696 | 0.4635 | 0.4678 | 0.4600 | 0.4470 | 0.4575 | 0.4748 | 0.4869 | 0.5138 | **0.5657** |
|  | F1-W | 0.4226 | 0.4533 | 0.4432 | 0.4413 | 0.4460 | 0.4238 | 0.4471 | 0.4607 | 0.4758 | 0.4984 | **0.5637** |
| SHU-MI | B-Acc | 0.5889 | 0.5900 | 0.5978 | 0.5873 | 0.5975 | 0.5692 | 0.5992 | 0.6179 | 0.6166 | 0.6370 | **0.6417** |
|  | AUROC | 0.6283 | 0.6351 | 0.6431 | 0.6273 | 0.6343 | 0.6326 | 0.6431 | 0.6609 | 0.6604 | 0.6988 | **0.7200** |
| PhysioNet-MI | B-Acc | 0.5814 | 0.6049 | 0.5932 | 0.5892 | 0.6053 | 0.5726 | 0.6035 | 0.6153 | 0.6173 | 0.6174 | **0.6304** |
|  | F1-W | 0.5796 | 0.6062 | 0.5937 | 0.5918 | 0.6041 | 0.5701 | 0.6053 | 0.6158 | 0.6177 | 0.6179 | **0.6308** |
| BCIC2020-3 | B-Acc | 0.4413 | 0.4506 | 0.4426 | 0.4257 | 0.4533 | 0.4678 | 0.4126 | 0.4920 | 0.5060 | 0.5373 | **0.6004** |
|  | F1-W | 0.4413 | 0.4488 | 0.4420 | 0.4407 | 0.4506 | 0.4689 | 0.4247 | 0.4917 | 0.5054 | 0.5383 | **0.6003** |
| SEED-V | B-Acc | 0.2961 | 0.3537 | 0.2949 | 0.3546 | 0.3678 | 0.3641 | 0.3052 | 0.3837 | 0.3976 | 0.4091 | **0.4197** |
|  | F1-W | 0.2749 | 0.3487 | 0.2979 | 0.3544 | 0.3642 | 0.3645 | 0.2833 | 0.3856 | 0.3974 | 0.4101 | **0.4280** |
| FACED | B-Acc | 0.4090 | 0.4559 | 0.4673 | 0.4887 | 0.4697 | 0.4673 | 0.4810 | 0.5118 | 0.5273 | 0.5509 | **0.5752** |
|  | F1-W | 0.4124 | 0.4514 | 0.4729 | 0.4884 | 0.4702 | 0.4699 | 0.4795 | 0.5136 | 0.5288 | 0.5618 | **0.5796** |
| TUEV | B-Acc | 0.3876 | 0.4074 | 0.4161 | 0.4384 | 0.4087 | 0.3979 | 0.3984 | 0.5281 | 0.6409 | 0.6671 | **0.6903** |
|  | F1-W | 0.6539 | 0.6983 | 0.7024 | 0.6893 | 0.6854 | 0.6783 | 0.6823 | 0.7492 | 0.8312 | **0.8342** | 0.8333 |
| TUAB | B-Acc | 0.7642 | 0.7758 | 0.7896 | 0.7746 | 0.7777 | 0.7848 | 0.7966 | 0.7959 | 0.8140 | 0.7891 | **0.8172** |
|  | AUROC | 0.8412 | 0.8445 | 0.8676 | 0.8456 | 0.8461 | 0.8569 | 0.8707 | 0.8815 | **0.9022** | 0.8606 | 0.8957 |
| MentalArithmetic | B-Acc | 0.6770 | 0.6805 | 0.6879 | 0.6631 | 0.6779 | 0.6798 | 0.6631 | 0.6875 | 0.6909 | 0.7256 | **0.7558** |
|  | AUROC | 0.7321 | 0.7424 | 0.7418 | 0.7332 | 0.7258 | 0.7330 | 0.7132 | 0.7536 | 0.7721 | 0.7905 | **0.8478** |
| CHB-MIT | B-Acc | 0.5658 | 0.5976 | 0.5876 | 0.6344 | 0.6389 | 0.6262 | 0.5915 | 0.7068 | 0.7075 | **0.7398** | 0.7262 |
|  | AUROC | 0.8048 | 0.8226 | 0.8143 | 0.8097 | 0.8662 | 0.8271 | 0.8237 | 0.8761 | 0.8679 | 0.8892 | **0.8915** |
| Siena | B-Acc | 0.7487 | 0.7556 | 0.6572 | 0.6546 | 0.6982 | 0.6616 | 0.7527 | 0.7352 | 0.7082 | 0.7317 | **0.7662** |
|  | AUROC | 0.8687 | 0.8159 | 0.7334 | 0.7819 | 0.8719 | 0.8154 | 0.8884 | 0.9029 | 0.8814 | 0.9038 | **0.9076** |
| SEED-VIG | Corr. | 0.5127 | 0.5800 | 0.5709 | 0.5235 | 0.5829 | 0.4923 | 0.6020 | 0.6114 | **0.6347** | 0.5502 | 0.6314 |
|  | R2 | 0.1960 | 0.2065 | 0.2185 | 0.0727 | 0.1796 | 0.1740 | 0.1138 | 0.1232 | 0.1808 | 0.0737 | **0.2363** |
| ISRUC | B-Acc | 0.7154 | 0.7400 | 0.7487 | 0.7402 | 0.7363 | 0.7277 | 0.7381 | 0.7527 | 0.7633 | 0.7865 | **0.7925** |
|  | F1-W | 0.7513 | 0.7634 | 0.7624 | 0.7610 | 0.7719 | 0.7614 | 0.7681 | 0.7790 | 0.7810 | **0.8011** | 0.7990 |
| HMC | B-Acc | 0.6534 | 0.7149 | 0.4756 | 0.4242 | 0.6573 | 0.4427 | 0.2559 | 0.6862 | 0.7277 | 0.7269 | **0.7345** |
|  | F1-W | 0.6536 | 0.7080 | 0.4108 | 0.2987 | 0.6896 | 0.2902 | 0.1428 | 0.7091 | 0.7454 | 0.7395 | **0.7506** |
| TUSL | B-Acc | 0.4562 | 0.5512 | 0.4185 | 0.5857 | 0.3575 | 0.3159 | 0.4000 | 0.5785 | 0.7625 | 0.7388 | **0.8571** |
|  | F1-W | 0.4405 | 0.4895 | 0.3500 | 0.5458 | 0.2235 | 0.2120 | 0.3793 | 0.2394 | 0.7614 | 0.7453 | **0.8568** |
| Mumtaz2016 | B-Acc | 0.9232 | 0.9308 | 0.9316 | 0.9195 | 0.9305 | 0.9314 | 0.9135 | 0.9358 | 0.9409 | 0.9560 | **0.9643** |
|  | AUROC | 0.9639 | 0.9702 | 0.9781 | 0.9621 | 0.9742 | 0.9753 | 0.9594 | 0.9758 | 0.9782 | 0.9921 | **0.9956** |
| **Macro-average** |  | 0.5886 | 0.6145 | 0.5817 | 0.5849 | 0.6007 | 0.5688 | 0.5686 | 0.6322 | 0.6697 | 0.6760 | **0.7095** |

## 3.3 Experimental Results

**Performance Comparison.** To fairly and comprehensively evaluate the effectiveness of CSBrain, we benchmark it against 10 strong baselines across 11 representative BCI tasks covering 16 public EEG datasets, as summarized in Table 2. All results are averaged over five runs with different random seeds. Due to space limitations, we report two representative metrics per dataset in the main paper: Balanced Accuracy (B-Acc) and Weighted F1 (F1-W) for multi-class classification, B-Acc and AUROC for binary classification, and Pearson correlation (Corr.) and R2 score for regression tasks. Full results, including standard deviations and detailed analyses, are provided in the supplementary material.

As shown in Table 2, we highlight two key observations: **(1) CSBrain consistently achieves state-of-the-art performance across nearly all tasks and metrics.** In particular, it significantly outperforms all baselines on datasets such as BCIC-IV-2a, BCIC2020-3, Siena, and TUSL. Even in cases where CSBrain ranks second on certain metrics, the performance gap to the best-performing method remains marginal. From the macro-average in the last row of Table 2, CSBrain achieves the highest overall score across all tasks, outperforming strong foundation models such as CbraMod, LaBraM, and BIOT by 3.35%, 3.98%, and 7.73%, respectively. These results demonstrate CSBrain's strong generalization and adaptability to diverse EEG decoding scenarios with varying spatiotemporal demands. **(2) Foundation models outperform task-specific models by a significant margin.** Across all 16 datasets, EEG foundation models—including BIOT, LaBraM, CbraMod, and CSBrain—consistently outperform task-specific models. This also validates the advantage of unified model architectures trained with large-scale pretraining, which can extract generalizable neural representations across heterogeneous EEG domains.

**Tokenization Comparison.** To investigate how spatiotemporal scale affects EEG representation from the perspective of token construction, we vary the number of kernels $K$ in the CST module to control token granularity. Specifically, we compare single-scale ($K = 1$), dual-scale ($K = 2$), and full cross-scale ($K = 3$) configurations. Additionally, to evaluate the effect of stacking, we compare our full design (with CST applied before each SSA module) against a variant where cross-scale tokenization ($K = 3$) is applied only once before the first SSA. For efficiency, all variants are pre-trained and fine-tuned using 30% of the training data. As shown in Fig. 4, our SSA ($K = 3$, stacked) consistently achieves the best performance across four representative tasks: emotion recognition, motor imagery,

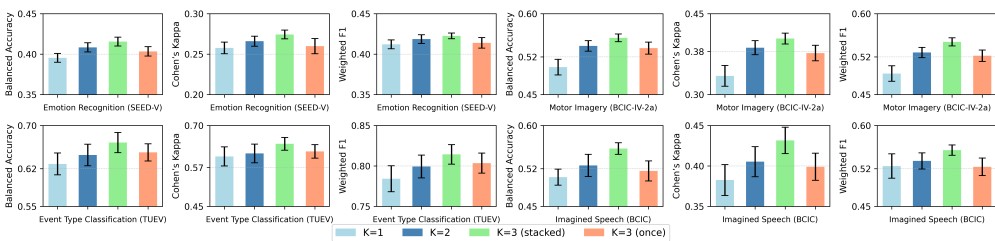

Figure 4: The performance comparisons of tokenization.

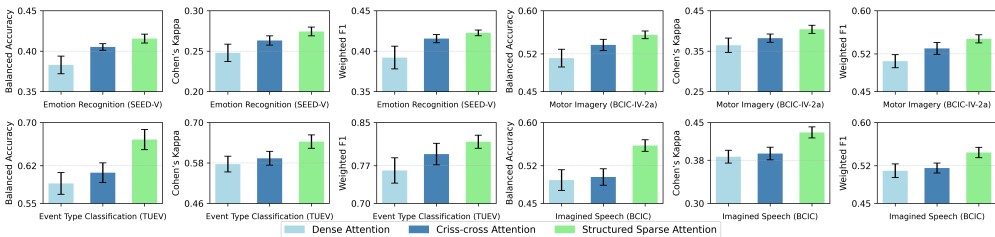

Figure 5: The performance comparisons of attention mechanism.

event classification, and imagined speech. In contrast, the single-scale variant ($K = 1$) performs the worst. Notably, on the motor imagery task, SSA outperforms the single-scale counterpart by 10.8%, 20.7%, and 12.1% on three evaluation metrics, respectively. Moreover, we observe that alternately stacking SSA modules throughout the network consistently outperforms applying SSA only once. This is because a single application of SSA before attention computation may fail to fully capture the broader cross-scale dependencies. These results highlight the importance of the proposed SSA for capturing diverse EEG dynamics across tasks.

**Attention Mechanism Comparison.** Furthermore, we compare different types of attention mechanisms to assess their impact on EEG decoding performance and computational efficiency. (1) *Dense Attention* performs full self-attention over all tokens with quadratic complexity $\mathcal{O}(N^2)$, where $N = T \times C$. While theoretically expressive, it introduces spurious dependencies, weakens representation quality in noisy EEG scenarios, and incurs substantial computational overhead. (2) *Criss-cross Attention*, which restricts attention to shared temporal or spatial indices, yielding approximate complexity $\mathcal{O}(N^{1.5})$; and (3) *Our Structured Sparse Attention (SSA)* limits attention to grouped tokens across temporal windows and brain regions, achieving linear complexity $\mathcal{O}(N \cdot k)$ with small $k \ll \sqrt{N}$. As shown in Figure 5, SSA consistently achieves the best performance across Balanced Accuracy, Cohen's Kappa, and Weighted F1. For instance, on the TUEV dataset, SSA improves Balanced Accuracy by +8.1% and Weighted F1 by +5.3% compared to Criss-cross Attention, and maintains low computational cost. These results highlight SSA as an effective and scalable inductive bias for EEG modeling.

**Topography Visualization.** To further examine the spatial dynamics captured by CSBrain, we visualize activation topographies across different EEG decoding tasks. Specifically, we apply Gradient-weighted Class Activation Mapping (Grad-CAM) [80] to compute the contribution of each EEG channel to the model's predictions, as shown in Figure 6. Different tasks elicit distinct activation patterns and scales. *Vigilance* states primarily activate the temporal and occipital lobes, indicating continuous engagement of auditory and visual systems. *Motor imagery* evokes highly localized activations over the contralateral motor cortex, reflecting ERD/ERS (event-related desynchronization / synchronization) phenomenon [81]. In contrast, *emotion recognition* and *imagined speech* exhibit broad, distributed activations. Emotion tasks elicit significant activation across the frontal and occipital lobes, consistent with the findings in [32]. Speech imagery evokes significant activation in frontal and temporal lobes [82], which are critically involved in imagined speech processing [83]. These observations further validate that different EEG decoding tasks exhibit distinct activation regions and scales, reinforcing the necessity of cross-scale modeling for EEG foundation models. By explicitly modeling cross-scale structure, CSBrain effectively adapts to task-specific neural patterns across diverse brain decoding scenarios.

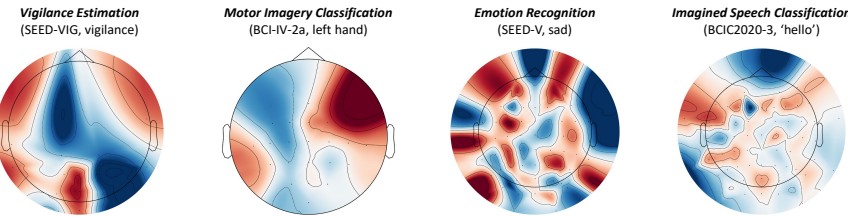

Figure 6: Topography visualization on four different tasks. More in the supplementary materials.

## 4 Conclusion

In this work, we demonstrate that modeling the cross-scale spatiotemporal structure of EEG signals is essential for building generalizable brain foundation models. Through a unified architecture composed of Cross-scale Spatiotemporal Tokenization and Structured Sparse Attention, our proposed CSBrain effectively captures neural dependencies spanning multiple spatiotemporal scales. Experimental results across 11 EEG decoding tasks and 16 datasets validate the effectiveness of CSBrain. These findings establish cross-scale spatiotemporal modeling as a critical inductive bias for robust, scalable, and physiologically-aligned EEG representation learning. We believe this work lays a solid foundation for future advancements in brain–AI integration and generalized neural decoding.

## Acknowledgements

This work was done during Yuchen Zhou's internship at Shanghai Artificial Intelligence Laboratory. This work was supported by Shanghai Artificial Intelligence Laboratory, the JC STEM Lab of AI for Science and Engineering, funded by The Hong Kong Jockey Club Charities Trust, the Research Grants Council of Hong Kong (Project No. CUHK14213224), National Natural Science Foundation of China under Grant 62373387, Natural Science Foundation of Guangdong Province (No. 2023A1515030264), Beijing Natural Science Foundation (4252048), and the Shenzhen Fundamental Research Program (Grant No. JCYJ20240813151301003).

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

# A   Related Work

Electroencephalography (EEG) provides a direct, non-invasive window into human brain activity, and plays a pivotal role in cognitive science, medical diagnostics, and brain-computer interface (BCI) systems [1–3]. By placing electrodes on the scalp, EEG captures and amplifies electrical signals produced by coordinated neuronal activity. Its high temporal fidelity, affordability, and portability make EEG especially attractive for large-scale deployment in both clinical and everyday settings [7–9]. As BCI applications expand across emotion recognition [15, 16], seizure detection [19, 20], sleep staging [17, 18], and various neurological disorders [21–23], there is a growing need for scalable, general-purpose EEG models that can robustly decode neural signals across diverse tasks and recording conditions.

## A.1   Task-Specific EEG Decoding Models

Conventional EEG decoding methods typically follow a two-stage pipeline: handcrafted feature extraction followed by shallow classification. A wide range of handcrafted features have been utilized, including time-domain statistics, frequency-domain power spectra (e.g., via Fast Fourier Transform [84]), time-frequency features (e.g., STFT [85], DWT [86]), and spatial filters such as Common Spatial Pattern [87] or Riemannian geometry-based methods [88]. These features are typically fed into classical classifiers including linear discriminant analysis [89], support vector machines [90], k-nearest neighbors [91], or logistic regression [92]. Despite their simplicity, these methods often struggle with generalization due to strong reliance on task-specific priors, limited capacity to model nonlinear dynamics, and poor robustness to inter-subject and inter-session variability.

With the rise of deep learning, end-to-end models have increasingly replaced handcrafted pipelines by directly learning representations from raw EEG signals. CNN-based architectures [24, 93, 94] have proven effective at capturing local spatiotemporal patterns, while RNNs, particularly LSTMs [28, 95, 96], are commonly adopted to model temporal dependencies. Transformers [37, 97] have recently gained popularity due to their superior ability to capture long-range interactions and global context. To combine local and global modeling, hybrid models such as CNN-LSTM [27] and CNN-Transformer [38, 36] have been explored. In parallel, GNN-based methods [31? –35] aim to incorporate topological priors by modeling spatial relationships among electrodes. While these deep models have achieved promising results in tasks such as motor imagery, emotion recognition, and sleep staging, they are typically tightly coupled to specific datasets and task formats. Adapting them to new tasks often requires retraining and architectural redesign. These limitations have motivated a shift toward EEG foundation models, which aim to learn robust and generalizable representations from unified architectures and large-scale self-supervised pretraining. Such models hold the potential to enhance generalization and enable effective transfer across tasks and domains.

## A.2   Brain Foundation Model

Foundation models have transformed fields such as natural language processing (NLP) and computer vision (CV), enabling strong generalization across tasks through unified architectures and large-scale pretraining. Models like BERT [98], GPT [99], CLIP [100], MAE [51], LLaVA [101], and SimMIM [64] demonstrate that unified architectures trained on large-scale data can be efficiently adapted to a wide range of downstream applications [102–107]. Inspired by these successes, researchers have begun to explore foundation models for neural signal processing, particularly EEG.

A number of EEG foundation models have emerged with the goal of learning generalized representations from large-scale data using unified model architectures. Early efforts such as BENDR [50], BrainBERT [108], Brant [109], EEG2Rep [110], and BIOT [40] adopt masked modeling or contrastive learning objectives to encode general-purpose neural features. LaBraM [41] further enhances cross-dataset and multi-paradigm generalization through structured tokenization and vector-quantized neural codes. More recently, EEGPT [42] introduces spatiotemporal alignment, while CBraMod [46] incorporates parallel spatial-temporal attention and dynamic positional encoding to better capture signal structure. These models represent a significant step toward generalizable EEG decoding, and most adopt a pipeline transplanted from NLP and CV, in which EEG signals are divided into fixed-scale tokens and processed via dense attention across all tokens. However, this scale-agnostic and fully dense modeling paradigm fails to align with the intrinsic cross-scale spatiotemporal nature of EEG signals, an essential property for accurately capturing task-specific neural dynamics.

To address these limitations, we argue that EEG foundation models must move beyond scale-invariant dense modeling and embrace cross-scale, structure-aware architectures that respect the inherent regional organization and temporal dynamics of brain activity. In this paper, we propose **CSBrain**, a **C**ross-scale **S**patiotemporal **B**rain foundation model for generalized EEG decoding. CSBrain achieves superior generalization across diverse EEG tasks with varying spatiotemporal demands.

## B    Additional Pretraining Details

### B.1    Hyperparameter

We summarize the detailed hyperparameter settings of CSBrain in Table 3, including the input configuration, model architecture, optimization settings, and training strategy.

Table 3: Pretraining hyperparameters for CSBrain.

| Component | Setting |
|---|---|
| Input Channels | 19 |
| Time Points per Sample | 6000 |
| Mask Ratio | 0.5 |
| Mask Token | Full-zero vector |
| Kernel Sizes (CST) | {1, 3, 5} |
| Number of Layers | 12 |
| Hidden Dimension | 200 |
| Attention Heads | 8 |
| Feed-forward Dimension | 800 |
| Dropout | 0.1 |
| Weight Initialization | Kaiming Normalization |
| Optimizer | AdamW |
| Learning Rate | 5e-4 |
| Adam $\beta$ | (0.9, 0.999) |
| Adam $\epsilon$ | 1e-8 |
| Weight Decay | 5e-2 |
| Gradient Clipping | 1.0 |
| Training Epochs | 40 |
| Batch Size | 128 |
| Learning Rate Scheduler | CosineAnnealingLR |
| Cosine Cycle Length | 40 epochs |
| Minimum Learning Rate | 1e-5 |

### B.2    Training Loss Curve

Figure 7 shows the training loss curve of CSBrain during pretraining over 40 epochs. The loss decreases rapidly in the initial phase and gradually stabilizes, demonstrating effective convergence under the masked reconstruction objective.

## C    Additional Downstream Experimental Details

### C.1    Hyperparameter Settings

For downstream finetuning, we load the pre-trained weights of CSBrain and replace the reconstruction head with a task-specific head composed of a multilayer perceptron. The learned EEG representations are flattened and fed into this head to perform downstream classification or regression. We then finetune the entire model on the target dataset. Table 4 lists the default hyperparameter settings used in downstream finetuning. We adopt binary cross-entropy (BCE) loss for binary classification, cross-entropy loss for multi-class classification, and mean squared error (MSE) for regression.

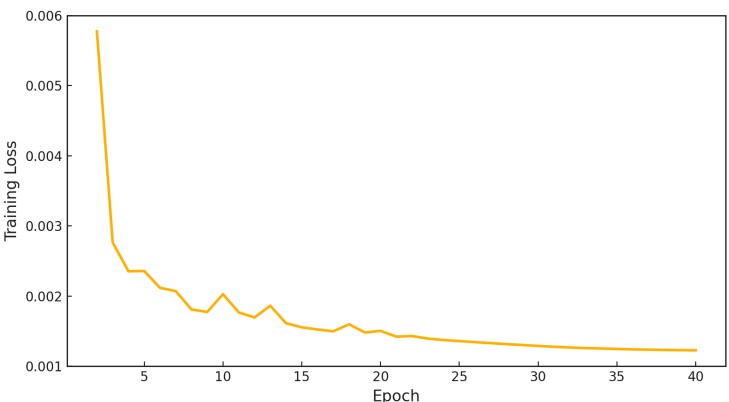

Figure 7: Training loss curve during CSBrain pretraining.

Table 4: Default hyperparameter settings for downstream finetuning.

| Component | Setting |
| --- | --- |
| Training Epochs | 50 |
| Batch Size | 64 |
| Optimizer | AdamW |
| Learning Rate | 1e-4 |
| Adam $\beta$ | (0.9, 0.999) |
| Adam $\epsilon$ | 1e-8 |
| Weight Decay | 1e-2 |
| Dropout | 0.1 |
| Learning Rate Scheduler | CosineAnnealingLR |
| Cosine Cycle Epochs | 50 |
| Minimum Learning Rate | 1e-6 |
| Label Smoothing (for multi-class) | 0.1 |

## C.2 Baselines

*Task-specific Models:*

- **EEGNet** [24] is a lightweight CNN that employs depthwise and separable convolutions for decoding EEG signals from various BCI paradigms.

- **EEGConformer** [36] is a hybrid architecture that combines CNN and transformer for EEG decoding. Specifically, the CNN module captures local spatiotemporal features, while the transformer models long-range dependencies for global temporal representation.

- **SPaRCNet** [25] is a 1D CNN with dense residual connections for EEG decoding, incorporating adaptive depth and channel selection mechanisms.

- **ContraWR** [26] converts EEG signals into multi-channel spectrograms, which are then classified using a ResNet-style 2D CNN.

- **CNN-Transformer** [38] is a CNN enhanced with transformer layers and trained using belief matching (BM) loss to detect channel-wise EEG artifacts.

- **FFCL** [27] integrates parallel CNN and LSTM branches, where the CNN extracts spatial features and the LSTM captures temporal dynamics. Features from the different branches are fused through a fully connected layer to enhance EEG classification.

- **ST-Transformer** [37] is a transformer-based architecture that applies attention mechanisms along both the feature-channel and segmented time dimensions to capture global spatiotemporal dependencies from EEG signals.

*Foundation Models:*

- **BIOT** [40][2] is a linear transformer designed for universal biosignal representation learning through a hybrid supervised–unsupervised pre-training strategy. Notably, the pre-trained BIOT accepts a maximum of 18 EEG channels as input. Therefore, for signals with more than 18 channels, a $1 \times 1$ convolution is applied to project them into an 18-channel format.
- **LaBraM** [41][3] is an EEG foundation model that learns generic representations by predicting masked neural tokens using a full-attention Transformer.
- **CBraMod** [46][4] is an EEG foundation model designed to address the heterogeneous spatiotemporal structure of EEG signals through a criss-cross Transformer backbone.

### C.3  Metrics

Consistency with prior work [40, 41, 46], we evaluate performance using the metrics as follows: **Balanced Accuracy** addresses class imbalance by averaging the recall across all classes, offering a more reliable evaluation metric than standard accuracy in imbalanced datasets:

$$\text{B-ACC} = \frac{1}{C} \sum_{i=1}^{C} \frac{TP_i}{TP_i + FN_i}, \tag{8}$$

where $C$ is the number of classes, $TP_i$ and $FN_i$ denote true positives and false negatives for class $i$.

**Cohen's Kappa** measures the agreement between predicted and true labels agreement while correcting for chance, offering insight into classifier performance beyond random predictions:

$$\kappa = \frac{p_o - p_e}{1 - p_e}, \tag{9}$$

where $p_o$ is the observed agreement and $p_e$ is the expected agreement.

**Weighted F1-score** is the harmonic mean of precision and recall for each class, weighted by the number of true instances (i.e., class support). It provides a performance measure that accounts for label imbalance in multi-class classification:

$$\text{F1-W} = \sum_{i=1}^{C} w_i \cdot \frac{2 \cdot \text{Precision}_i \cdot \text{Recall}_i}{\text{Precision}_i + \text{Recall}_i}, \tag{10}$$

where $w_i$ is the weight of class $i$ based on its support. The precision is defined as:

$$\text{Precision}_i = \frac{TP_i}{TP_i + FP_i}, \quad \text{Recall}_i = \frac{TP_i}{TP_i + FN_i}, \tag{11}$$

where $FP_i$ denotes false positives for class $i$.

**AUC-PR** evaluates the trade-off between precision and recall across various decision thresholds. It is computed as the area under the curve plotted with precision on the y-axis and recall on the x-axis:

$$\text{AUC-PR} = \int_0^1 \text{Precision}(r)\, dr, \quad r = \text{Recall}. \tag{12}$$

A higher AUC-PR indicates better performance in correctly identifying positive instances with fewer false positives. In practice, AUC-PR is typically estimated by numerical integration of discrete precision-recall points.

**AUROC** measures the model's ability to distinguish between positive and negative classes by plotting the True Positive Rate (TPR) against the False Positive Rate (FPR) at various thresholds. These are defined as:

$$\text{TPR} = \frac{TP}{TP + FN}, \quad \text{FPR} = \frac{FP}{FP + TN}. \tag{13}$$

where $TN$ denotes true negatives. The AUROC is the area under this curve:

$$\text{AUROC} = \int_0^1 \text{TPR}(f)\, df, \quad f = \text{FPR}. \tag{14}$$

---

[2]https://github.com/ycq091044/BIOT
[3]https://github.com/935963004/LaBraM
[4]https://github.com/wjq-learning/CBraMod

The AUROC is the area under the ROC curve, typically computed using numerical integration of discrete (FPR, TPR) points. An AUROC of 0.5 indicates random guessing, while 1.0 indicates perfect discrimination.

**Pearson Correlation** measures the linear dependence between predicted and true values, ranging from $-1$ (perfect negative correlation) to $+1$ (perfect positive correlation):

$$r = \frac{\sum_{i=1}^{N}(y_i - \bar{y})(\hat{y}_i - \bar{\hat{y}})}{\sqrt{\sum_{i=1}^{N}(y_i - \bar{y})^2} \cdot \sqrt{\sum_{i=1}^{N}(\hat{y}_i - \bar{\hat{y}})^2}}, \tag{15}$$

where $y_i$ is the $i$-th ground truth value, $\hat{y}_i$ is the corresponding predicted value, $\bar{y}$ and $\bar{\hat{y}}$ denote the mean of true and predicted values, respectively, and $N$ is the total number of samples.

**R2 Score** represents the proportion of variance in the dependent variable that is explained by the model, where a value of 1 indicates perfect predictive performance:

$$R^2 = 1 - \frac{\sum_{i=1}^{N}(y_i - \hat{y}_i)^2}{\sum_{i=1}^{N}(y_i - \bar{y})^2}. \tag{16}$$

An R2 score less than 0 indicates that the model performs worse than simply predicting the mean of the target variable.

**Root Mean Squared Error (RMSE)** measures the standard deviation of prediction errors, quantifying the average magnitude of differences between predicted and actual values, where lower values indicate better model fit:

$$\text{RMSE} = \sqrt{\frac{1}{N}\sum_{i=1}^{N}(y_i - \hat{y}_i)^2}. \tag{17}$$

In this work, we report Balanced Accuracy, Cohen's Kappa, and Weighted F1 for multiclass classification tasks; Balanced Accuracy, AUC-PR, and AUROC for binary classification; and Pearson correlation, R2 score, and RMSE for regression tasks.

# D   Additional Results on Downstream Tasks

To comprehensively evaluate the generalizability of our model, we conduct experiments on 11 representative BCI tasks across 16 publicly available EEG datasets. These tasks cover a wide range of applications, including *Motor Imagery Classification* (BCIC-IV-2a [66], PhysioNet-MI [67], SHU-MI [68]), *Emotion Recognition* (FACED [69], SEED-V [70]), *Seizure Detection* (CHB-MIT [71, 72], Siena [71, 73]), *Sleep Staging* (ISRUC [74], HMC [75]), *Imagined Speech Classification* (BCIC2020-3 [56]), *Vigilance Estimation* (SEED-VIG [76]), *Mental Stress Detection* (MentalArithmetic [71, 77]), *Mental Disorder Diagnosis* (Mumtaz2016 [78]), *Event Type Classification* (TUEV [79]), *Abnormal Detection* (TUAB [79]), and *Slowing Event Classification* (TUSL [79]). For all tasks, we strictly follow the established train/validation/test splits to ensure fair and reproducible evaluation. This region assignment follows a standard practice in the literature [111–115], where the 10–20 system's electrode labels are used as proxies for anatomical brain regions, specifically including the Frontal, Central, Parietal, Temporal, and Occipital areas. When existing baselines are unavailable or incompatible due to inconsistent settings, we re-implement and re-train them under a unified protocol for consistent comparison. Note that to ensure stable training, we apply minor hyperparameter adjustments (e.g., learning rate) for a few datasets when necessary. In the following sections, We detail the dataset setups for each task and analyze the corresponding results.

## D.1   Motor Imagery Classification

Motor Imagery (MI) classification aims to decode a user's intention by recognizing EEG patterns associated with imagined movements of specific body parts. It serves as a key component in BCI systems, enabling direct neural control in assistive technologies. We evaluate our model on three widely used MI datasets, **BCIC-IV-2a** [66], **PhysioNet-MI** [67], and **SHU-MI** [68], covering both multiclass and binary classification settings.

**BCIC-IV-2a** comprises EEG recordings from 9 subjects performing 4 motor imagery tasks: imagining movements of the left hand, right hand, both feet and tongue. Recordings were collected over two sessions on separate days using 22 electrodes at a sampling rate of 250 Hz. Each session contains 288 EEG trials (72 per class). Following [46], we extract the [2, 6]-second interval from each trial, yielding a total of 5,088 samples. The signals are subsequently resampled at 200 Hz. We adopt a strict subject-independent split: subject 1–5 for training, 6–7 for validation, and 8–9 for testing. We train our model with a learning rate of 0.0001 and a weight decay of 0.05. Table 5 summarizes the performance of our model and various baselines on BCIC-IV-2a. CSBrain achieves the highest performance across all three metrics: 0.5657 balanced accuracy, 0.4209 Cohen's kappa, and 0.5637 weighted F1. CSBrain outperforms the second-best CBraMod by +5.2% in balanced accuracy and +6.9% in kappa. These results highlight the effectiveness of our cross-scale spatiotemporal modeling design in capturing nuanced motor imagery dynamics.

Table 5: Results on Motor Imagery Classification (BCIC-IV-2a, 4-class).

| | Method | Balanced Accuracy | Cohen's Kappa | Weighted F1 |
|---|---|---|---|---|
| *Task-specific Models* | EEGNet [24] | 0.4482 ± 0.0094 | 0.2693 ± 0.0121 | 0.4226 ± 0.0108 |
| | EEGConformer [36] | 0.4696 ± 0.0106 | 0.2924 ± 0.0141 | 0.4533 ± 0.0128 |
| | SPaRCNet [25] | 0.4635 ± 0.0117 | 0.2847 ± 0.0147 | 0.4432 ± 0.0126 |
| | ContraWR [26] | 0.4678 ± 0.0125 | 0.2905 ± 0.0160 | 0.4413 ± 0.0142 |
| | CNN-Transformer [38] | 0.4600 ± 0.0108 | 0.2800 ± 0.0148 | 0.4460 ± 0.0114 |
| | FFCL [27] | 0.4470 ± 0.0143 | 0.2627 ± 0.0176 | 0.4238 ± 0.0139 |
| | ST-Transformer [37] | 0.4575 ± 0.0145 | 0.2733 ± 0.0198 | 0.4471 ± 0.0142 |
| *Foundation Models* | BIOT [40] | 0.4748 ± 0.0093 | 0.2997 ± 0.0139 | 0.4607 ± 0.0125 |
| | LaBraM-Base [41] | 0.4869 ± 0.0085 | 0.3159 ± 0.0154 | 0.4758 ± 0.0103 |
| | CBraMod [46] | 0.5138 ± 0.0066 | 0.3518 ± 0.0094 | 0.4984 ± 0.0085 |
| | **CSBrain (Ours)** | **0.5657 ± 0.0071** | **0.4209 ± 0.0093** | **0.5637 ± 0.0087** |

**SHU-MI** is a binary motor imagery dataset focused on classifying left-hand versus right-hand motor imagery. It contains EEG recordings from 25 subjects using 32 channels at a sampling rate of 250 Hz. SHU-MI dataset contains [4,8]-second segments extracted from each trial, corresponding to the motor imagery execution period. All signals are resampled to 200 Hz and yield a total of 11,988 samples. We adopt a subject-independent split with subjects 1–15 for training, 16–20 for validation, and 21–25 for testing. Model training is performed using a learning rate of 0.0001 and a weight decay of 0.05. Table 6 presents the results on SHU-MI. CSBrain again achieves the best performance across all metrics: 0.6417 balanced accuracy, 0.7230 AUC-PR, and 0.7200 AUROC. Compared to CBraMod, which is the strongest baseline, CSBrain improves by +0.47% in balanced accuracy and +2.12% in AUROC. The consistent gains highlight the general applicability of our model across both fine-grained and coarse-grained MI tasks.

Table 6: Results on Motor Imagery Classification (SHU-MI, 2-class).

| | Method | Balanced Accuracy | AUC-PR | AUROC |
|---|---|---|---|---|
| *Task-specific Models* | EEGNet [24] | 0.5889 ± 0.0177 | 0.6311 ± 0.0142 | 0.6283 ± 0.0152 |
| | EEGConformer [36] | 0.5900 ± 0.0107 | 0.6370 ± 0.0093 | 0.6351 ± 0.0101 |
| | SPaRCNet [25] | 0.5978 ± 0.0097 | 0.6510 ± 0.0062 | 0.6431 ± 0.0117 |
| | ContraWR [26] | 0.5873 ± 0.0128 | 0.6315 ± 0.0105 | 0.6273 ± 0.0113 |
| | CNN-Transformer [38] | 0.5975 ± 0.0169 | 0.6412 ± 0.0076 | 0.6343 ± 0.0082 |
| | FFCL [27] | 0.5692 ± 0.0252 | 0.5943 ± 0.0171 | 0.6326 ± 0.0082 |
| | ST-Transformer [37] | 0.5992 ± 0.0206 | 0.6394 ± 0.0122 | 0.6431 ± 0.0111 |
| *Foundation Models* | BIOT [40] | 0.6179 ± 0.0183 | 0.6770 ± 0.0119 | 0.6609 ± 0.0127 |
| | LaBraM-Base [41] | 0.6166 ± 0.0192 | 0.6761 ± 0.0083 | 0.6604 ± 0.0091 |
| | CBraMod [46] | 0.6370 ± 0.0151 | 0.7139 ± 0.0088 | 0.6988 ± 0.0068 |
| | **CSBrain (Ours)** | **0.6417 ± 0.0037** | **0.7230 ± 0.0079** | **0.7200 ± 0.0058** |

**PhysioNet-MI** is a large-scale motor imagery dataset comprising EEG recordings from 109 subjects, each performing four MI tasks: imagining the movement of the left fist, right fist, both fists, and both feet. The signals were collected using 64 electrodes at a sampling rate of 160 Hz. We segment the EEG into 4-second windows, resample to 200 Hz, and obtain a total of 9,837 samples. We adopt a

subject-independent split with subjects 1–70 for training, 71–89 for validation, and 90–109 for testing. For this dataset, we use a learning rate of 0.0001 and a weight decay of 0.05 for model training. As shown in Table 7, our CSBrain model achieves top performance on all three metrics: 0.6304 balanced accuracy, 0.5071 Cohen's kappa, and 0.6308 weighted F1. Compared to the best-performing baseline CBraMod, CSBrain yields consistent improvements (+1.3% in balanced accuracy and +1.7% in kappa). Furthermore, CSBrain surpasses strong task-specific models like EEGConformer and CNN-Transformer by large margins, further validating the advantage of foundation model pretraining and our cross-scale architecture.

Table 7: Results on Motor Imagery Classification (PhysioNet-MI, 4-class).

| | Method | Balanced Accuracy | Cohen's Kappa | Weighted F1 |
|---|---|---|---|---|
| *Task-specific Models* | EEGNet [24] | 0.5814 ± 0.0125 | 0.4468 ± 0.0199 | 0.5796 ± 0.0115 |
| | EEGConformer [36] | 0.6049 ± 0.0104 | 0.4736 ± 0.0171 | 0.6062 ± 0.0095 |
| | SPaRCNet [25] | 0.5932 ± 0.0152 | 0.4564 ± 0.0234 | 0.5937 ± 0.0147 |
| | ContraWR [26] | 0.5892 ± 0.0133 | 0.4527 ± 0.0248 | 0.5918 ± 0.0116 |
| | CNN-Transformer [38] | 0.6053 ± 0.0118 | 0.4725 ± 0.0223 | 0.6041 ± 0.0105 |
| | FFCL [27] | 0.5726 ± 0.0092 | 0.4323 ± 0.0182 | 0.5701 ± 0.0079 |
| | ST-Transformer [37] | 0.6035 ± 0.0081 | 0.4712 ± 0.0199 | 0.6053 ± 0.0075 |
| *Foundation Models* | BIOT [40] | 0.6153 ± 0.0154 | 0.4875 ± 0.0272 | 0.6158 ± 0.0197 |
| | LaBraM-Base [41] | 0.6173 ± 0.0122 | 0.4912 ± 0.0192 | 0.6177 ± 0.0141 |
| | CBraMod [46] | 0.6174 ± 0.0036 | 0.4898 ± 0.0048 | 0.6179 ± 0.0035 |
| | **CSBrain (Ours)** | **0.6304 ± 0.0090** | **0.5071 ± 0.0120** | **0.6308 ± 0.0095** |

## D.2 Emotion Recognition

Emotion recognition from EEG signals aims to decode a subject's affective state by analyzing brain activity patterns associated with different emotional experiences. EEG offers a direct, real-time, and non-invasive approach to capturing internal emotional responses. However, this task remains highly challenging due to the subjective nature of emotions, significant inter-subject variability, and the inherently noisy and non-stationary characteristics of EEG signals. To evaluate the effectiveness and generalizability of our model in decoding emotional states, we conduct experiments on two widely-used EEG emotion datasets: **SEED-V** [70], which involves five emotion categories and high-resolution short-segment EEG recordings across multiple sessions, and **FACED** [69], which focuses on fine-grained nine-class classification with long EEG segments.

**SEED-V** is a benchmark dataset for EEG-based emotion recognition. It includes five categories (happy, sad, neutral, disgust, fear), with EEG collected from 16 subjects over three sessions each, using 62 channels at a high sampling rate of 1000 Hz. The data is segmented into 117,744 1-second samples, resampled to 200 Hz. We divide each 15-trial session into three equal parts (5:5:5) for training, validation, and testing. The model is trained using the same settings as FACED (learning rate 0.0001, weight decay 0.05). Table 8 shows the performance on SEED-V. CSBrain again surpasses all baselines, achieving 0.4091 balanced accuracy, 0.2569 Cohen's kappa, and 0.4101 weighted F1. Compared to CBraMod, which previously achieved the best results, CSBrain improves the kappa score by +2.1%, indicating better discriminative capacity across short-time emotional states.

**FACED** is a large-scale EEG dataset for fine-grained emotion recognition, comprising 32-channel EEG signals recorded at 250 Hz from 123 subjects. It covers nine emotion categories: amusement, inspiration, joy, tenderness, anger, fear, disgust, sadness, and neutral. The raw signals are segmented into 10-second samples and resampled to 200 Hz, resulting in 10,332 instances. We split the data by subject: subjects 1–80 for training, 81–100 for validation, and 101–123 for testing. A learning rate of 0.0005 and a weight decay of 0.05 are used for model training. As shown in Table 9, CSBrain achieves state-of-the-art performance on this challenging 9-class task, with 0.5752 balanced accuracy, 0.5204 Cohen's kappa, and 0.5796 weighted F1. Compared to the best-performing baseline CBraMod, CSBrain yields consistent improvements across all metrics (+2.4% in balanced accuracy and +1.6% in kappa). These consistent gains across both long and short windowed EEG emotion datasets highlight the robustness and adaptability of CSBrain's cross-scale representation.

Table 8: Results on Emotion Recognition (SEED-V, 5-class).

| | Method | Balanced Accuracy | Cohen's Kappa | Weighted F1 |
|---|---|---|---|---|
| *Task-specific Models* | EEGNet [24] | 0.2961 ± 0.0102 | 0.1006 ± 0.0143 | 0.2749 ± 0.0098 |
| | EEGConformer [36] | 0.3537 ± 0.0112 | 0.1772 ± 0.0174 | 0.3487 ± 0.0136 |
| | SPaRCNet [25] | 0.2949 ± 0.0078 | 0.1121 ± 0.0139 | 0.2979 ± 0.0083 |
| | ContraWR [26] | 0.3546 ± 0.0105 | 0.1905 ± 0.0188 | 0.3544 ± 0.0121 |
| | CNN-Transformer [38] | 0.3678 ± 0.0078 | 0.2072 ± 0.0183 | 0.3642 ± 0.0088 |
| | FFCL [27] | 0.3641 ± 0.0092 | 0.2078 ± 0.0201 | 0.3645 ± 0.0132 |
| | ST-Transformer [37] | 0.3052 ± 0.0072 | 0.1083 ± 0.0121 | 0.2833 ± 0.0105 |
| *Foundation Models* | BIOT [40] | 0.3837 ± 0.0187 | 0.2261 ± 0.0262 | 0.3856 ± 0.0203 |
| | LaBraM-Base [41] | 0.3976 ± 0.0138 | 0.2386 ± 0.0209 | 0.3974 ± 0.0111 |
| | CBraMod [46] | 0.4091 ± 0.0097 | 0.2569 ± 0.0143 | 0.4101 ± 0.0108 |
| | **CSBrain (Ours)** | **0.4197 ± 0.0033** | **0.2785 ± 0.0034** | **0.4280 ± 0.0023** |

Table 9: Results on Emotion Recognition (FACED, 9-class).

| | Method | Balanced Accuracy | Cohen's Kappa | Weighted F1 |
|---|---|---|---|---|
| *Task-specific Models* | EEGNet [24] | 0.4090 ± 0.0122 | 0.3342 ± 0.0251 | 0.4124 ± 0.0141 |
| | EEGConformer [36] | 0.4559 ± 0.0125 | 0.3858 ± 0.0186 | 0.4514 ± 0.0107 |
| | SPaRCNet [25] | 0.4673 ± 0.0155 | 0.3978 ± 0.0289 | 0.4729 ± 0.0133 |
| | ContraWR [26] | 0.4887 ± 0.0078 | 0.4231 ± 0.0151 | 0.4884 ± 0.0074 |
| | CNN-Transformer [38] | 0.4697 ± 0.0132 | 0.4017 ± 0.0166 | 0.4702 ± 0.0125 |
| | FFCL [27] | 0.4673 ± 0.0158 | 0.3987 ± 0.0338 | 0.4699 ± 0.0145 |
| | ST-Transformer [37] | 0.4810 ± 0.0079 | 0.4137 ± 0.0133 | 0.4795 ± 0.0096 |
| *Foundation Models* | BIOT [40] | 0.5118 ± 0.0118 | 0.4476 ± 0.0254 | 0.5136 ± 0.0112 |
| | LaBraM-Base [41] | 0.5273 ± 0.0107 | 0.4698 ± 0.0188 | 0.5288 ± 0.0102 |
| | CBraMod [46] | 0.5509 ± 0.0089 | 0.5041 ± 0.0122 | 0.5618 ± 0.0093 |
| | **CSBrain (Ours)** | **0.5752 ± 0.0042** | **0.5204 ± 0.0036** | **0.5796 ± 0.0031** |

## D.3 Seizure Detection

Seizure detection aims to identify pathological brain activity associated with epileptic seizures by analyzing abnormal EEG patterns. Accurate and timely seizure detection is critical for clinical diagnosis, long-term monitoring, and closed-loop therapeutic systems. We evaluate our model on two benchmark datasets: **CHB-MIT** [71, 72], and **Siena** [71, 73].

**CHB-MIT** is a widely-used clinical EEG benchmark collected from 23 pediatric patients with intractable epilepsy at the Children's Hospital Boston. EEG recordings span multiple days per subject, captured under seizure-monitoring protocols following withdrawal of anti-seizure medication. All signals are recorded using the international 10–20 electrode placement system at 256 Hz, and labeled into binary classes: seizure and non-seizure. Following prior work [40, 46], we retain the 16 commonly used bipolar montage channels and resample all EEG signals to 200 Hz. The data is then segmented into 10-second windows, resulting in 326,993 samples. To ensure fair and consistent evaluation, we adopt a subject-independent split: subjects 1–19 for training, 20–21 for validation, and 22–23 for testing. The model is trained with a learning rate of 0.0001 and a weight decay of 0.05. Table 10 shows the performance on CHB-MIT. Our CSBrain achieves the best performance in terms of AUC-PR and AUROC, outperforming all baselines by a large margin. Specifically, it obtains an AUC-PR of 0.5164, exceeding the second-best CBraMod (0.3689) by +14.75%, and an AUROC of 0.8915, higher than CBraMod's 0.8892 (+0.23%). Although its balanced accuracy (0.7262) is marginally lower than CBraMod (0.7398), the substantial gain in AUC-PR, an especially informative metric under class imbalance, demonstrates the robustness of CSBrain for seizure detection.

**Siena** is a clinical EEG database collected from 14 adult patients at the Unit of Neurology and Neurophysiology, University of Siena. EEG recordings were acquired via Video-EEG monitoring at 512 Hz using the international 10–20 electrode placement system. Data annotation follows the International League Against Epilepsy (ILAE) criteria, with seizure and non-seizure labels rigorously verified by expert clinicians based on combined electrophysiological and clinical review. We retained the 29 EEG channels that were consistently available across all 14 subjects. All signals

Table 10: Results on Seizure Detection (CHB-MIT, 2-class).

| | Method | Balanced Accuracy | AUC-PR | AUROC |
|---|---|---|---|---|
| *Task-specific Models* | EEGNet [24] | 0.5658 ± 0.0106 | 0.1914 ± 0.0182 | 0.8048 ± 0.0136 |
| | EEGConformer [36] | 0.5976 ± 0.0141 | 0.2209 ± 0.0215 | 0.8226 ± 0.0170 |
| | SPaRCNet [25] | 0.5876 ± 0.0191 | 0.1247 ± 0.0119 | 0.8143 ± 0.0148 |
| | ContraWR [26] | 0.6344 ± 0.0002 | 0.2264 ± 0.0174 | 0.8097 ± 0.0114 |
| | CNN-Transformer [38] | 0.6389 ± 0.0067 | 0.2479 ± 0.0227 | 0.8662 ± 0.0082 |
| | FFCL [27] | 0.6262 ± 0.0104 | 0.2049 ± 0.0346 | 0.8271 ± 0.0051 |
| | ST-Transformer [37] | 0.5915 ± 0.0195 | 0.1422 ± 0.0094 | 0.8237 ± 0.0491 |
| *Foundation Models* | BIOT [40] | 0.7068 ± 0.0457 | 0.3277 ± 0.0460 | 0.8761 ± 0.0284 |
| | LaBraM-Base [41] | 0.7075 ± 0.0358 | 0.3287 ± 0.0402 | 0.8679 ± 0.0199 |
| | CBraMod [46] | **0.7398 ± 0.0284** | 0.3689 ± 0.0382 | 0.8892 ± 0.0154 |
| | **CSBrain (Ours)** | 0.7262 ± 0.0115 | **0.5164 ± 0.0449** | **0.8915 ± 0.0321** |

are resampled to 200 Hz. We segment the data into 10-second samples, yielding a total of 51,307 samples. For fair evaluation, we employ a subject-independent split: all data from the last two subjects (PN16 and PN17) are held out as the test set, while the remaining 12 subjects' data are split into train and validation sets at an 8:2 ratio, maintaining this proportion for each subject and across all label categories. Table 11 summarizes the results on the Siena dataset. CSBrain achieves the best performance across all evaluation metrics, with a balanced accuracy of 0.7662, an AUC-PR of 0.4871, and an AUROC of 0.9076. Compared to the strongest baseline CBraMod, CSBrain improves AUC-PR by 7.6 points (0.4871 vs. 0.4107) and AUROC by 0.38 points (0.9076 vs. 0.9038), while also achieving higher balanced accuracy. These results highlight the robustness of CSBrain in detecting seizures, validating the effectiveness of cross-scale spatiotemporal modeling in learning discriminative neural patterns across diverse seizure morphologies.

Table 11: Results on Seizure Detection (Siena, 2-class).

| | Method | Balanced Accuracy | AUC-PR | AUROC |
|---|---|---|---|---|
| *Task-specific Models* | EEGNet [24] | 0.7487 ± 0.0521 | 0.3753 ± 0.0867 | 0.8687 ± 0.0527 |
| | EEGConformer [36] | 0.7556 ± 0.0210 | 0.2091 ± 0.0786 | 0.8159 ± 0.0261 |
| | SPaRCNet [25] | 0.6572 ± 0.0381 | 0.3164 ± 0.0659 | 0.7334 ± 0.0857 |
| | ContraWR [26] | 0.6546 ± 0.0311 | 0.3711 ± 0.0405 | 0.7819 ± 0.0596 |
| | CNN-Transformer [38] | 0.6982 ± 0.0560 | 0.3835 ± 0.0709 | 0.8719 ± 0.0403 |
| | FFCL [27] | 0.6616 ± 0.0391 | 0.3938 ± 0.0903 | 0.8154 ± 0.1155 |
| | ST-Transformer [37] | 0.7527 ± 0.0381 | 0.3636 ± 0.0252 | 0.8884 ± 0.0091 |
| *Foundation Models* | BIOT [40] | 0.7352 ± 0.0669 | 0.3809 ± 0.0892 | 0.9029 ± 0.0304 |
| | LaBraM-Base [41] | 0.7082 ± 0.0329 | 0.3122 ± 0.0976 | 0.8814 ± 0.0328 |
| | CBraMod [46] | 0.7317 ± 0.0647 | 0.4107 ± 0.0720 | 0.9038 ± 0.0218 |
| | **CSBrain (Ours)** | **0.7662 ± 0.0471** | **0.4871 ± 0.0343** | **0.9076 ± 0.0119** |

## D.4 Sleep Staging

Sleep staging refers to the task of classifying EEG signals into standard sleep stages (Wake, N1, N2, N3, REM), providing essential information for sleep quality assessment and clinical diagnosis of sleep disorders. Compared to conventional classification tasks, sleep staging involves strong temporal dependencies and stage transition patterns across time, making sequence modeling particularly important. We evaluate our model on two benchmark datasets: **ISRUC** [74], and **HMC** [75].

**ISRUC** contains overnight polysomnographic (PSG) recordings from 100 adult subjects. We use Sub-group 1, where EEG signals are recorded using six standard channels (F3-A2, C3-A2, O1-A2, F4-A1, C4-A1, O2-A1) at 200 Hz. All signals are segmented into 30-second samples and annotated by sleep experts according to the American Academy of Sleep Medicine (AASM) criteria [116], resulting in 89,240 labeled samples across five classes. Following prior work [117], we treat sleep staging as a sequence-to-sequence task: EEG segments are grouped into sequences of 20 30-second samples, and models are trained to predict one sleep stage per sample. Following prior work [46], we adopt a subject-independent split for evaluation: subjects 1–80 are used for training, 81–90 for

validation, and 91–100 for testing. For consistency across models, all sample encoders (including CSBrain and baselines) are followed by a one-layer Transformer as a shared sequence encoder. Table 12 reports the performance on ISRUC. CSBrain achieves the highest balanced accuracy of 0.7925 and strong overall performance across Cohen's kappa (0.7406) and weighted F1 (0.7990). Compared to CBraMod, CSBrain improves balanced accuracy by 0.6 points (0.7925 vs. 0.7865), while maintaining highly competitive scores in the other two metrics.

Table 12: Results on Sleep Staging (ISRUC, 5-class).

| | Method | Balanced Accuracy | Cohen's Kappa | Weighted F1 |
|---|---|---|---|---|
| *Task-specific Models* | EEGNet [24] | 0.7154 ± 0.0121 | 0.7040 ± 0.0173 | 0.7513 ± 0.0124 |
| | EEGConformer [36] | 0.7400 ± 0.0133 | 0.7143 ± 0.0162 | 0.7634 ± 0.0151 |
| | SPaRCNet [25] | 0.7487 ± 0.0075 | 0.7097 ± 0.0132 | 0.7624 ± 0.0092 |
| | ContraWR [26] | 0.7402 ± 0.0126 | 0.7178 ± 0.0156 | 0.7610 ± 0.0137 |
| | CNN-Transformer [38] | 0.7363 ± 0.0087 | 0.7129 ± 0.0121 | 0.7719 ± 0.0153 |
| | FFCL [27] | 0.7277 ± 0.0182 | 0.7016 ± 0.0292 | 0.7614 ± 0.0197 |
| | ST-Transformer [37] | 0.7381 ± 0.0205 | 0.7013 ± 0.0352 | 0.7681 ± 0.0175 |
| *Foundation Models* | BIOT [40] | 0.7527 ± 0.0121 | 0.7291 ± 0.0230 | 0.7790 ± 0.0146 |
| | LaBraM-Base [41] | 0.7633 ± 0.0102 | 0.7231 ± 0.0182 | 0.7810 ± 0.0133 |
| | CBraMod [46] | 0.7865 ± 0.0110 | **0.7442 ± 0.0152** | **0.8011 ± 0.0099** |
| | **CSBrain (Ours)** | **0.7925 ± 0.0030** | 0.7406 ± 0.0102 | 0.7990 ± 0.0091 |

**HMC** contains overnight polysomnography (PSG) recordings of 151 adult subjects. EEG signals were recorded from four channels (F4-M1, C4-M1, O2-M1, and C3-M2) at 256 Hz. All signals were segmented into 30-second epochs and annotated by sleep experts according to the AASM guidelines, resulting in 137,243 labeled samples across five classes. All signals are resampled to 200 Hz. Following prior work [118], we adopt a subject-independent evaluation split: the first 100 subjects are used for training, the middle 25 for validation, and the remaining 26 for testing. Table 13 summarizes the performance of all methods on HMC. CSBrain achieves the highest balanced accuracy of 0.7345, along with top scores in Cohen's kappa (0.6818) and weighted F1 (0.7506), outperforming all baselines. Compared to the best-performing baseline LaBraM-Base, CSBrain improves balanced accuracy by 0.68 points and weighted F1 by 0.52 points, while maintaining a comparable Cohen's kappa score.

Table 13: Results on Sleep Staging (HMC, 5-class).

| | Method | Balanced Accuracy | Cohen's Kappa | Weighted F1 |
|---|---|---|---|---|
| *Task-specific Models* | EEGNet [24] | 0.6534 ± 0.0122 | 0.5886 ± 0.0201 | 0.6536 ± 0.0168 |
| | EEGConformer [36] | 0.7149 ± 0.0086 | 0.6432 ± 0.0055 | 0.7080 ± 0.0039 |
| | SPaRCNet [25] | 0.4756 ± 0.1109 | 0.3147 ± 0.1315 | 0.4108 ± 0.1310 |
| | ContraWR [26] | 0.4242 ± 0.0541 | 0.2340 ± 0.0554 | 0.2987 ± 0.0288 |
| | CNN-Transformer [38] | 0.6573 ± 0.0141 | 0.5961 ± 0.0105 | 0.6896 ± 0.0065 |
| | FFCL [27] | 0.4427 ± 0.0702 | 0.2542 ± 0.0654 | 0.2902 ± 0.0485 |
| | ST-Transformer [37] | 0.2559 ± 0.0141 | 0.0503 ± 0.0183 | 0.1428 ± 0.0122 |
| *Foundation Models* | BIOT [40] | 0.6862 ± 0.0041 | 0.6295 ± 0.0113 | 0.7091 ± 0.0147 |
| | LaBraM-Base [41] | 0.7277 ± 0.0101 | 0.6813 ± 0.0053 | 0.7454 ± 0.0027 |
| | CBraMod [46] | 0.7269 ± 0.0041 | 0.6685 ± 0.0104 | 0.7395 ± 0.0089 |
| | **CSBrain (Ours)** | **0.7345 ± 0.0047** | **0.6818 ± 0.0046** | **0.7506 ± 0.0042** |

### D.5 Imagined Speech Classification

Imagined speech classification aims to decode phonological representations embedded in neural activity. This task enables individuals with speech impairments resulting from conditions such as stroke, trauma, or amyotrophic lateral sclerosis (ALS) to communicate without overt articulation. Moreover, this approach enhances the understanding of the neural mechanisms underlying speech production and perception, facilitating advances in cognitive neuroscience and augmentative communication technologies. We evaluate our model on the **BCIC2020-3** dataset [56], released as part of the 2020 International BCI Competition. During the data collection experiment, 15 subjects were instructed to

imagine five speech-related words or phrases ("hello," "help me," "stop," "thank you," and "yes"), and EEG signals were recorded from 64 channels at a sampling rate of 256 Hz. BCIC2020-3 dataset is split into training, validation, and test sets as defined by the competition. Specifically, for each subject, 60 trials per class are provided for training, 10 for validation, and 10 for testing. Each sample consists of a 3-second EEG recording. Therefore, we obtained 6000 64-channel 3-second EEG trials, which are resampled to 200 Hz. A learning rate of 0.0001 and a weight decay of 0.01 are used for our model training. Table 14 reports the performance on BCIC2020-3. CSBrain achieves the highest balanced accuracy (0.6004), along with strong overall performance in terms of Cohen's kappa (0.5006) and weighted F1 (0.6003). Compared to CBraMod, CSBrain improves the balanced accuracy by 6.3 points (0.6004 vs. 0.5373), while maintaining competitive performance on the other two metrics.

Table 14: Results on Imagined Speech Classification (BCIC2020-3, 5-class).

| | Method | Balanced Accuracy | Cohen's Kappa | Weighted F1 |
|---|---|---|---|---|
| *Task-specific Models* | EEGNet [24] | 0.4413 ± 0.0096 | 0.3016 ± 0.0123 | 0.4413 ± 0.0102 |
| | EEGConformer [36] | 0.4506 ± 0.0133 | 0.3133 ± 0.0183 | 0.4488 ± 0.0154 |
| | SPaRCNet [25] | 0.4426 ± 0.0156 | 0.3033 ± 0.0233 | 0.4420 ± 0.0108 |
| | ContraWR [26] | 0.4257 ± 0.0162 | 0.3078 ± 0.0218 | 0.4407 ± 0.0182 |
| | CNN-Transformer [38] | 0.4533 ± 0.0092 | 0.3166 ± 0.0118 | 0.4506 ± 0.0127 |
| | FFCL [27] | 0.4678 ± 0.0197 | 0.3301 ± 0.0359 | 0.4689 ± 0.0205 |
| | ST-Transformer [37] | 0.4126 ± 0.0122 | 0.2941 ± 0.0159 | 0.4247 ± 0.0138 |
| *Foundation Models* | BIOT [40] | 0.4920 ± 0.0086 | 0.3650 ± 0.0176 | 0.4917 ± 0.0079 |
| | LaBraM-Base [41] | 0.5060 ± 0.0155 | 0.3800 ± 0.0242 | 0.5054 ± 0.0205 |
| | CBraMod [46] | 0.5373 ± 0.0108 | 0.4216 ± 0.0163 | 0.5383 ± 0.0096 |
| | **CSBrain (Ours)** | **0.6004 ± 0.0187** | **0.5006 ± 0.0233** | **0.6003 ± 0.0192** |

## D.6 Vigilance Estimation

Vigilance estimation plays a critical role in monitoring cognitive states in real-world scenarios such as driving, aviation, and industrial operations. By measuring fatigue levels from EEG signals, this approach can enhance safety by reducing the risk of accidents in high-stakes environments. Advances in vigilance estimation may facilitate the development of more robust and reliable monitoring systems. We evaluate our model on the **SEED-VIG** dataset [76]. SEED-VIG simulates a real-world driving environment to induce a gradual transition from a vigilance to a fatigued state. The dataset was collected while subjects operated the simulated driving system. Vigilance levels were labeled using the PERCLOS indicator obtained from SMI eye-tracking glasses. EEG signals were recorded from 21 subjects using 17 channels at 200 Hz, and segmented into 20,355 8-second samples. We adopt a subject-independent evaluation split: subject 1 to 13 for training, subject 14 to 17 for validation and subject 18 to 21 for test. To stabilize training, we use a learning rate of 0.0005 and a weight decay of 0.05, slightly differing from the default settings. As shown in Table 15, CSBrain achieves the best overall performance, outperforming all baselines in both $R^2$ score (0.2363) and RMSE (0.2774), while achieving a competitive Pearson's correlation coefficient (0.6314) close to the strongest baseline (LaBraM, 0.6347). Compared to the best-performing foundation model LaBraM, CSBrain improves $R^2$ by a substantial margin (+5.6 points) and reduces RMSE by 0.97, highlighting the effectiveness of its cross-scale modeling strategy in capturing subtle fluctuations in cognitive state.

## D.7 Mental Stress Detection

Mental stress detection holds significant potential for both clinical and real-world applications. It enables objective assessment of psychological stress levels, overcoming the limitations of subjective self-reports. Therefore, it is particularly valuable for workplace mental health monitoring and the diagnosis of stress-related disorders. We evaluate our model on the **MentalArithmetic** dataset [71, 77]. MentalArithmetic comprises EEG recordings from 36 subjects collected before and during a mental arithmetic task. EEG trials were recorded before the task were labeled as "no mental stress," while those were recorded during the task were labeled as "mental stress." All EEG signals were recorded from 20 electrodes at a sampling rate of 500Hz. The signals were resampled to 200Hz and segmented into 1,707 samples of 5-second trails. We adopt a subject-independent evaluation

Table 15: Results on Vigilance Estimation (SEED-VIG, regression).

| | Method | Pearson's Correlation | $R^2$ Score | RMSE↓ |
|---|---|---|---|---|
| *Task-specific Models* | EEGNet [24] | 0.5127 ± 0.0357 | 0.1960 ± 0.0427 | 0.2847 ± 0.0076 |
| | EEGConformer [36] | 0.5800 ± 0.0174 | 0.2065 ± 0.0230 | 0.2829 ± 0.0041 |
| | SPaRCNet [25] | 0.5709 ± 0.0362 | 0.2185 ± 0.0601 | 0.2806 ± 0.0108 |
| | ContraWR [26] | 0.5235 ± 0.0335 | 0.0727 ± 0.0540 | 0.3057 ± 0.0090 |
| | CNN-Transformer [38] | 0.5829 ± 0.0246 | 0.1796 ± 0.0105 | 0.2877 ± 0.0018 |
| | FFCL [27] | 0.4923 ± 0.0313 | 0.1740 ± 0.0530 | 0.2885 ± 0.0093 |
| | ST-Transformer [37] | 0.6020 ± 0.0327 | 0.1138 ± 0.1151 | 0.2983 ± 0.0199 |
| *Foundation Models* | BIOT [40] | 0.6114 ± 0.0169 | 0.1232 ± 0.0778 | 0.2971 ± 0.0128 |
| | LaBraM-Base [41] | **0.6347 ± 0.0135** | 0.1808 ± 0.0958 | 0.2871 ± 0.0166 |
| | CBraMod [46] | 0.5502 ± 0.0115 | 0.0737 ± 0.0167 | 0.3057 ± 0.0027 |
| | **CSBrain (Ours)** | 0.6314 ± 0.0356 | **0.2363 ± 0.0519** | **0.2774 ± 0.0094** |

split: subject 1 to 28 for training, subject 29 to 32 for validation, and subject 33 to 36 for testing. Table 16 reports the performance on MentalArithmetic. CSBrain achieves the highest balanced accuracy (0.7558), along with strong overall performance in terms of AUC-PR (0.6696) and AUROC (0.8478). Compared to CBraMod, CSBrain improves the balanced accuracy by 3.0 points (0.7558 vs. 0.7256), while maintaining comparable performance on the other two metrics. The superior results of CSBrain can be attributed to its cross-scale spatiotemporal modeling, which effectively captures the short-term neural fluctuations and longer-range contextual dependencies underlying stress-induced brain dynamics. These findings highlight the importance of structure-aware foundation models in cognitive state classification tasks.

Table 16: Results on Mental Stress Detection (MentalArithmetic, 2-class).

| | Method | Balanced Accuracy | AUC-PR | AUROC |
|---|---|---|---|---|
| *Task-specific Models* | EEGNet [24] | 0.6770 ± 0.0116 | 0.5763 ± 0.0102 | 0.7321 ± 0.0108 |
| | EEGConformer [36] | 0.6805 ± 0.0123 | 0.5829 ± 0.0134 | 0.7424 ± 0.0128 |
| | SPaRCNet [25] | 0.6879 ± 0.0107 | 0.5825 ± 0.0193 | 0.7418 ± 0.0132 |
| | ContraWR [26] | 0.6631 ± 0.0097 | 0.5787 ± 0.0164 | 0.7332 ± 0.0082 |
| | CNN-Transformer [38] | 0.6779 ± 0.0268 | 0.5777 ± 0.0285 | 0.7258 ± 0.0336 |
| | FFCL [27] | 0.6798 ± 0.0142 | 0.5786 ± 0.0266 | 0.7330 ± 0.0198 |
| | ST-Transformer [37] | 0.6631 ± 0.0173 | 0.5672 ± 0.0259 | 0.7132 ± 0.0174 |
| *Foundation Models* | BIOT [40] | 0.6875 ± 0.0186 | 0.6004 ± 0.0195 | 0.7536 ± 0.0144 |
| | LaBraM-Base [41] | 0.6909 ± 0.0125 | 0.5999 ± 0.0155 | 0.7721 ± 0.0093 |
| | CBraMod [46] | 0.7256 ± 0.0132 | 0.6267 ± 0.0099 | 0.7905 ± 0.0073 |
| | **CSBrain (Ours)** | **0.7558 ± 0.0106** | **0.6696 ± 0.0221** | **0.8478 ± 0.0297** |

## D.8 Mental Disorder Diagnosis

Mental disorder diagnosis serves as a valuable diagnostic tool for conditions such as depression, schizophrenia, and bipolar disorder. These disorders are often associated with characteristic abnormalities in EEG spectral features or connectivity patterns across different episodes or disease stages, reflecting underlying neural dysfunction. Owing to its non-invasive nature and low cost, EEG is expected to partially compensate for the subjectivity and limitations of traditional diagnostic approaches. We evaluate our model on the **Mumtaz2016** dataset [78]. Mumtaz2016 includes EEG recordings from 34 patients diagnosed with major depressive disorder (MDD) and 30 healthy control (HC) participants. The data collection experiment includes three sessions: eyes-open, eyes-closed, and task. Following prior work [46], only the eyes-open and eyes-closed sessions are used in this work. EEG signals were recorded from 19 electrodes at a sampling rate of 256 Hz. All signals are resampled to 200 Hz. Subsequently, the signals are segmented into 5-second 7,143 trails. We adopt a subject-independent evaluation split: data from 24 MDD patients and 19 HCs are used for training, 5 MDD patients and 4 HCs for validation, and 5 MDD patients and 5 HCs for testing. Table 17 reports the performance on Mumtaz2016. CSBrain achieves the highest balanced accuracy (0.9643), along with strong overall performance in terms of AUC-PR (0.9936) and AUROC (0.9956). Compared

to CBraMod, CSBrain improves the balanced accuracy by 0.8 points (0.9643 vs. 0.9560), while maintaining comparable performance on the other two metrics.

Table 17: Results on Mental Disorder Diagnosis (Mumtaz2016, 2-class).

| | Method | Balanced Accuracy | AUC-PR | AUROC |
|---|---|---|---|---|
| *Task-specific Models* | EEGNet [24] | 0.9232 ± 0.0104 | 0.9626 ± 0.0095 | 0.9639 ± 0.0093 |
| | EEGConformer [36] | 0.9308 ± 0.0117 | 0.9684 ± 0.0105 | 0.9702 ± 0.0101 |
| | SPaRCNet [25] | 0.9316 ± 0.0095 | 0.9754 ± 0.0065 | 0.9781 ± 0.0063 |
| | ContraWR [26] | 0.9195 ± 0.0115 | 0.9589 ± 0.0102 | 0.9621 ± 0.0092 |
| | CNN-Transformer [38] | 0.9305 ± 0.0068 | 0.9757 ± 0.0074 | 0.9742 ± 0.0059 |
| | FFCL [27] | 0.9314 ± 0.0083 | 0.9717 ± 0.0021 | 0.9753 ± 0.0033 |
| | ST-Transformer [37] | 0.9135 ± 0.0103 | 0.9578 ± 0.0086 | 0.9594 ± 0.0059 |
| *Foundation Models* | BIOT [40] | 0.9358 ± 0.0052 | 0.9736 ± 0.0034 | 0.9758 ± 0.0042 |
| | LaBraM-Base [41] | 0.9409 ± 0.0079 | 0.9798 ± 0.0093 | 0.9782 ± 0.0057 |
| | CBraMod [46] | 0.9560 ± 0.0056 | 0.9923 ± 0.0032 | 0.9921 ± 0.0025 |
| | **CSBrain (Ours)** | **0.9643 ± 0.0155** | **0.9936 ± 0.0034** | **0.9956 ± 0.0020** |

## D.9 Event Type Classification

Event type classification holds significant clinical and research potential, particularly in neurological disorder diagnosis and monitoring. **TUEV** [79] is an EEG dataset commonly used to evaluate the event type classification performance of decoding models. It contains annotations of EEG segments categorized into six classes: (1) spike and sharp wave (SPSW), (2) generalized periodic epileptiform discharges (GPED), (3) periodic lateralized epileptiform discharges (PLED), (4) eye movement (EYEM), (5) artifact (ARTF), and (6) background (BCKG). The EEG signals are recorded from 23 channels at 256 Hz. Following prior work [46], we use 16 commonly adopted bipolar montage channels based on the international 10–20 system for TUEV. All signals are resampled to 200 Hz. Subsequently, the signals are segmented into 112,491 5-second samples. The original dataset provides predefined training and test splits. We further split the training subjects into training and validation sets using an 8:2 ratio. Table 18 reports the performance comparison on TUEV. CSBrain achieves the highest balanced accuracy (0.6903) and Cohen's kappa (0.6833), while maintaining a competitive weighted F1 score (0.8333) close to CBraMod (0.8342). Compared to LaBraM and CBraMod, CSBrain yields consistent improvements in overall classification robustness, particularly in capturing minority classes, as reflected by gains in balanced accuracy. By explicitly modeling these cross-scale dependencies, CSBrain offers a more reliable decoding paradigm for real-world clinical EEG interpretation.

Table 18: Results on Event Type Classification (TUEV, 6-class).

| | Method | Balanced Accuracy | Cohen's Kappa | Weighted F1 |
|---|---|---|---|---|
| *Task-specific Models* | EEGNet [24] | 0.3876 ± 0.0143 | 0.3577 ± 0.0155 | 0.6539 ± 0.0120 |
| | EEGConformer [36] | 0.4074 ± 0.0164 | 0.3967 ± 0.0195 | 0.6983 ± 0.0152 |
| | SPaRCNet [25] | 0.4161 ± 0.0262 | 0.4233 ± 0.0181 | 0.7024 ± 0.0104 |
| | ContraWR [26] | 0.4384 ± 0.0349 | 0.3912 ± 0.0237 | 0.6893 ± 0.0136 |
| | CNN-Transformer [38] | 0.4087 ± 0.0161 | 0.3815 ± 0.0134 | 0.6854 ± 0.0293 |
| | FFCL [27] | 0.3979 ± 0.0104 | 0.3732 ± 0.0188 | 0.6783 ± 0.0120 |
| | ST-Transformer [37] | 0.3984 ± 0.0228 | 0.3765 ± 0.0306 | 0.6823 ± 0.0190 |
| *Foundation Models* | BIOT [40] | 0.5281 ± 0.0225 | 0.5273 ± 0.0249 | 0.7492 ± 0.0082 |
| | LaBraM-Base [41] | 0.6409 ± 0.0065 | 0.6637 ± 0.0093 | 0.8312 ± 0.0052 |
| | CBraMod [46] | 0.6671 ± 0.0107 | 0.6772 ± 0.0096 | **0.8342 ± 0.0064** |
| | **CSBrain (Ours)** | **0.6903 ± 0.0059** | **0.6833 ± 0.0047** | 0.8333 ± 0.0057 |

## D.10 Abnormal Detection

Abnormal detection enables the identification of pathological neuroelectric activity, which can significantly reduce the workload of medical personnel while ensuring timely alerts for various

abnormal EEG events through continuous monitoring. **TUAB** [79] is an EEG dataset commonly used to evaluate the abnormal detection performance of decoding models. All EEG signals were recorded from 23 channels at 256 Hz and annotated as either normal or abnormal. Following prior work [46], we use 16 commonly adopted bipolar montage channels based on the international 10–20 system for TUAB. All signals are resampled to 200 Hz. Subsequently, the signals are segmented into 409,455 10-second samples. The original dataset provides predefined training and test splits. We further split the training subjects into training and validation sets using an 8:2 ratio. As shown in Table 19, CSBrain achieves the highest balanced accuracy of 0.8172 and the best AUC-PR of 0.9005, while maintaining a strong AUROC of 0.8957. Compared to the strongest baseline LaBraM, CSBrain improves AUC-PR by 0.4 points and balanced accuracy by 0.32 points, with slightly lower AUROC. This performance demonstrates that CSBrain consistently enhances the model's ability to discriminate subtle abnormalities across patients, despite class imbalance and recording variability.

Table 19: Results on Abnormal Detection (TUAB, 2-class).

| | Method | Balanced Accuracy | AUC-PR | AUROC |
|---|---|---|---|---|
| *Task-specific Models* | EEGNet [24] | 0.7642 ± 0.0036 | 0.8299 ± 0.0043 | 0.8412 ± 0.0031 |
| | EEGConformer [36] | 0.7758 ± 0.0049 | 0.8427 ± 0.0054 | 0.8445 ± 0.0038 |
| | SPaRCNet [25] | 0.7896 ± 0.0018 | 0.8414 ± 0.0018 | 0.8676 ± 0.0012 |
| | ContraWR [26] | 0.7746 ± 0.0041 | 0.8421 ± 0.0104 | 0.8456 ± 0.0074 |
| | CNN-Transformer [38] | 0.7777 ± 0.0022 | 0.8433 ± 0.0039 | 0.8461 ± 0.0013 |
| | FFCL [27] | 0.7848 ± 0.0038 | 0.8448 ± 0.0065 | 0.8569 ± 0.0051 |
| | ST-Transformer [37] | 0.7966 ± 0.0023 | 0.8521 ± 0.0026 | 0.8707 ± 0.0019 |
| *Foundation Models* | BIOT [40] | 0.7959 ± 0.0057 | 0.8792 ± 0.0023 | 0.8815 ± 0.0043 |
| | LaBraM-Base [41] | 0.8140 ± 0.0019 | 0.8965 ± 0.0016 | **0.9022 ± 0.0009** |
| | CBraMod [46] | 0.7891 ± 0.0030 | 0.8636 ± 0.0063 | 0.8606 ± 0.0057 |
| | **CSBrain (Ours)** | **0.8172 ± 0.0043** | **0.9005 ± 0.0066** | 0.8957 ± 0.0046 |

## D.11 Slowing Event Classification

**TUSL** [79] is a clinically relevant benchmark for slowing event classification, containing three classes: seizure, slowing, and complex background. Following previous work [118], the raw EEG signals from 23 channels were retained, segmented into 10-second samples, and resampled to 200 Hz, resulting in 245 samples. The dataset is split into training, validation, and test sets in a 6:2:2 ratio. Table 20 presents the performance comparison on TUSL. CSBrain outperforms all baselines by a notable margin, achieving a balanced accuracy of 0.8571, Cohen's kappa of 0.7828, and weighted F1 of 0.8568. Compared to the strongest baseline LaBraM, CSBrain improves balanced accuracy by 9.46 points and Cohen's kappa by 14.21 points. These results indicate that CSBrain's cross-scale spatiotemporal modeling strategy enables more reliable discrimination of subtle, short-lived slowing patterns from background and seizure activity, which is a key requirement in clinical EEG event classification.

Table 20: Results on Slowing Event Classification (TUSL, 3-class).

| | Method | Balanced Accuracy | Cohen's Kappa | Weighted F1 |
|---|---|---|---|---|
| *Task-specific Models* | EEGNet [24] | 0.4562 ± 0.0729 | 0.1955 ± 0.1115 | 0.4405 ± 0.0960 |
| | EEGConformer [36] | 0.5512 ± 0.0666 | 0.3072 ± 0.0877 | 0.4895 ± 0.0733 |
| | SPaRCNet [25] | 0.4185 ± 0.0452 | 0.1399 ± 0.0799 | 0.3500 ± 0.0968 |
| | ContraWR [26] | 0.5857 ± 0.0662 | 0.3567 ± 0.0968 | 0.5458 ± 0.0798 |
| | CNN-Transformer [38] | 0.3575 ± 0.0151 | 0.0306 ± 0.0179 | 0.2235 ± 0.0251 |
| | FFCL [27] | 0.3159 ± 0.0688 | 0.0628 ± 0.0880 | 0.2120 ± 0.0786 |
| | ST-Transformer [37] | 0.4000 ± 0.0329 | 0.0866 ± 0.0449 | 0.3793 ± 0.0459 |
| *Foundation Models* | BIOT [40] | 0.5785 ± 0.0303 | 0.2012 ± 0.0212 | 0.2394 ± 0.0404 |
| | LaBraM-Base [41] | 0.7625 ± 0.0131 | 0.6407 ± 0.0304 | 0.7614 ± 0.0210 |
| | CBraMod [46] | 0.7388 ± 0.0320 | 0.6149 ± 0.0550 | 0.7453 ± 0.0360 |
| | **CSBrain (Ours)** | **0.8571 ± 0.0240** | **0.7828 ± 0.0270** | **0.8568 ± 0.0180** |

# E Additional Analysis and Ablations

## E.1 The Effectiveness of Pretraining

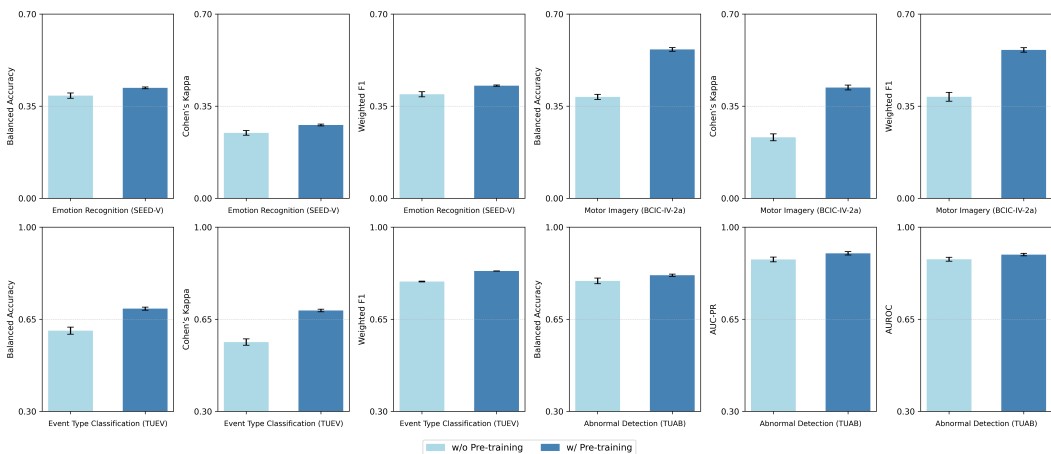

Figure 8: Performance comparison between CSBrain without and with pretraining.

We evaluate the effectiveness of large-scale pretraining by comparing CSBrain with and without pretraining across four representative EEG decoding tasks. The non-pre-trained model uses the same architecture but is trained from scratch on the downstream dataset only. Figure 8 summarizes the performance across three standard metrics. Pretraining leads to consistent performance gains across all datasets, confirming its general effectiveness. The improvement is especially pronounced in data-limited scenarios. On BCIC-IV-2a (5,000 samples), pretraining improves balanced accuracy from 0.39 to 0.57, Cohen's kappa from 0.23 to 0.42, and weighted F1 from 0.39 to 0.56. This demonstrates that pretraining is especially beneficial when downstream data is scarce and task difficulty is high. On SEED-V and TUEV, both of which contain over 100,000 samples, pretraining still yields substantial improvements. For TUAB, which contains over 400,000 labeled EEG samples, the gains from pretraining are relatively modest. This suggests that when downstream training data is sufficiently large and diverse, training from scratch becomes more competitive. Nevertheless, pretraining still brings consistent improvements.

In summary, large-scale pretraining is universally beneficial across tasks. The performance gain is especially prominent in data-limited settings, while still being meaningful on larger-scale datasets. These results support the core motivation of EEG foundation models: leveraging unlabeled EEG at scale to improve downstream performance, particularly when labeled data is limited.

## E.2 Scaling Analysis

As a foundation model for EEG decoding, CSBrain is expected to benefit from larger pretraining datasets and higher model capacities. To examine this, we perform a scaling analysis from two perspectives: (i) the amount of pretraining data, and (ii) the model capacity (number of parameters).

**Effect of Pretraining Data Volume.** We vary the scale of unlabeled EEG data used for pretraining, ranging from 10 hours to 9000 hours. As shown in Figure 9, model performance improves consistently with increased pretraining data. Accuracy, Cohen's kappa, and F1 score all show steady gains as data size increases, particularly between 500 and 5000 hours. Beyond 5000 hours, the improvement begins to plateau, suggesting diminishing returns under fixed model capacity.

**Effect of Model Capacity.** We investigate the impact of model size by varying the number of Transformer layers in CSBrain from 2 to 12, which corresponds to model sizes from 1.4M to 4.9M parameters. As illustrated in Figure 10, downstream performance improves steadily with larger model capacity, indicating that deeper networks provide stronger representational power for EEG decoding.

Together, these results follow the general trend of neural scaling laws [119], where both data and model scale contribute to performance improvements. While our current exploration is constrained

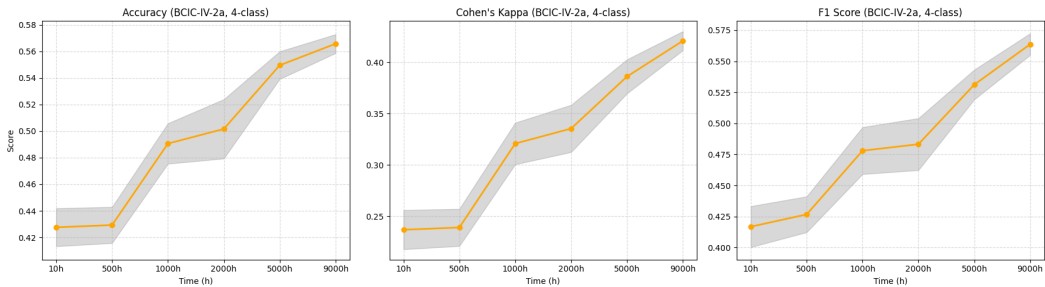

Figure 9: Effect of pretraining data volume on CSBrain performance.

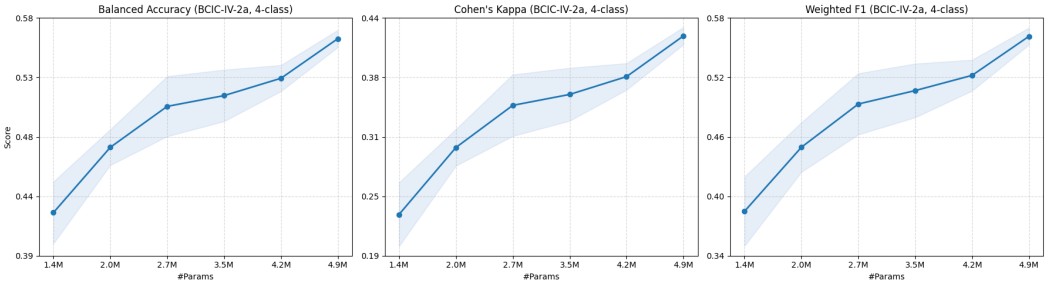

Figure 10: Effect of model parameter size on CSBrain performance.

by computational resources and may not reach the scaling limits, we believe further scaling in both dimensions will continue to yield gains. Determining how much EEG data is required to effectively pretrain large-scale EEG foundation models remains a critical open question. We call for broader community efforts to collaboratively explore this direction, in order to unlock the full potential of unified architectures and large-scale pretraining for EEG-based foundation modeling.

## E.3 Ablation on Regional Descriptor

We conduct an ablation study to evaluate the impact of different regional descriptor designs in the inter-region attention module. In our default configuration, for each region $R_r$, we construct a descriptor by combining two components: a sequentially sampled token $x_r^{\text{rep}}$ and the mean-pooled feature across all tokens in $R_r$. This hybrid design aims to preserve both localized temporal structure and global regional context. Here, we compare four variants: **(1) Token only**: Using only the sequentially sampled token $x_r^{\text{rep}}$ from each region. **(2) Mean only**: Using only the average feature of all tokens in each region. **(3) Random Sample**: Randomly sampling a token from each region, and combining it with the regional average. **(4) Ours**: Sequentially sampling a token from each region and combining it with the average feature.

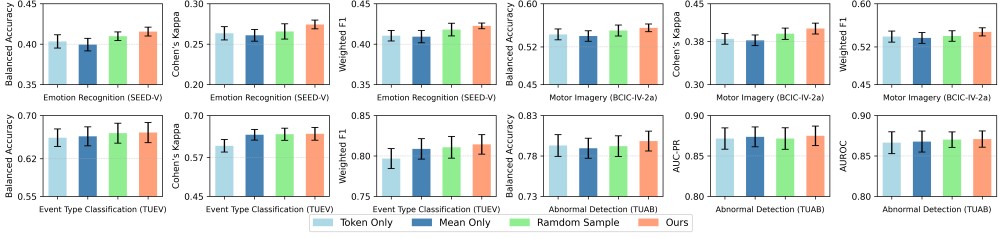

Figure 11: Ablation on regional descriptor design.

Figure 11 reports the results of our ablation study on regional descriptor design across four representative tasks. The default setting consistently achieves the best performance across all tasks and metrics. Using only the token or only the mean leads to noticeable drops in accuracy and robustness,

indicating that both local and regional information is essential. Random sampling performs slightly better than the single-source variants but still underperforms the default design. These results confirm that combining local token features with region-level context enhances inter-region attention, while sequential sampling (covering partition) ensures consistent spatial coverage.

# F  Additional Visualizations

## F.1  Topography Visualization

In this section, we present topography visualizations for four representative datasets: SEED-VIG, BCIC-IV-2a, SEED-V, and BCIC2020-3. Specifically, we apply Gradient-weighted Class Activation Mapping (Grad-CAM) [80] to estimate the contribution of each EEG channel to the model's predictions. Across different EEG tasks, the visualizations reveal distinct activation topographies, which reflect task-specific activation patterns and demonstrate the model's ability to capture spatially meaningful representations.

For the **SEED-VIG** dataset, as shown in Figure 12, vigilance states primarily activate the temporal and occipital lobes, indicating sustained engagement of the auditory and visual systems. In contrast, the fatigued state exhibits reduced activation in these lobes, suggesting a marked decline in auditory and visual information processing.

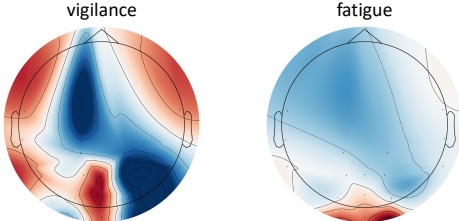

Figure 12: Topography visualization of SEED-VIG.

For the **BCIC-IV-2a** dataset, as shown in Figure 13, left- or right-hand motor imagery induces localized activation in the contralateral motor cortex, whereas motor imagery involving both feet or the tongue results in more symmetrical activation across the bilateral motor cortex. These findings suggest that motor imagery effectively modulates neural activity in the motor cortex, as reflected by event-related desynchronization (ERD) and synchronization (ERS) [81].

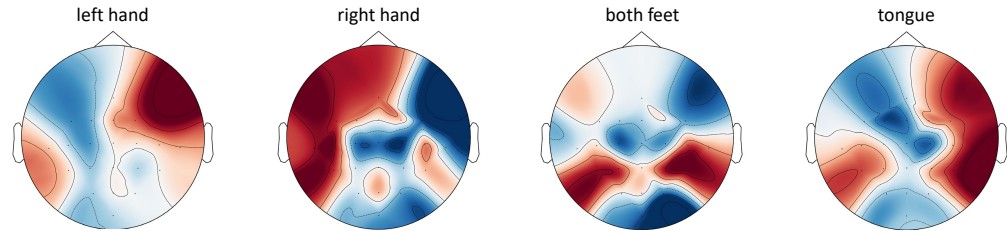

Figure 13: Topography visualization of BCIC-IV-2a.

For the **SEED-V** dataset, as shown in Figure 14, emotional tasks elicit significant activation across the frontal and occipital lobes, consistent with the finding in [32]. Notably, different emotional states produce distinct activation patterns, providing a critical neurophysiological basis for their identification.

For the **BCIC2020-3** dataset, as shown in Figure 15, speech imagery evokes significant activation in the frontal and temporal lobes [82], which are crucial for imagined speech processing [83]. In particular, strong activation is observed in left-hemisphere electrodes F7, FT7, FC5, and F5, which correspond primarily to Broca's area. This finding further supports the involvement of Broca's area in speech processing [83]. Overall, these visualizations confirm that different EEG decoding tasks

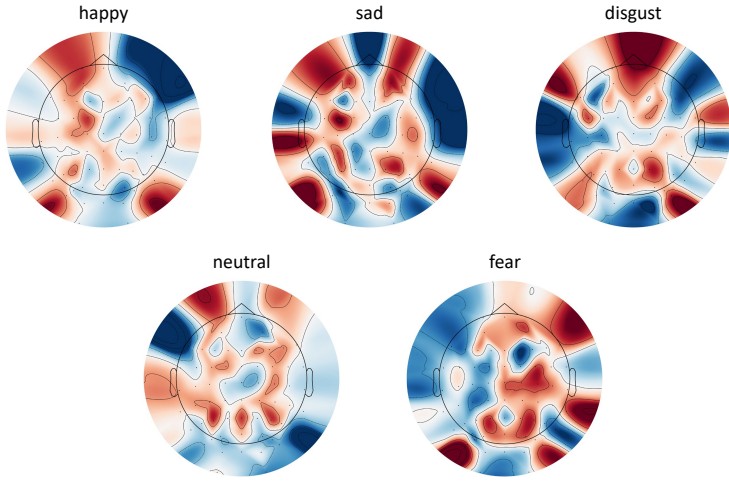

Figure 14: Topography visualization of SEED-V.

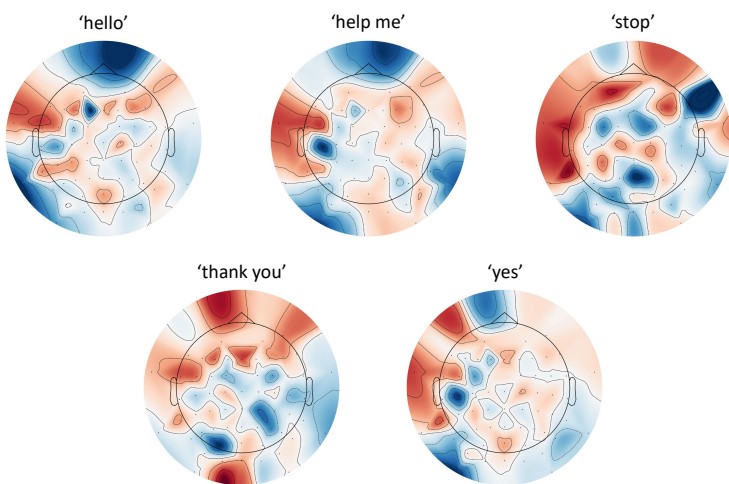

Figure 15: Topography visualization of BCIC2020-3.

elicit distinct activation regions and spatial scales, reinforcing the importance of modeling cross-scale neural dynamics. By explicitly capturing such multi-scale patterns, CSBrain effectively adapts to the heterogeneous spatial characteristics of brain activity across diverse cognitive domains.

## F.2    Representation Visualization

To qualitatively assess the discriminative capability of the representations learned by our model, we conduct t-SNE visualizations [120] on two representative datasets: BCIC-IV-2a (motor imagery classification) and BCIC2020-3 (imagined speech classification). For each dataset, we compare the feature distributions of raw EEG signals and those produced by CSBrain after encoding. As shown in Figure 16 and Figure 17, the features learned by CSBrain exhibit significantly improved clustering effects compared to the raw EEG input. In BCIC-IV-2a (Figure 16), different motor imagery classes form more compact and well-separated clusters, indicating that CSBrain effectively captures task-relevant discriminative patterns. Similarly, for BCIC2020-3 (Figure 17), imagined speech categories that are difficult to distinguish in the raw EEG space become more distinguishable after CSBrain encoding.

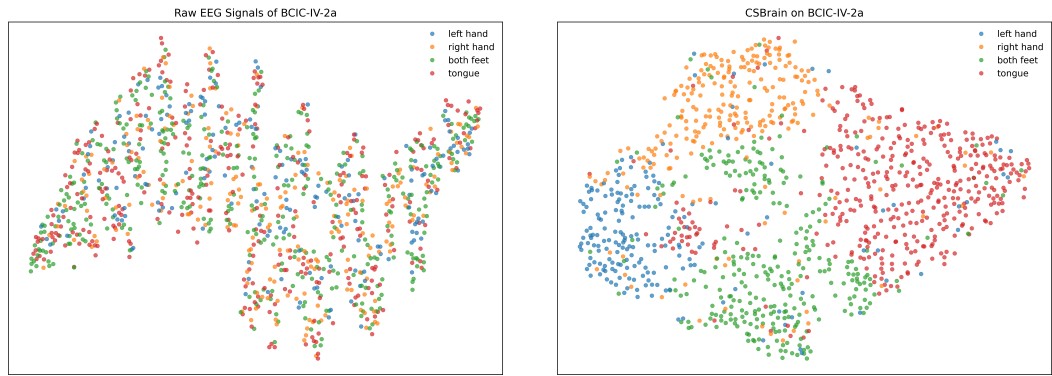

Figure 16: t-SNE visualization of feature representations on BCIC-IV-2a. Left: raw EEG signals; Right: CSBrain representations.

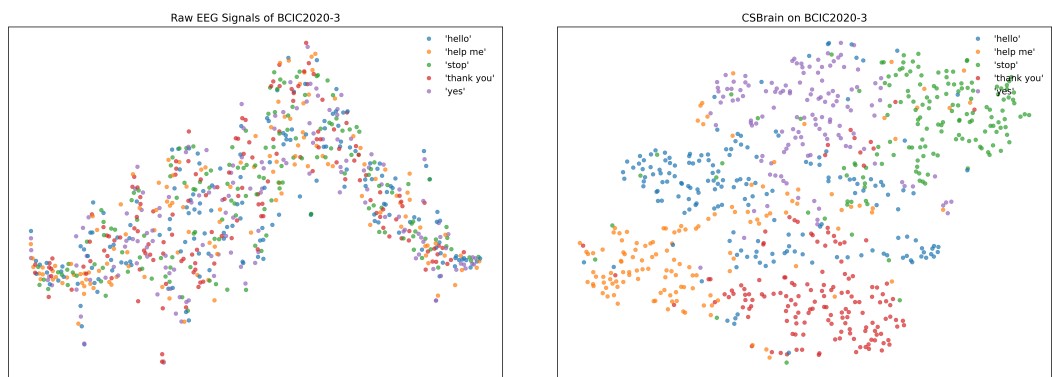

Figure 17: t-SNE visualization of feature representations on BCIC2020-3. Left: raw EEG signals; Right: CSBrain representations.

### F.3    Attention Visualization

The attention visualizations highlight the cross-scale modeling capability of CSBrain. As shown in Fig. 18, attention patterns across multiple layers reveal the model's ability to capture temporal dependencies at various scales. For example, in the lower layers, attention focuses more on local features, while in deeper layers, attention spans broader time windows, demonstrating CSBrain's proficiency in modeling both local and long-range dependencies. The hierarchical attention across multiple layers enhances its ability to capture complex spatiotemporal relationships in EEG data, a key advantage of the cross-scale approach.

### F.4    Channel Relationship Visualization

To further investigate the spatial dependencies captured by CSBrain, we visualize the inter-channel relationships learned on two representative datasets: BCIC-IV-2a and BCIC2020-3. Specifically, we first compute the average token embedding for each EEG channel from the final layer representations. We then calculate the cosine similarity between every pair of channel embeddings to quantify inter-channel relationships and visualize those with a similarity greater than 0.5. As shown in Figure 19 and Figure 20, the learned channel similarity patterns reveal two notable characteristics. First, channels within the same anatomical brain region tend to exhibit high similarity, indicating that CSBrain captures strong intra-region dependencies. Second, we observe meaningful long-range inter-region connections, such as between frontal and parietal regions or temporal and occipital regions, suggesting that our model also learns physiologically plausible global interactions across distant brain areas. These results validate the effectiveness of our cross-scale modeling approach in capturing both local and global spatial dynamics, which are essential for robust and generalizable EEG decoding.

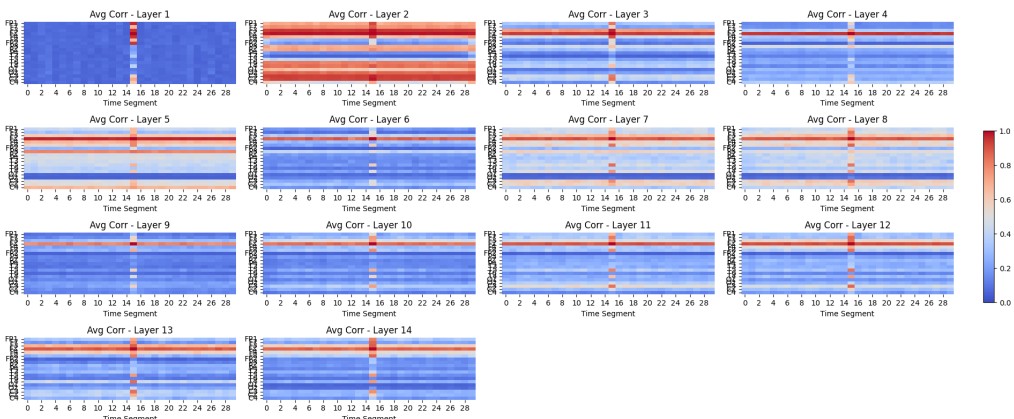

Figure 18: Attention Visualization.

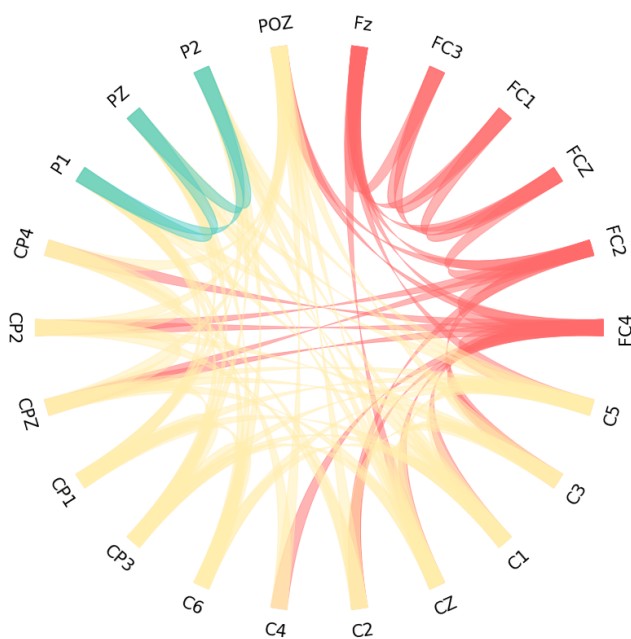

Figure 19: Channel similarity relationships of BCIC-IV-2a.

## G Discussion

### G.1 Implication

Our work emphasizes the critical role of cross-scale spatiotemporal modeling in the design of brain foundation models, a factor often overlooked in prior efforts. CSBrain represents a paradigm shift from scale-agnostic, densely connected modeling to a cross-scale, structure-aware architecture. By explicitly encoding multi-resolution dynamics through Cross-scale Spatiotemporal Tokenization (CST) and Structured Sparse Attention (SSA), our approach more effectively reflects the complex patterns of neural activity, thereby achieving superior generalization across EEG decoding tasks with inherently diverse spatiotemporal demands.

This modeling perspective provides a reusable architectural template for future EEG foundation models and has the potential to generalize to other neurophysiological modalities. The strong performance of CSBrain across 11 representative tasks and 16 public datasets suggests that unified

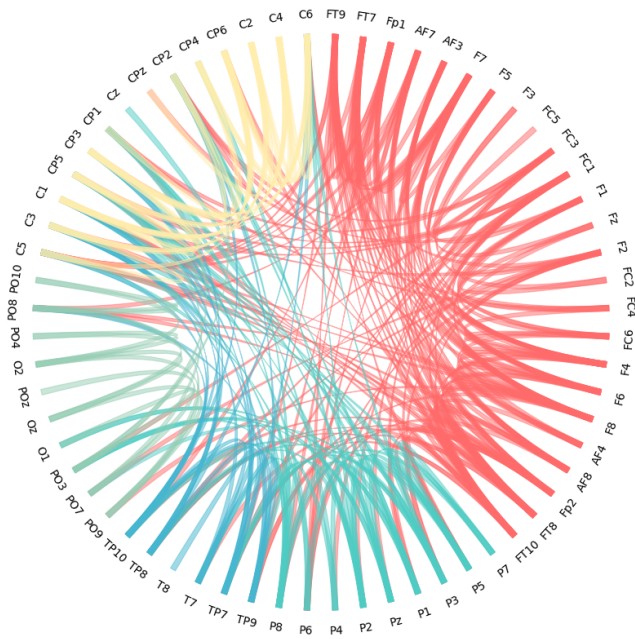

Figure 20: Channel similarity relationships of BCIC2020-3.

architectures when guided by neurophysiological priors, can scale effectively across heterogeneous brain decoding scenarios.

## G.2 Limitation

Despite its strong generalization ability, CSBrain still faces several limitations. Firstly, our current exploration of model and data scaling is constrained by computational resources. Following the general trend observed in neural scaling laws [119], model performance tends to improve predictably with increased model capacity and training data size. While our results suggest similar patterns in EEG modeling, we are unable to empirically assess scaling behavior at the scale of vision-language or language-only models (e.g., CLIP or GPT). Moreover, an open question remains: how much EEG data is sufficient to pretrain high-quality foundation models? Due to the sparse, noisy, and heterogeneous nature of EEG datasets, further work is needed to address large-scale dataset unification, normalization, and cross-institutional collaboration. We believe such efforts will be essential to unlocking the full potential of EEG-based foundation modeling and call for broader community involvement to collectively advance this direction.

## G.3 Future Work

A natural extension of this work is to investigate the scaling behavior of EEG foundation models further. This includes expanding pretraining datasets, increasing model capacity, and generalization across downstream tasks. Understanding the efficiency and limits of scaling in EEG foundation models will be key to maximizing their representational capacity and real-world applicability. In response to insightful comments from the reviewers, we recognize the importance of exploring EEG foundation models across different model architectures. Specifically, we plan to investigate large-scale pretraining using architectures beyond CSBrain, such as GNN and Mamba models. GNNs excel at capturing local dependencies due to their graph-based structure. On the other hand, Mamba models, which leverage multi-scale learning and attention mechanisms, have demonstrated strong potential in handling multi-dimensional data efficiently. A comparative analysis of these models with CSBrain will help illuminate the strengths and limitations of each, contributing to more robust EEG foundation modeling. Additionally, we believe that incorporating multiple optimization objectives during pretraining could capture more complex, nuanced patterns in EEG signals. Combining different

objectives could enhance model robustness and its ability to generalize across diverse scenarios. Additionally, the cross-scale modeling paradigm in CSBrain opens the door to unified cross-modal brain foundation models. By integrating signals beyond EEG, such as MEG or fMRI, which offer complementary properties in spatiotemporal scales, we can construct richer and more generalizable neural representations and facilitate real-world applications in brain-computer interfaces, clinical diagnostics, and cognitive augmentation.

