# OpenReview forum: "CSBrain: A Cross-scale Spatiotemporal Brain Foundation Model for EEG Decoding"
_NeurIPS.cc/2025/Conference — NeurIPS 2025 spotlight_

### Official Review · Reviewer_6YFP · 2025-06-26

**Clarity:** 3
**Significance:** 4
**Originality:** 3
**Rating:** 5
**Confidence:** 4

**Summary:**

The paper introduces CSBrain: a foundation model used in the electroencephalography signals (EEG) domain of functional neuroimaging. The authors combine cross-scale spatiotemporal tokenization (CST) with structured sparse attention (SSA) to capture neural patterns on multiple scales and with sparsity in a hierarchical framework. Since EEG tasks exhibit signals at varying scales, this stacked cross-scale spatiotemporal structure is used to capture this diversity which is important for building a foundation model that can work in many settings and environments.

The introduced CST module starts by applying convolutions with varying kernel sizes across a temporal neighborhood which are then concatenated across scales to yield a temporal token. The temporal tokens are then split into localized spatial neighborhoods based on spatial adjacency in the 10-20 system. Spatial convolutions with varying kernels sizes are then applied to these neighborhoods and concatenated as before to yield the spatiotemporal tokens as output.

The introduced SSA module takes the spatiotemporal tokens and applies attention in two rounds. First, inter-window attention is applied between tokens which share the same relative index in the chosen window neighborhoods of the temporal convolutions. Second, inter-region attention is performed where each region is summarized through a region descriptor using a projection of the average token together with a randomly sampled token, then attention is applied across all of the regions' descriptor tokens.

Self-supervised learning is performed using masked reconstruction loss where whole regions of the input are erased for the model to reconstruct. The model is pre-trained on the TUEG dataset.

The authors present a very comprehensive evaluation of the EEG foundation model evaluating on 16 EEG datasets with 11 different BCI tasks against 10 baseline models. Baseline models are reimplemented and retrained where results are not available on the specific dataset. The proposed model achieves on-par or better fine-tuning performance on all downstream tasks. An ablation study is applied to the number of kernels, showing that the repeated and stacked model with three kernel sizes is the best choice of architecture. An ablation study on the chosen attention method is performed against dense attention and criss-cross attention which showed that the proposed SSA performs well across datasets.

The authors use Grad-CAM to explore the changes of model prediction by variation in input EEG channels, yielding findings consistent with prior works.

**Questions:**

### Questions regarding the number of kernels comparison
For better comparison, the number of parameters and/or compute should be kept constant between ablations in the number of kernels comparison.
- Can the authors provide insight into the number of parameters and FLOPS used in this part of the study?
- Is it just an increase in compute or parameters that gives the perceived performance when increasing the number of kernels used? How much of the performance increase is because of a stronger model (more filters, parameters etc.)?
- Have the authors experimented with using multiple kernels but keeping the scale the same? Using three kernels of the form [3, 3, 3], for example? Here we are adding more parameters (more kernels) but forgoing the multiple scales.
- What about [3, 3, 3] + Dense Attention? This would seem like the naïve baseline.

### Questions for the cross-scale spatiotemporal tokenization procedure
The inter-region self attention has been proposed before (in Wang et al.), and the proposed alternating stacks of CST and SSA follows the conventional methodology of hierarchical models.
Usually, in hierarchical transformer models, instead of a convolution, attention is applied intra-regionally and then inter-regionally, e.g. in SWIN transformers.
- Have the authors considered a model where an intra- region/window attention module replaces the convolutions?

It is a bit unclear how inter-region attention provides inputs to the next stack in the model. The inter-window attention mechanism makes sense since the attention is just limited using a periodic mask. However, in the inter-region attention, after pooling and randomly sampling tokens from the region, I assume the signal is reduced to having only $R_r$ tokens in total over which the attention is performed.
- How are these tokens then added back to $x_{i,j}$ in the next part?
- The authors state that the multi-scale kernels are only applied to the relevant neighborhood. Do you then extract each of these and use zero-padding to perform the convolution (e.g. in regions where there are less than the kernel scale electrodes)?

The attention mechanism of the inter-window attention seems a bit arbitrary.
- What is the justification for relative indices being grouped together? As I see it, these are not necessarily similar unless there is a periodicity of the size of the window in the signal.
- Why not use the same token descriptor method as in inter-region attention? Does it perform worse?

### Questions regarding the spatial assignment into brain regions
It would be helpful if the authors provided the brain region assignment in the 10-20 system, and justify why the chosen assignment is chosen as it is.
- What is the chosen assignment? Is it based on prior works?
- Why have the authors chosen this assignment?
- Have the authors experimented with other assignments?

### References
Szegedy, Christian, et al. "Going deeper with convolutions." Proceedings of the IEEE conference on computer vision and pattern recognition. 2015.

Wang, Zhe, et al. "Transformers for EEG-based emotion recognition: A hierarchical spatial information learning model." IEEE Sensors Journal 22.5 (2022).

**Ethical Concerns:**

["NO or VERY MINOR ethics concerns only"]

**Final Justification:**

The paper presents a technically sound and well-motivated EEG foundation model that leverages cross-scale spatiotemporal tokenization and structured sparse attention to achieve strong performance across diverse EEG decoding tasks. The rebuttal and discussion convincingly addressed all major concerns. In particular, the authors provided TUEx-excluded macro-average results and additional kernel configuration experiments that clearly supported their claims that the performance gains come from principled multi-scale modeling rather than increased model capacity. Based on the strength of the technical contribution, thorough evaluation, and the clear evidence provided in response to my concerns, I recommend acceptance.

**Limitations:**

yes

**Paper Formatting Concerns:**

There are no major formatting issues.

**Quality:**

4

**Strengths And Weaknesses:**

### Strengths
- The paper is well written and technically sound with only minor spelling mistakes.
- The proposed method is thoroughly justified and the claims are comprehensively evaluated.
- The present paper tackles a timely and significant problem in EEG research paving the way for a generalized framework for EEG representation.
- Importantly, using natural inductive biases yields a more structured, cost‐effective model with no loss in performance, a prospect that is considerably valuable for future directions in EEG modeling research.
### Weaknesses
- All of the TUEx datasets (TUEV, TUEB, TUSL) are subsets of the TUEG dataset used for pretraining. There is thus data contamination in the evaluation pipeline since some of the evaluation datasets are also part of the pretraining set. The results on these are still valuable, but I think the authors should note and discuss this, and possibly keep them separate in the results Table 2. (Also leaving it out of the macro average for the same reason).
- It is unclear which regions are grouped together. The authors state that the grouping is based on "anatomically defined brain regions" and that each region comprises "spatially adjacent electrodes", but the assignment of electrodes into regions is not specified.
- Although limited to spatial neighborhoods in the present paper, the idea of using multiple kernel sizes has already been explored in prior work (Szegedy et al.), where they also indeed used kernels of size [1, 3, 5] for each layer in the hierarchical model. This prior work would serve as a good reference for the importance of multi-scale processing.
- It is unclear from the main manuscript precisely how the present work differs from previous state-of-the-art foundation models for EEG.

---

> ### Author Rebuttal · Authors · 2025-07-26
>
> Dear Reviewer 6YFP,
>
> Thank you for your encouraging feedback and constructive suggestions! We are greatly encouraged by your recognition that **the paper tackles a timely and significant problem in EEG, paving the way for a generalized framework for EEG representation, and is considerably valuable for future directions in EEG research**. We are also pleased that you found **our proposed method thoroughly justified** and **technically sound**. Due to time and page limitations, we have made every effort to address your questions and suggestions through the clarifications provided. We hope these responses resolve most of your concerns and will be considered in your final assessment.
>
> > W1. About the TUEx Datasets
>
> We clarify that our pretraining task, masked signal reconstruction, is purely unsupervised, involving no downstream task labels. Thus, there is no label leakage. The aim is to learn general spatiotemporal EEG representations. In prior foundation model work, such as [NeurIPS2024] BIOT and [ICLR2025] Cbramod, TUEx datasets were also used for pretraining, and they reported fine-tuned results on TUEV and TUAB datasets. To further enhance transparency, we will explicitly mark the TUEx results and compute an additional macro-average excluding TUEx datasets in Table 2. On this new macro-average, CSBrain achieves 0.6829, outperforming CBraMod (0.6537) and LaBraM (0.6430), consistently surpassing competitors by 3+%.
>
> > W2&Q3. Spatial Assignment into Brain Regions
>
> Our fundamental goal is to develop a generalizable EEG foundation model applicable to diverse EEG tasks. To ensure broader applicability, the spatial region assignment adheres to the standard 10-20 system compatible with all used datasets. In this system, electrode labels serve as proxies for anatomical brain regions, and electrodes are mapped to five primary regions: Frontal (F), Central (C), Parietal (P), Temporal (T), and Occipital (O). This information was briefly introduced in Sec D.1 (L 727-728). To further enhance readability, we will expand this explanation into a dedicated subsection (e.g., C.5) in the supp. mat., detailing electrode assignments for each dataset. We will also release the code and model to support community use of our work.
>
> > W3. Classic Work on Multi-Scale Modeling in CV
>
> We appreciate your insightful suggestion regarding Szegedy et al. as a key early work that introduced multi-scale convolutions in the CV domain. We will gladly include this classic work as a reference in the main paper's Introduction (L76-80) to further elaborate on the significance of multi-scale modeling.
>
> > W4. Differences to Prior Works
>
> As shown in Fig. 1 of the main paper, CSBrain fundamentally differs from prior EEG foundation models by explicitly addressing and leveraging the intrinsic cross-scale spatiotemporal structure of neural activity, a property often overlooked by previous scale-agnostic dense modeling paradigms.
> - **Difference in Tokenization:** Prior models typically rely on one-time, fixed-scale tokenization strategy, yielding token representations that struggle with diverse neural patterns across tasks. In contrast, our proposed CST aggregates multi-scale features within localized temporal windows and anatomical brain regions via various convolution kernels. This produces compact, scale-aware neural tokens, enabling adaptable alignment with diverse spatiotemporal requirements of EEG decoding tasks.
> - **Difference in Spatio-Temporal Modeling:** Previous works commonly adopt structure-agnostic attention (e.g., dense attention or Criss-Cross Attention) for spatio-temporal modeling, which can introduce redundancies and computational overhead. In contrast, our proposed SSA models cross-window and cross-region dependencies. This structure-aware design enhances scale diversity, reduces spurious dependencies, and lowers computational overhead, yielding more discriminative and efficient representations.
> - **Hierarchical Architecture:** Unlike prior works, CSBrain establishes a hierarchical architecture by alternately stacking CST and SSA, enabling synergistic enhancement and robust EEG representations that capture complex neural patterns. This leads to strong generalization across diverse EEG decoding tasks, as evidenced by significant performance gains on 16 datasets.
>
>
> We greatly appreciate your suggestion and will revise the Introduction by adding a separate paragraph after Line 88 to more clearly emphasize the differences between our approach and prior state-of-the-art EEG foundation models.
>
> > Q1. Comparison of Number of Kernels
> - **Computational Cost Comparison**: For kernel configurations K=1, K=2, and K=3, parameters are 4.88M, 4.92M, 4.94M respectively, and FLOPs are 246.22M, 265.42M, 275.02M. The parameter count increases only slightly, and FLOPs show a modest increase from K=1 to K=3. This is because we keep the total feature embedding dimension (=200) constant via an exponentially decaying scheme for individual kernel dimensions, balancing representational capacity and efficiency (main paper L168-172). Despite these constrained increases, our model demonstrates significantly improved performance at K=3 across multiple tasks (main paper Fig. 4, L248-253). This outcome strongly supports the effectiveness of cross-scale spatiotemporal modeling, proving performance gains stem from respecting EEG's intrinsic cross-scale properties, not solely from capacity increase.
>
> - **Ablation with Additional Kernel Variants**: To address your concerns further, we added experiments with the kernel configurations [1,1,1], [3,3,3], and [5,5,5], as well as a baseline with [3,3,3]+Dense Attention. Due to time and computational constraints, these experiments were performed using a subset of the TUEG dataset (2000 hours, excluding TUEV) and evaluated on two downstream tasks:
> |Kernel Config|TUEV B-Acc|TUEV Kappa|TUEV F1-W|BCIC2020-3 B-Acc|BCIC2020-3 Kappa|BCIC2020-3 F1-W|
> |-|-|-|-|-|-|-|
> |[1,3,5]|0.6414±0.0187|0.6236±0.0198|0.7971±0.0183|0.5443±0.0232|0.4254±0.0301|0.5527±0.0297|
> |[1,1,1]|0.6122±0.0193|0.5596±0.0212|0.7713±0.0205|0.5041±0.0208|0.3821±0.0223|0.5019±0.0202|
> |[3,3,3]|0.6338±0.0184|0.6137±0.0179|0.7870±0.0188|0.5298±0.0247|0.4354±0.0298|0.5399±0.0280|
> |[5,5,5]|0.6188±0.0190|0.5765±0.0209|0.7732±0.0195|0.4998±0.0237|0.3809±0.0242|0.4931±0.0279|
> |[3,3,3]+Dense Attention|0.5901±0.0178|0.5380±0.0199|0.7531±0.0212|0.5076±0.0221|0.3844±0.0269|0.5023±0.0222|
>
> These results further validate the effectiveness of CST for cross-scale modeling, where the [1,3,5] configuration achieves the best performance, while [3,3,3] provides the second-best results. This demonstrates that explicitly modeling diverse scales is more effective than simply adding more filters of the same scale. This comprehensive analysis reinforces that CSBrain's performance gains are indeed attributable to its principled cross-scale spatiotemporal modeling. *We will update Supp. Sec. E to reflect these insights.*
>
> > Q2. CST Procedure
>
> **Q2-1. Replacement of intra-region/window attention**:
> - We considered replacing CST's convolutions with intra-region/window attention modules during the early stages of our project. However, preliminary experiments showed negligible performance improvements (~1% on key metrics for Seed-V, TUEV) while substantially increasing parameters due to doubled attention computations.
> - We will gladly cite and discuss the classic work by Wang et al. in the Introduction and Related work.
> - We also explored Swin-style models early in our project. However, we found that their shifted window partitioning for efficient computation (as shown in Fig. 4 of their paper) would likely disrupt the temporal and spatial order inherent in EEG signals. While highly effective for images, transference of such mechanisms to EEG data poses fundamental challenges, as preserving sequential and local spatial dependencies is paramount. *We will update the manuscript to reflect these considerations more explicitly.*
>
> **Q2-2. Inter-region Attention**:
> - To clarify, as stated in L193, tokens are sequentially sampled from each region, not randomly. In Supp.E.3, we compare four strategies for constructing Regional Descriptors, including random sampling. Results show that sequential sampling with the average feature gives the best performance, ensuring each token represents its region’s spatial context.
> - In inter-region attention, token count is not reduced. As in Equation 5, each position $x_{i,j}$ is assigned to a spatial group for attention, ensuring that no information is lost during sampling. This design preserves the spatial and temporal relationships between regions and guarantees that tokens flow consistently throughout the model. *We will rewrite L191-198 for further clarity.*
>
> **Q2-3. Zero-padding**: Yes. For regions with fewer electrodes than the kernel size, we apply zero-padding during the convolution operation.
>
> **Q2-4. Inter-window Attention**:
> - In inter-window attention, each token is enriched with information from other tokens in the same window via CST, ensuring tokens share a common spatiotemporal context. This allows relative indices to be grouped together, as windowed tokens have a consistent temporal structure.
> - In contrast, for inter-region attention, varying electrode counts prevent tokens from fully representing their regions. Therefore, we use token descriptors, adding the average feature of the region to each token.
> - Our preliminary experiments showed that adding a token descriptor to windowed tokens did not significantly impact performance (~<1% difference).
>
> *Due to rebuttal space constraints, some brief experimental results will be included in revised versions of the paper. We will expand L191-198 in the main paper to clarify the specific implementation details and design motivations for SSA. Furthermore, we will open-source the relevant code to facilitate community use of our work.*

---

> > ### Comment · Reviewer_6YFP · 2025-08-05
> >
> > Thank you for the detailed response and for providing the additional macro-average results excluding the TUEx datasets. While I understand that the pretraining is unsupervised and does not involve label leakage, the fact remains that the model has been exposed to the same data distribution during pretraining, which will in any case influence downstream performance on such tasks. This kind of data overlap can inflate performance on those subsets compared to truly unseen datasets. The new macro-average excluding TUEx datasets, where CSBrain still significantly outperforms competitors, is a valuable insight that strengthens the paper's claims markedly. This transparency and thorough addressing of the concern is appreciated.
> >
> > The provided parameter and FLOPs analysis, along with the additional ablation experiments on kernel configurations offer strong empirical support for the paper's claims. The analysis convincingly validates that the observed performance gains stem from principled cross-scale spatiotemporal modeling rather than mere increases in model capacity.
> >
> > Equation (4) seems to indicate that the output is reduced to only $R$ tokens, which, together with the manuscript statement on L196-L197 about assembling descriptors into a spatial group for self-attention, caused the confusion regarding inter-region attention. The authors should clarify what "sequentially sampled tokens" entails and explain how multiple tokens per region are preserved and mapped back to spatial positions in the tokenization module. Are these regional descriptors used as conditioning (i.e. by addition to the full C x n x d signal)? (Which seems to be the case by the explanation provided under Q2-4. Inter-window Attention in the rebuttal).

---

> > > ### Author Response · Authors · 2025-08-06
> > >
> > > Thank you for your detailed and insightful comments, which have helped us to further improve the quality of our paper. We are glad to hear that you appreciate how our response has increased transparency and thoroughly addressed your concerns. We agree with your perspective on the TUEx pre-training data, as this has been overlooked in previous work on EEG foundation models. We will incorporate your point into the discussion in subsequent versions of the paper. Here, we'll further explain your questions about Inter-region Attention.
> > >
> > > **Spatial Grouping and Sequential Sampling**
> > >
> > > Given an EEG feature with shape \([C, n, d]\), we first perform Spatial Grouping to construct multiple spatial groups *across brain regions*. For the \(i\)-th group, the \(i\)-th electrode is selected from each region's electrode list (predefined in 10-20 system). If a region has fewer electrodes than \(i\),  the selection is cyclic (i.e., using \(i \% N_r\), where \(N_r\) is the number of electrodes in the region). This process ensures that every electrode is assigned to at least one spatial group for Inter-region Attention computation.
> > >
> > > **How is the output fed into the tokenization module?**
> > >
> > > In practice, the Spatial Grouping operation does not physically rearrange token positions. Instead, we construct a C $\times$ C attention mask based on the grouping results, which suppresses interactions between electrodes from different groups while enabling intra-group communication. This design eliminates the need for explicit token reordering, preserving computational efficiency and maintaining the original output dimensionality for feeding into tokenization module.
> > >
> > > **Regional Descriptor Construction**
> > >
> > > The regional descriptor $\tilde{x}_r$ is not a conditioning feature but an augmented EEG token embedding with region awareness. $\tilde{x}_r$ is the sum of two components: (1) the token $\tilde{x}_r^{\text{rep}} \in \mathbb{R}^{1 \times n \times d}$ from sequential sampling, and (2) the mean feature $\phi(\mathcal{P}(R_r)) \in \mathbb{R}^{1 \times n \times d}$ of the token's brain region. This ensures that each token is enriched by its region’s context while preserving original information before inter-region attention.
> > >
> > > **Regarding Equation (4)**
> > >
> > > Equation (4) describes the process "For each group" (L193), where each spatial group will have $R$ tokens. We construct multiple spatial groups, with each electrode assigned to at least one spatial group for attention computation. In implementation, different groups are organized through an attention mask (C $\times$ C mask matrix) to perform parallel attention operations. The number of tokens in the output of inter-region attention remains the same, as all tokens are grouped and not discarded.
> > >
> > > *We will rewrite L191-198 for further clarity to help future readers more easily understand our work.*

---

> > > > ### Comment · Reviewer_6YFP · 2025-08-06
> > > >
> > > > Thank you for your detailed clarification regarding the Spatial Grouping and sequential sampling procedure. Your explanation helps clarify how each electrode is assigned to at least one group via cyclic indexing and how this grouping is encoded through an attention mask without physically reordering tokens.
> > > >
> > > > However, I believe the term "sequential sampling" may lead to misunderstandings, as it is commonly associated with concepts like autoregressive sampling. In the present context, the term refers to a process of creating a covering partition over the electrodes, i.e., assigning every electrode to at least one group, possibly with cyclic overlap.
> > > >
> > > > To improve conceptual clarity and avoid ambiguity, I suggest using "covering partition" to describe this procedure instead or similar nomenclature.

---

> ### Author Response · Authors · 2025-08-07
>
> Dear Reviewer 6YFP,
>
> Thank you for your thoughtful feedback. We agree that the term "covering partition" more accurately and conceptually describes our procedure. We will update the manuscript, particularly the expression in Line 193, to enhance conceptual clarity and facilitate future readers' understanding.
>
> Furthermore, we are very grateful for your active and insightful feedback throughout the entire review process, and we believe it has significantly improved the quality of our work. We hope that, through the collective efforts of the entire community, we can advance EEG modeling toward a more generalizable direction and facilitate its application in the real world (including clinical diagnostics, cognitive augmentation, and brain-computer interfaces) to improve human life.

---

### Official Review · Reviewer_yshg · 2025-06-28

**Clarity:** 3
**Significance:** 2
**Originality:** 3
**Rating:** 4
**Confidence:** 4

**Summary:**

The paper argues that cross-scale modeling is lacking from LLM-inspired brain models. Introduces cross-scale spatiotemporal tokenization and structured sparse attention. Problems with current models include scale-agnostic tokenization and dense attention.

**Questions:**

In section 2.1 you mention FFT for spectral information, why not use learned 1D convolutions, aren’t those enough?

SSA: How is this different from combining temporal and spatial attention layers?

Kernel stacking comparison: Does this mean than even more scales would increase performance?

More comparisons with other foundational models on the pre-training task.

**Ethical Concerns:**

["NO or VERY MINOR ethics concerns only"]

**Final Justification:**

After carefully reviewing the authors' responses I decided to raise my final score by 1. The authors thoroughly explained how their propose model works, and these clarifications will be integrated into the paper.

**Limitations:**

yes

**Paper Formatting Concerns:**

No issues.

**Quality:**

3

**Strengths And Weaknesses:**

### Strength
- large pre-training dataset
- very comprehensive downstream evaluation

### Weaknesses
I think the term tokenization is misleading, as these are simply convolutional layers applied between attention layers.
- Temporal tokenization is really just a stack of convolutions
- Spatial tokenization is just standard convolution, but nearby regions are correlated anyway so I don't see what's the point.
- The masked learning uses MSE which has fundamental issues, such as focusing on low-frequency features.
- Is the pre-training amount used for the other foundation models similar? Otherwise comparison might not be fair
- Lack of comparisons with other foundational models in terms of evaluating why yours is better. Not just downstream performance.
- No direct evaluation of the pre-trained model itself

---

> ### Author Rebuttal · Authors · 2025-07-28
>
> Dear Reviewer yshg,
>
> Thank you for your encouraging feedback and constructive suggestions! We are greatly encouraged by your recognition of **our very comprehensive downstream evaluation** and the use of a **large pre-training dataset**. Due to time and page limitations, we have made every effort to address your questions and suggestions through the clarifications provided. We hope these responses resolve most of your concerns and will be considered in your final assessment.
>
> > W1-3. Clarification of Tokenization
>
> - In the filed of EEG foundation model, which commonly employ Transformer architectures, tokenization is not just a convolutional operation but the essential process of converting continuous spatiotemporal signals into discrete, Transformer‑friendly units (i.e., EEG tokens). This terminology is well established in prior works (e.g., [ICLR 2024] LaBraM; [NeurIPS 2024] BIOT), and since each EEG token serves as the minimal modeling unit, effective tokenization is a fundamental prerequisite for any EEG foundation model.
> - In CSBrain, our Cross-Scale Spatiotemporal Tokenization (CST) module aggregates multi-scale features across both temporal and spatial dimensions into compact, structured token representations. This design is guided by the intrinsic property of neural activity — the cross-scale spatiotemporal structure — and aligns with the underlying structure of EEG signals. The key distinction of our design is that tokenization in CSBrain is not simply a stack of convolutions, but a hierarchical process that aggregates information across scales and spatial regions. This allows the model to capture richer, more granular representations of EEG signals. The resulting tokens, which encapsulate both local and global spatiotemporal information, are passed to the attention layers where the model learns to capture dependencies across different time windows and brain regions.
>
> *We will rewrite L131-140 of the main paper to more explicitly clarify the role of tokenization and its contribution to the CSBrain architecture.*
>
> > W4. About MSE loss
>
> For masked pretraining, MSE is a widely used and intuitive loss function in signal and image reconstruction tasks, as seen in prior works like MAE and Cbramod.. While MSE does prioritize low-frequency features due to its focus on minimizing larger amplitude errors, this characteristic aligns well with EEG signal properties, where energy is predominantly concentrated in lower frequency bands.
>
> Importantly, in CSBrain, input EEG segments undergo preliminary encoding to extract both temporal and spectral (FFT) features (Sec.2.1 L124-129). During pretraining, the model reconstructs the raw EEG based on these spectral-rich features, compelling it to utilize high-frequency FFT components for accurate reconstruction. Additionally, CSBrain's CST module extracts features across multiple temporal scales, capturing information from a range of EEG frequencies. Therefore, our masked reconstruction task could enforce the model to learn beyond low-frequency features, enabling a more comprehensive understanding of EEG signals.
>
> Our extensive experiments across 11 EEG tasks and 16 datasets demonstrate CSBrain’s strong performance, including tasks sensitive to high-frequency or transient features, such as Motor Imagery Classification and Event Type Classification. *To enhance clarity, we will integrate this discussion into Sec. 2.4 (L205-215). Furthermore, inspired by your insightful comment, we agree that augmenting pretraining with multiple objectives could capture more complexity, and we will incorporate this into our “Future Work” section (Sec. G.3) to encourage broader community efforts.*
>
> > W5-7&Q4. About Pre-trained Model Evaluation
>
> **1. Pretraining Data Amount.**
>
> - **Fairness of evaluation protocols:** In the EEG community, *comparing pretrained models in downstream task performance under varying pretraining data is a standard practice adopted by prior works such as [ICLR2024]LaBraM, and [ICLR2025]Cbramod*. Similarly, in broader AI, foundation models like CLIP, BLIP, and GPT are rarely pretrained on identical datasets, as uniform pretraining with large-scale data is impractical. Thus, our evaluation protocol is NOT unfair.
>
> - **Comparison under the same pretrained data amount:** CSBrain uses more pretraining data than LaBraM but less than BIOT, and uses the **SAME pretraining data** as the previous SOTA model Cbramod (using the TUEG dataset). Across 11 representative tasks and 16 datasets, CSBrain consistently outperforms Cbramod, achieving a macro-average score of 0.7095, surpassing Cbramod’s score of 0.6760. This demonstrates that even with the same amount of pretraining data, our architecture holds a clear advantage. *We will add a comparison of the pretraining data amount in L263 of the main paper for greater clarity and transparency.*
>
> Your insightful comments inspire us to consider future work in building sufficiently large-scale, high-quality open-source EEG databases for unified pretraining. *We will incorporate this direction into Sec. G.3 of our future work to encourage community efforts toward these advancements.*
>
> **2. Regarding Why We Focus on Downstream Performance For Comparisons.**
>
> Evaluation on downstream performance aligns with the core goal of EEG foundation models, which is to achieve performance gains and generalization on downstream tasks through large-scale unsupervised pretraining and a unified model architecture. Since pretraining data and objectives vary across models, prior foundational models such as LaBraM, BIOT, and Cbramod have **followed similar evaluation pipelines**, focusing on downstream performance after pretraining, without directly evaluating the pretraining task itself. In line with these models, we prioritize the evaluation on downstream tasks.
>
> **3. More Validation on Pretraining.**
>
> In Supp. Mat., we have provided further validation of the quality of our pre-trained model's representations through several approaches:
> - **Pre-training Effectiveness:** In Supp. Mat. E.1, we show significant performance improvements on downstream tasks by comparing models with and without pre-training, validating the effectiveness of our pre-training strategy.
> - **Pre-training Scale Analysis:** Pre-training Scale Analysis: In Supp. Mat. E.2, our experiments demonstrate that increasing pre-training data leads to better performance, proving our compliance with the Neural Scaling Laws for foundation models and good scalability.
> - **Pre-training Process Transparency:** In Supp. Mat. B.2, we provide the Training Loss Curve and hyperparameters, demonstrating effective convergence under our pretraining objective.
>
> > Q1. FFT for Spectral Information
>
> The use of Fourier spectrum information has been widely adopted in previous EEG foundational models ([ICLR2024]LaBraM, [NIPS2024]BIOT, and [ICLR2025]CBraMod). These models consistently show that integrating frequency-domain information with temporal features significantly enhances decoding performance. Following their practice, our model integrates FFT‑based spectral extraction with 1D convolutions, harnessing their complementary strengths. and the physiological relevance of EEG frequency‑domain information.  While 1D convolutions excel at learning local temporal dependencies, FFT provides an explicit representation of the signal's frequency components, enabling the model to capture essential frequency-based patterns that 1D convolutions might miss. By combining both approaches, our model effectively leverages both temporal and spectral information, providing a comprehensive representation of the EEG signal.
>
> > Q2. The difference of SSA and spatiotemporal Attention
>
> We compare SSA with Temporal and Spatial Attention (TSA) from both architectural and experimental perspectives.
> - **Architectural Comparison:** The key difference between SSA and TSA lies in how they model long-range spatiotemporal dependencies. TSA typically focuses on independently modeling dependencies along the time and space axes, without distinguishing between intra- and inter-region, or short-term and long-term cross-scale structures. In contrast, SSA respects the intrinsic property of neural activity, accounting for cross-scale interactions across brain regions and time windows. It works in close conjunction with CST to integrate these spatiotemporal dependencies in a more structured and efficient manner. By leveraging the inherent cross-scale structure of EEG signals, our model effectively captures both local and global patterns.
> - **Experimental Comparison:** We compare SSA with Cbramod [ICLR2025] that explores TSA by a Criss-Cross Transformer for modeling dependencies along the time and space axes. As shown in Section 3.3 (L258-277) and Table 2 of our paper, our model achieves a 0.7095 macro-average across 11 representative EEG tasks and 16 datasets, significantly outperforming Cbramod (0.6760 macro-average). Additionally, in L293-305 and Figure 5, our proposed SSA demonstrates superior efficiency compared to TSA (Criss-Cross Attention).
>
> *We will open-source the relevant code and models to facilitate community use and further advancements in this area.*
>
> > Q3. Kernel Stacking Comparison
>
> As discussed in L278-292, our "Tokenization Comparison" ablation study clearly demonstrates that full cross-scale tokenization (K=3) significantly outperforms both single-scale (K=1) and dual-scale (K=2) variants. Specifically, our SSA (K=3, stacked) consistently achieves the best performance across four representative tasks. This suggests that explicitly modeling multiple temporal and spatial scales contributes to learning richer EEG representations, further validating the effectiveness of our core insight—cross-scale spatiotemporal modeling.

---

> ### Comment · Reviewer_yshg · 2025-08-01
>
> Thank you for the detailed responses. I now understand better the details of the model. I understand that tokenization is a hierarchical process, but still it doesn't produce discrete vectors, which I would call tokens.
>
> For a lot of your arguments you are citing Labram and CBramod, two papers which came out in the last year. I would appreciate a more grounded view of EEG modeling literature. Just because one recent paper does something does not mean that it is well established.

---

> > ### Author Response · Authors · 2025-08-08
> >
> > Dear Reviewer yshg,
> >
> > With the author-reviewer discussion period nearing its end, we would be very grateful if you could let us know if you have any further questions or concerns regarding our detailed rebuttal.
> >
> > We remain available to provide any additional clarifications to assist you in your final assessment.
> >
> > Thank you for your time and consideration.

---

> > > ### Comment · Reviewer_yshg · 2025-08-08
> > >
> > > Thank you, I have raised my score to reflect this.

---

> > > > ### Author Response · Authors · 2025-08-09
> > > >
> > > > Dear Reviewer yshg,
> > > >
> > > > Thank you for your positive feedback and for raising your score. We greatly appreciate the thoughtful and constructive comments you provided throughout the review process. We will incorporate your insights into the final version of the paper. We hope these revisions will inspire further research and contribute to advancing more generalizable EEG modeling, ultimately facilitating real-world applications.
> > > >
> > > > Thank you again for your time and consideration.

---

> ### Author Response · Authors · 2025-08-06
>
> Dear Reviewer yshg,
>
> Thank you for your further comments. We are glad our response has clarified the model details.
>
> In EEG modeling, tokenization refers to converting continuous spatiotemporal EEG signals into discrete token sequences that serve as fundamental units for Transformer-based modeling. Each token itself is a compact, continuous vector representation. *This is consistent with established practices in the broader Vision Transformer literature [1],where images are transformed into sequences of tokens, each representing a continuous vector of image patches. Within the EEG community, numerous studies have explored tokenizing EEG signals for Transformer-based models [2-7].*
>
> [1] Dosovitskiy, Alexey, et al. "An Image is Worth 16x16 Words: Transformers for Image Recognition at Scale." International Conference on Learning Representations.
>
> [2] Song, Yonghao, et al. "EEG conformer: Convolutional transformer for EEG decoding and visualization." IEEE Transactions on Neural Systems and Rehabilitation Engineering 31 (2022): 710-719.
>
> [3]  Shen, Qi, et al. "LGSleepNet: an automatic sleep staging model based on local and global representation learning." IEEE Transactions on Instrumentation and Measurement 72 (2023): 1-14.
>
> [4] Mentzelopoulos, Georgios, et al. "Neural decoding from stereotactic EEG: accounting for electrode variability across subjects." Advances in Neural Information Processing Systems 37 (2024): 108600-108624.
>
> [5] Khadka, Rabindra, et al. "Inducing inductive bias in vision transformer for EEG classification." ICASSP IEEE International Conference on Acoustics, Speech and Signal Processing (ICASSP). IEEE, 2024.
>
> [6] Kim, Sung-Jin, et al. "Toward domain-free transformer for generalized EEG pre-training." IEEE Transactions on Neural Systems and Rehabilitation Engineering 32 (2024): 482-492.
>
> [7] Cui, Wenhui, et al. "Neuro-gpt: Towards a foundation model for eeg." 2024 IEEE International Symposium on Biomedical Imaging (ISBI). IEEE, 2024.
>
> LaBraM and Cbramod are cutting-edge models in EEG foundation research, which is why we have compared and referenced them extensively. **We agree that a grounded view of EEG modeling literature is important.** In fact, we cite and discuss 74 and 81 papers in the main paper and supplementary materials, including many classic works in EEG modeling.  *We hope that, building on these prior community efforts, our work will contribute to advancing EEG modeling toward more generalizable solutions, facilitating real-world applications such as clinical diagnostics, cognitive augmentation, and brain-computer interfaces.*
>
> *Thank you again for your review. If you have any further questions, please feel free to contact us.*

---

### Official Review · Reviewer_Yw1n · 2025-07-03

**Clarity:** 3
**Significance:** 3
**Originality:** 3
**Rating:** 4
**Confidence:** 4

**Summary:**

The authors present CSBrain, a foundation model for EEG decoding that combines cross-scale spatiotemporal tokenization (CST) and structured sparse attention (SSA) to capture multi-scale, anatomically-aware dynamics in EEG signals. They pretrained their model on ~9,000 hours of EEG and evaluated across 11 tasks on 16 public datasets.

**Questions:**

1.	Why was concatenation chosen for feature fusion in 2.1 rather than addition or another method?
2.	What guided the specific design of SSA’s formulation? Can the authors provide deeper motivation or derivation?
3.	Can the authors clarify whether aggregation in 155–156 refers to functional or anatomical context? If functional, how is it designed in the model to do so?
4.	Could SSA’s formulation (185–203) be explained in greater depth or with additional derivation?
5.	Could the authors provide qualitative brain map analyses for different attention variants or for baselines?

Minor:

There are minor typos, such as in line 154, where the superscript appears as t but should likely be s (referring to spatial rather than temporal context).

**Ethical Concerns:**

["NO or VERY MINOR ethics concerns only"]

**Final Justification:**

The authors addressed several of my concerns with new analyses and clarifications. However, I am still not convinced of their claim regarding GNNs being inherently limited in capturing long-range dependencies due to their receptive field, as the receptive field can be extended or adapted through graph construction strategies. Additionally, I still believe that a more comprehensive qualitative analysis of the attention mechanisms were needed to assess whether the proposed method truly offers improved interpretability. Overall, I will keep my borderline accept score.

**Limitations:**

yes

**Quality:**

3

**Strengths And Weaknesses:**

Strengths:

1.	The paper is well-written, with a clear structure and logical flow. Most sections are supplemented by additional material in the appendix.
2.	The authors perform a comprehensive evaluation, applying CSBrain to a wide range of tasks and datasets.

Weaknesses:

1.	In Section 2.1, the choice to concatenate features into a unified embedding is not justified — why not addition or another fusion strategy?
2.	In Section 2.2, the formulation of SSA lacks deeper derivation or motivation. How was this design chosen? What theoretical or empirical reasoning supports it?
3.	The explanation of SSA (lines 185-203) is hard to follow — more derivation or intuition would strengthen this section.
4.	Details on baseline training (e.g., whether competing foundation models were re-trained with the same objective or used published weights) are sparse, making it harder to assess comparison fairness.
5.	Ablations focus more on SSA variations but less on CST: no analysis of token granularity choices, window/region design, or their effect on generalization.
6.	As GNN architectures have shown strong performance in EEG domain for purposes such as emotion recognition and seizure detection by explicitly modeling electrode connectivity, including a GNN baseline would be valuable. For instance, GGN[1] for seizure detection, and HetEmotionNet[2] for emotion classification. Adding such baselines would provide meaningful context for evaluating CSBrain’s approach to spatial dependency modeling.
7.	Qualitative analyses of attention maps across attention types (dense, criss-cross, SSA) are missing; this would enhance interpretability.
8.	The paper provides brain maps for CSBrain but not for baselines — including these would better contextualize the model’s interpretability gains.

[1] Li, Zhengdao, et al. "Graph-generative neural network for EEG-based epileptic seizure detection via discovery of dynamic brain functional connectivity." Scientific reports 12.1 (2022): 18998.

[2] Jia, Ziyu, et al. "HetEmotionNet: two-stream heterogeneous graph recurrent neural network for multi-modal emotion recognition." Proceedings of the 29th ACM international conference on multimedia. 2021.

---

> ### Author Rebuttal · Authors · 2025-07-28
>
> Dear Reviewer Yw1n,
>
> Thank you for your encouraging feedback and constructive suggestions! We are greatly encouraged by your recognition of our **comprehensive evaluation** and that the paper is **well-written, with a clear structure and logical flow**. Due to time and page limitations, we have made every effort to address your questions and suggestions through the clarifications provided. We hope these responses resolve most of your concerns and will be considered in your final assessment.
>
> > W1&Q1. Feature Fusion Strategy
>
> Thanks for the suggestion. In EEG analysis, both concatenation and addition are commonly used to fuse temporal and spectral information. We opted for concatenation as it preserves the integrity and independence of each feature type, thereby providing a richer input space. The choice of concatenation-based feature fusion is also widely supported by prior validated studies [1-2].
>
> Notably, the core contribution of our work lies in the CSBrain architecture itself, particularly the Cross-Scale Spatiotemporal Tokenization (CST) and Structured Sparse Attention (SSA) modules. These components build cross-scale dependencies upon these fused initial features, capturing intricate spatiotemporal patterns. The effectiveness of our work has been validated across 11 representative tasks and 16 datasets.
>
> We agree that examining their impact on EEG representation learning is valuable and will consider this in future work, given the limited rebuttal timeframe. *Your insightful comment will be included in Sec. G.3 (Future Work) to guide further research.*
>
> [1] Li, Yang, et al. "A temporal-spectral-based squeeze-and-excitation feature fusion network for motor imagery EEG decoding." IEEE T-NSRE2021, Cited by 142
>
> [2] Jia, Ziyu, et al. "Sst-emotionnet: Spatial-spectral-temporal based attention 3d dense network for eeg emotion recognition." ACM MM2020,  Cited by 170
>
> > W2/3&Q2/4. Design Motivation of SSA
>
> - **Motivation:** SSA is designed to model long-range spatiotemporal dependencies across time windows and brain regions via efficient sparse attention. It complements CST that captures structured, cross-scale representations within local temporal and spatial contexts. This synergy enables CSBrain to effectively model brain activity from local to global scales, aligning with the cross-scale nature of neural dynamics.
>
> - **Details:** CST aggregates contextual information within local temporal windows and brain regions, endowing each token with window/region-level awareness. Based on this, SSA performs sparse attention computation across windows and regions. Specifically, cross-scale tokens are sampled and grouped based on the same indices, and attention is then computed within each group, effectively having a representative from each window/region interact across regions/windows. Multiple groups are set up to ensure every token is included in at least one group for computation, preventing information loss.
> Additionally, in inter-region attention, we introduce a regional descriptor, which adds the average feature of the region to the sampled token. Unlike temporal windows (which have fixed lengths), the number of electrodes varies across brain regions. In regions with more electrodes, CST might not fully capture all information. By introducing the regional descriptor, we ensure complete information flow between regions.
>
> As shown in Sec. 2.2 (L293–305), SSA achieves superior performance and efficiency compared to other attention mechanisms. *We will integrate this explanation into L185–203 of the main paper for clarity, and release our code and models to support future research.*
>
> > W4. Details on Foundation Model Baseline
>
> For all compared foundation models, we utilized their officially published weights from their respective GitHub repositories. This is a common and validated practice adopted by prior works including [ICLR24]Labram and [ICLR25]CbraMod for evaluating foundation models, ensuring fair comparison against their intended performance.
>
> The specific evaluation protocols for each task and dataset are outlined in Supp. Sec. C&D, following practices of CbraMod and other well-established studies. *To further enhance transparency, we will include an additional note in Sec.C.2 to specify the use of the published weights and provide weight download links.*
>
> > W5. More Ablations on CST
>
> - As shown in L278-292 and Fig. 4 of the main paper, we have already provided an analysis of token granularity choices in CST, as well as the effect of CST stacking strategy on performance. These results demonstrate CST's role in learning richer, more flexible EEG representations and the importance of token granularity in effectively capturing diverse EEG dynamics across tasks.
> - To further address your concerns and provide deeper insights into token granularity and window/region design, we conduct additional experiments to explore the configuration of using multiple kernels of the same scale: [1,1,1], [3,3,3], and [5,5,5], in contrast to our multi-scale [1,3,5] design. Due to time and computational constraints, these experiments were pre-trained on a subset of the TUEG dataset (2000 hours, excluding TUEV) and evaluated on two representative tasks.
>
> |Config|TUEV B-Acc|TUEV Kappa|TUEV F1-W|BCIC2020-3 B-Acc|BCIC2020-3 Kappa|BCIC2020-3 F1-W|
> |-|-|-|-|-|-|-|
> |[1,3,5]|0.6414±0.0187|0.6236±0.0198|0.7971±0.0183|0.5443±0.0232|0.4254±0.0301|0.5527±0.0297|
> |[1,1,1]|0.6122±0.0193|0.5596±0.0212|0.7713±0.0205|0.5041±0.0208|0.3821±0.0223|0.5019±0.0202|
> |[3,3,3]|0.6338±0.0184|0.6137±0.0179|0.7870±0.0188|0.5298±0.0247|0.4354±0.0298|0.5399±0.0280|
> |[5,5,5]|0.6188±0.0190|0.5765±0.0209|0.7732±0.0195|0.4998±0.0237|0.3809±0.0242|0.4931±0.0279|
>
> These results further validate CST’s effectiveness in cross-scale modeling. Our proposed [1,3,5] multi-scale configuration consistently outperforms single-scale alternatives, showing that modeling diverse scales is more effective than simply increasing uniform kernels. This reinforces the importance of CST’s design for capturing EEG’s complex multi-scale spatiotemporal patterns and enhancing generalization. *We will update Supp. Sec. E to reflect these insights.*
>
> > W6. GNN Baselines
>
> *We will reference and discuss the two classic GNN works you mentioned in both the Introduction and Related Work, and include them as baselines in our evaluation.* Due to time constraints, we conducted tests on 4 representative tasks and *will include the full results in the revised paper.* Note that HetEmotionNet was originally designed for heterogeneous multimodal inputs, so when applied to EEG-only datasets, the graph is simplified, which may reduce its performance.
>
> |Model|BCIC-IV-2a(B-Acc,Kappa,F1-W)|TUEV(B-Acc,Kappa,F1-W)|MentalArithmetic(B-Acc,AUC-PR,AUROC)|FACED(B-Acc,Kappa,F1-W)|
> |-|-|-|-|-|
> |HetEmotionNet|0.4479,0.2639,0.4295|0.4771,0.4356,0.7105|0.6424,0.5924,0.7857|0.3928,0.3134,0.3849|
> |GCN|0.4653,0.2870,0.4436|0.5778,0.5669,0.7631|0.6250,0.4836,0.7254|0.4274,0.3541,0.4277|
> |CSBrain|0.5657,0.4209,0.5637|0.6903,0.6833,0.8333|0.7558,0.6696,0.8478|0.5752,0.5204,0.5796|
>
> As shown, CSBrain consistently outperforms GNNs, demonstrating its superior capability in capturing spatiotemporal dependencies.
>
> > W7/8&Q5. More Visualization Results
>
> We greatly appreciate your valuable suggestions. We fully agree that such analyses would significantly enhance model interpretability and highlight CSBrain's advantages. However, *due to NeurIPS conference committee regulations, we are unable to submit new PDFs or external links to showcase additional visualization results during the rebuttal phase.*
>
> Nevertheless, we have already provided a rich set of visualizations in the main paper and supplementary materials, designed to illustrate CSBrain's interpretability and the effectiveness of its cross-scale modeling:
> - **Topography Visualizations:** Presented in the main paper (Fig. 6, L337-345) on four representative tasks. Supp. F.1 (L474-501, Figs.6/7/8/9) provides more detailed topography visualizations for various categories within the four different tasks.
> - **Representation Visualizations (t-SNE):** In Supp. F.2 (Fig. 10/11, L502-512), we visualize the feature distributions of raw EEG signals versus CSBrain's learned representations on two tasks. These demonstrate CSBrain's ability to learn highly discriminative and well-separated representations.
> - **Channel Relationship Visualizations:** Supp. F.3 (Fig. 12/13, L513-525) includes visualizations and analysis of learned inter-channel relationships, highlighting our model's capacity to capture both local and global global spatial dynamics.
>
> These existing visualizations collectively demonstrate CSBrain's strong interpretability and the effectiveness of its cross-scale modeling. We believe this constitutes one of the most comprehensive sets of visualizations in published EEG foundation model papers to date. *We sincerely hope you understand our inability to upload more visualization results due to NeurIPS regulations, and we look forward to your support.*
>
> > Q3. Aggregation Context
>
> To clarify, the aggregation in L155-156 refers specifically to  anatomically defined brain regions. We do not explicitly perform functional aggregation as precise functional connectivities are difficult to define. Instead, our spatial tokenization explicitly incorporates anatomical priors by assigning electrodes to five brain regions (e.g., frontal, central, parietal, temporal, occipital lobes) anatomically defined in the standardized 10-20 system. These lobes broadly reflect the distribution patterns of functional regions and provide a physiologically meaningful framework for spatial grouping. As evidenced by Fig. 6 and visualizations in the supp, our model effectively learns functional characteristics across tasks in a coarse-grained manner. by facilitating the modeling of cross-scale neural structures that respects brain anatomy, our model allows effective adaptation to multiple downstream EEG tasks.

---

> > ### Comment · Reviewer_Yw1n · 2025-08-04
> >
> > I thank the authors for the detailed rebuttal and additional experiments, which helped address several concerns. However, a few important issues remain:
> >
> > -  **Feature Fusion**:
> > While citing prior work justifies the choice of concatenation to some extent, it would have been helpful to empirically evaluate alternative fusion strategies (e.g., addition, learned fusion) within your architecture. Since this design choice could impact the downstream representation quality, relying solely on literature precedence without testing alternatives limits interpretability of this modeling decision.
> >
> > -  **GNN Baselines**:
> >  The results show that CSBrain outperforms GNN-based methods across several tasks. However, it would be valuable to include a discussion of why this happens. Do SSA or CST capture spatiotemporal dependencies in ways that GNNs cannot? Any insight or hypothesis on this?
> >
> > -  **Visualization & Interpretability**:
> > A more comprehensive qualitative analysis of attention mechanisms (e.g., comparing dense, criss-cross, and SSA) is still missing. Visualizing and comparing attention maps would provide crucial interpretability and evidence that SSA captures meaningful spatiotemporal patterns. Similarly, brain maps for CSBrain are helpful, but their absence for baseline models makes it difficult to assess whether interpretability is truly improved relative to prior methods.

---

> > > ### Comment · Reviewer_6YFP · 2025-08-05
> > > **On Feature Fusion**
> > >
> > > Regarding the suggestion to empirically evaluate alternative fusion strategies (e.g., addition, learned fusion) instead of concatenation, I believe this concern is somewhat tangential to the core contributions of the paper. From a modeling perspective, concatenation can be viewed as a simple form of learned fusion, and summation likewise can be represented within a learned fusion framework. Given this, if concatenation performs well in the current setting, it is reasonable to expect alternative fusion methods to yield similar performance. Thus, a dedicated empirical comparison may not substantially affect the interpretability or conclusions of the paper.

---

> > > ### Author Response · Authors · 2025-08-07
> > >
> > > Dear Reviewer Yw1n,
> > >
> > > Thank you for your further and detailed comments. We are glad that our detailed rebuttal and additional experiments have addressed your several concerns.
> > >
> > > **Feature Fusion**
> > >
> > > In response, we conduct experiments comparing two fusion strategies: concatenation and addition, to assess their impact on model performance. Due to time and computational constraints, these experiments were pre-trained on a subset of the TUEG dataset (2000 hours, excluding TUEV) and evaluated on two downstream tasks.
> > >
> > > | Fusion Strategy | TUEV B-Acc | TUEV Kappa | TUEV F1-W | BCIC2020-3 B-Acc | BCIC2020-3 Kappa | BCIC2020-3 F1-W |
> > > |-|-|-|-|-|-|-|
> > > | Concat          | 0.6414 ± 0.0187 | 0.6236 ± 0.0198 | 0.7971 ± 0.0183 | 0.5443 ± 0.0232 | 0.4254 ± 0.0301 | 0.5527 ± 0.0297 |
> > > | Addition        | 0.6451 ± 0.0209 | 0.6228 ± 0.0217 | 0.7995 ± 0.0198 | 0.5418 ± 0.0193 | 0.4272 ± 0.0273 | 0.5459 ± 0.0259 |
> > >
> > > As shown, both strategies yield very similar performance. **This aligns with a comment from Reviewer 6YFP**, who noted that both concatenation and addition (summation) are simple and effective fusion methods. If concatenation performs well in the current setting, it is reasonable to expect alternative fusion methods to yield similar performance.
> > >
> > > **Further Analysis of GNN Baselines**
> > >
> > > GNNs have shown promise in EEG modeling, especially for emotion recognition and seizure detection, where they excel at modeling spatial relationships due to their graph-based structure. These models excel at encoding the graph topology, which is physiologically meaningful in certain cases, such as spatially adjacent electrodes. However, GNNs may struggle to capture long-range dependencies due to their fixed receptive fields, limiting their ability to model cross-scale spatiotemporal patterns in EEG signals and their application to diverse tasks.
> > >
> > > In contrast, CSBrain leverages large-scale pretraining and a unified cross-scale model architecture, enabling it to adapt to a wider range of tasks and exhibit better generalization performance. CSBrain, through the mutual cooperation of its CST and SSA modules, explicitly incorporates cross-scale modeling to better handle the complexity and diversity of EEG data. While GNNs can effectively represent spatial relationships in a specific decoding task, their capacity to generalize across long-range temporal and spatial dependencies remains constrained.
> > >
> > > *We will further revise the introduction and related work sections, citing more EEG-related papers based on GNNs and providing a deeper discussion of the strengths and weaknesses of different model architectures for EEG modeling. The two excellent GNN works you mentioned will be included as evaluation baselines in Sections 3.2 and 3.3 of the main paper.*  We look forward to advancing EEG modeling towards more generalizable solutions through collective efforts within the community, which will certainly involve exploring diverse technical approaches, and truly facilitating real-world applications.
> > >
> > > [1] Klepl, Dominik, Min Wu, and Fei He. "Graph neural network-based eeg classification: A survey." IEEE Transactions on Neural Systems and Rehabilitation Engineering 32 (2024): 493-503.
> > >
> > > [2] Liu, Chenyu, et al. "Graph neural networks in eeg-based emotion recognition: a survey." arXiv preprint arXiv:2402.01138 (2024).
> > >
> > > [3] Jia, Ziyu, et al. "HetEmotionNet: two-stream heterogeneous graph recurrent neural network for multi-modal emotion recognition." Proceedings of the 29th ACM international conference on multimedia. 2021.
> > >
> > > [4] Li, Zhengdao, et al. "Graph-generative neural network for EEG-based epileptic seizure detection via discovery of dynamic brain functional connectivity." Scientific reports 12.1 (2022): 18998.
> > >
> > >
> > >
> > > **Visualization & Interpretability**
> > >
> > > As mentioned in our previous response, **due to NeurIPS' submission policies**, we are unable to provide new PDFs or external links for additional visualizations during the rebuttal phase. *We promise you that we will include additional visualization results in Supplementary Section F of the revised version.* We kindly emphasize that our core contribution lies in CSBrain's architecture, which offers a cross-scale, structure-aware approach to capture multi-scale spatiotemporal neural patterns across diverse tasks. This has been extensively validated through our comprehensive quantitative analysis across 11 tasks and 16 datasets , as well as supplementary visualization analyses on topography, representation, and channel relationship.
> > >
> > > We look forward to contributing to the community’s ongoing efforts, with the goal of advancing EEG modeling toward more generalizable solutions and facilitating real-world applications such as clinical diagnostics, cognitive augmentation, and brain-computer interfaces.

---

> > > > ### Comment · Reviewer_Yw1n · 2025-08-08
> > > >
> > > > I thank the authors for the additional clarifications. However, I would like to note that the claim regarding GNNs being inherently limited in capturing long-range dependencies due to their receptive field is not entirely accurate. The receptive field can be extended or adapted through graph construction strategies, and therefore this limitation is not intrinsic to GNNs themselves.
> > > > Additionally, I still believe that a more comprehensive qualitative analysis of the attention mechanisms is needed to strengthen interpretability claims and help with their assessment. While the brain maps for CSBrain are helpful, the absence of similar interpretability tools for baseline models makes it difficult to assess whether the proposed method truly offers improved interpretability. My other questions are addressed and I have no further questions. Thanks to the authors.

---

> > > > > ### Author Response · Authors · 2025-08-09
> > > > >
> > > > > Dear Reviewer Yw1n,
> > > > >
> > > > > Thank you for your positive and further insightful comments. We are glad that our responses have addressed your concerns and left no further questions.
> > > > >
> > > > > Regarding the visualizations, due to NeurIPS policies during the rebuttal and discussion phase, we are unable to upload additional images. We will include more qualitative analysis and visualizations in Supp. Sec. F of the final version. We appreciate your perspective on GNNs, especially on the receptive field. We agree that GNNs represent an important architecture for EEG modeling, and your feedback has further inspired us. We believe that large-scale pretraining-based GNN-based brain foundation models could be a valuable direction for future research. We will integrate your valuable insights and our discussion into the final version of our paper and hope that our collective efforts will inspire further research, advancing more generalizable EEG modeling, and ultimately facilitating real-world applications.
> > > > >
> > > > > Thank you for your time and consideration.

---

### Official Review · Reviewer_5sTW · 2025-07-03

**Clarity:** 2
**Significance:** 3
**Originality:** 2
**Rating:** 4
**Confidence:** 4

**Summary:**

This paper proposes CSBrain, a foundation model for EEG decoding that addresses a key limitation of existing approaches: their failure to capture the cross-scale spatiotemporal structure inherent in neural activity. CSBrain introduces two main components: Cross-scale Spatiotemporal Tokenization (CST) aggregates multi-scale features within temporal windows and brain regions into compact tokens, while Structured Sparse Attention (SSA) efficiently models long-range dependencies across windows and regions. These modules are alternately stacked to progressively integrate cross-scale dependencies. The authors demonstrate CSBrain's effectiveness across 11 EEG tasks and 16 datasets, showing consistent improvements over both task-specific models and foundation baselines. The work establishes cross-scale modeling as a crucial inductive bias for generalized EEG decoding.

**Questions:**

1.	How does CST specifically handle varying temporal scales and channel numbers across datasets? Please provide technical details on the adaptation mechanism for different sampling rates and electrode configurations.
2.	Why use uniform 30s windows and 19 channels during pretraining if cross-scale modeling is key? Please justify this design choice and provide ablation studies on scale diversity impact.
3.	How are 19 channels selected from the Temple University Hospital corpus with various configurations? How does the model handle missing/additional channels in practice?
4.	Please provide detailed complexity analysis and runtime comparisons for SSA versus dense attention across different temporal lengths and channel numbers.

**Ethical Concerns:**

["NO or VERY MINOR ethics concerns only"]

**Final Justification:**

The authors have provided clear explanations and new experimental results that address several of the concerns raised in my initial review. The paper is now a more complete work, as the authors have resolved the main points of uncertainty. Therefore, I am maintaining my score of **4: Borderline accept**.

**Limitations:**

Yes

**Quality:**

3

**Strengths And Weaknesses:**

Strengths:
1. The paper is well-written with clear logical flow and comprehensive method descriptions. The motivation for cross-scale modeling is well-articulated, and the technical components (CST and SSA) are described with sufficient detail for understanding and reproducibility.
2. The experimental evaluation is thorough and convincing, spanning 11 diverse EEG decoding tasks across 16 datasets with consistent improvements over both task-specific models and strong foundation baselines, demonstrating the generalizability of the approach.
3. The paper introduces a principled architectural innovation that explicitly captures the inherent multi-scale spatiotemporal structure of neural activity, moving beyond scale-agnostic dense modeling paradigms to establish cross-scale modeling as a key inductive bias for EEG decoding.

Weakness:

1.	The paper lacks detailed explanation of how CST handles differences across temporal scales and varying channel numbers, with the cross-scale adaptation mechanism not thoroughly described in the main text.
2. Despite proposing cross-scale modeling, CSBrain uses uniform 30s temporal windows and 19 channels during pretraining, only encountering scale diversity in downstream tasks, which may limit learning of truly scale-invariant representations.
3. The methodology for selecting 19 channels from the Temple University Hospital EEG corpus with various channel configurations is not clearly described, affecting reproducibility and understanding of generalization capabilities.

---

> ### Author Rebuttal · Authors · 2025-07-26
>
> Dear Reviewer 5sTW,
>
> Thank you for your encouraging feedback and constructive suggestions! We are greatly encouraged by your recognition of our **principled architectural innovation, the thorough and convincing experimental evaluation**, and **the paper's well-written and comprehensive method descriptions**.  We are also pleased that you found **the motivation for cross-scale modeling well-articulated**, and that **the CST and SSA components are described with sufficient detail for understanding and reproducibility**. Due to time and page limitations, we have made every effort to address your questions through the clarifications provided. We hope these responses resolve most of your concerns and will be considered in your final assessment.
>
> > W1&Q1. Handling of Varying Spatiotemporal Settings.
>
> 1. **Different Sampling Rates&Temporal Scales:**
> As mentioned in Sec.2.1 (L120), all raw EEG signals undergo a standardized preprocessing pipeline, including resampling to a uniform 200 Hz. This ensures that temporal differences across datasets are normalized before entering the CST, **aligning with standard practices in foundation models like [ICLR2024] LaBraM, [NIPS2024] BIOT, and [ICLR2025] Cbramod.**
> Subsequently, CST further adapts to different temporal scales through its multi-scale temporal convolutions, using kernels of different sizes to capture temporal patterns at various scales. CST and the subsequent SSA module closely collaborate and complement each other to jointly capture diverse temporal scales from local to global. This combined setup collectively fulfills the varying temporal scale requirements of different datasets.
>
> 2. **Different Electrode Configurations:**
> In CSBrain, we first partition electrodes based on the standardized 10-20 system into brain regions (Frontal, Central, Parietal, Temporal, Occipital). Then, within the electrodes of each defined brain region, CST applies local multi-scale spatial convolutions. CST enables the model to learn localized yet multi-scale patterns robustly, irrespective of channel count variations within a region. This design ensures that while electrode numbers and spatial distributions may vary across different datasets, the unified 10-20 system-based spatial region partitioning and the application of local multi-scale spatial convolutions allow CST to effectively adapt to diverse electrode configurations, ensuring robust compatibility.
>
> *To make our design motivation and process even clearer, we will modify Sec. 2.1 and 2.2. Specifically, Sec. 2.1 will further emphasize the role of data preprocessing in handling diverse datasets. Sec. 2.2 (L141-153) will add clarifications to highlight CST's adaptation to different temporal scales and channel configurations. Additionally, we will open-source our code and models to facilitate further research and community use.*
>
> > Q2. Pretraining Data Setting.
>
> **The cross-scale capability of CSBrain comes from the intricate model architecture, not from the variations in input signal format.** Through the synergistic combination of the CST and SSA modules, we effectively capture cross-scale spatiotemporal dependencies, as demonstrated by our experiments across 11 tasks and 16 datasets. Speciffcally:
>
> **1. Design Justification:** The main objective of pretraining is to learn robust, generalizable EEG representations from a large, unlabeled corpus. To achieve this efficiently, we chose a uniform data setting to simplify the data pipeline, ensuring stability and efficiency during large-scale training. Using varying input settings, such as varying EEG segment lengths or channel counts, would complicate preprocessing and model training, leading to instability. This is consistent with other widely validated models like CBraMod, which also use uniform input for pretraining. Additionally, *as shown in Supp. Sec. E.1, we have demonstrated that our pretraining improves performance across datasets with various spatiotemporal settings.*
>
> **2. More Ablation Studies:** we further conduct ablation studies to evaluate the impact of pretraining diversity and model configurations on downstream task performance. Due to time and resource constraints, the total pretraining data is fixed at 2000 hours.
>
> (1) Multiple Pretraining (MP): We pretrain on diverse data configurations, including 5s, 20s, and 30s windows with the TUEG dataset (19 channels), and datasets like FACED (10s, 32 channels), SEED-VIG (8s, 17 channels), and BCIC-IV-2a (4s, 22 channels), totaling 2000 hours.
>
> (2) Single Pretraining (SP): We pretrain on a subset of the TUEG dataset (30s, 19 channels), totaling 2000 hours.
>
> (3) Single-Scale CSBrain: A single-scale configuration by adjusting CST convolution kernels to [1, 1, 1].
>
> |Config|TUEV B-A |TUEV Kappa|TUEV F1|BCIC2020-3 B-A|BCIC2020-3 Kappa|BCIC2020-3 F1|
> |-|-|-|-|-|-|-|
> |MP+Full CSBrain|0.5390±0.0229|0.4399±0.0290|0.5501±0.0198|0.6465± 0.0178|0.6264±0.0169|0.7892±0.0180|
> |SP+Full CSBrain|0.5443±0.0232|0.4254±0.0301|0.5527±0.0297|0.6414±0.0187|0.6236±0.0198|0.7971±0.0183|
> |MP+Single-Scale CSBrain|0.5041±0.0208|0.3821±0.0223|0.5019±0.0202|0.6122±0.0193|0.5596±0.0212|0.7713±0.0205|
>
> *Results show that, under the same pretraining data volume, variations in data format has a minimal impact on downstream performance. The cross-scale model architecture (CST and SSA) is the primary factor in improving performance, further validating the effectiveness of our pretraining setup and the core idea of cross-scale spatiotemporal modeling.*
>
> > W3&Q3 Pretrained Channels Selection & Handling of Diverse Downstream Channel Settings.
>
> **Pretrained Channel Selection.**
> As noted in Sec.3.1 L225, we follow established practices by [ICLR2025]CBraMod using the TUEG dataset for pretraining. The EEG signals in TUEG range from 19 to 23 channels. We select 19 common EEG channels (Fp1, Fp2, F7, F3, Fz, F4, F8, T3, C3, Cz, C4, T4, T5, P3, Pz, P4, T6, O1, O2) that correspond to a subset of the 10-20 international electrode placement system to ensure clean and uniformly formatted pretraining data. The dropped channels include A1, A2, T1, and T2. A1 and A2 are located near the auricular region, which contain less brain information and are typically used for reference electrodes. T1 and T2 are obsolete and no longer utilized.
>
> **Handling of Diverse Downstream Channel Settings.**
>
> - **Architecture.** CSBrain's architecture is inherently designed to adapt to the actual channel count of each downstream dataset. During fine-tuning, for any given downstream dataset, we map its electrodes to a set of anatomically predefined brain regions $\mathcal{R}$ based on the 10-20 system. Within each region ($R_r$), CST applies multi-scale spatial convolution kernels. This allows CST to aggregate cross-scale features from the available electrodes within that region, regardless of their exact number. Subsequently, SSA computes attention based on regional descriptors formed via sequentially sampled tokens $x_r^{rep}$ and average features of $R_r$, establishing long-range dependencies between these regions. *Thus, despite varying electrode configurations, our brain-region-based assignment allows diverse downstream setups to benefit from large-scale pretraining.*
> - **Implementation.** From an implementation perspective, *except for the task-specific heads with fixed input-output dimensions, all model weights are inherited directly from pretraining.* This is because the underlying convolutional and Transformer components of CSBrain are flexible enough to handle varying input sizes and dynamically grouped region information.
> - **Experiment Validation.** Furthermore, *as stated in supp. mat. Sec. E.1 (L418-435), our work, consistent with prior foundation models like Cbramod and BIOT, clearly demonstrates the effectiveness of pretraining*. These pre-trained models significantly improve performance on downstream tasks, even those with different channel settings.
>
> *To further enhance transparency, new Sec. B.3 and C.4 will be added to the Supp., detailing the channel settings and region assignments for the pretraining data and each downstream task dataset, respectively.* Our code and models for pretraining and downstream fine-tuning will be **open-sourced** to facilitate community use.
>
> > Q4. Complexity Analysis.
>
> We compare the runtime, computational complexity, and performance of SSA versus Dense Attention (DA) on TUEV (C=16, T=5s), BCIC2020-3 (C=64, T=3s), and FACED (C=32, T=10s) datasets. Runtimes are averaged over 500 runs on a single A100 GPU with a batch size of 64. For DA, complexity is $O(N^2)$, where $N = T \times C$. For SSA, complexity is $O(N \cdot k)$, where $k$ = max(electrodes within a brain region) + window length.
>
> |Model|Runtime|Complexity|B-Acc|Kappa|F1-W|
> |-|-|-|-|-|-|
> |DA-TUEV|0.0675s|$(16\times5)^2$|0.5957|0.5781|0.7655|
> |SSA-TUEV|0.0603s|$(16\times5)\times8$|0.6901|0.6839|0.8339|
> |DA-BCIC|0.0778s|$(64\times3)^2$|0.5075|0.3856|0.5156|
> |SSA-BCIC|0.0695s|$(64\times3)\times26$|0.6001|0.5018|0.6008|
> |DA-FACED|0.0921s|$(32\times10)^2$|0.5272|0.4798|0.5231|
> |SSA-FACED|0.0799s|$(32\times10)\times13$|0.5754|0.5202|0.5802|
>
> *We observe that SSA consistently exhibits lower runtime and complexity across various time and channel settings.* It should be noted that while SSA offers clear efficiency gains, the absolute runtime difference between SSA and DA is not as dramatically pronounced as the complexity analysis suggests, owing to PyTorch's highly optimized parallel computations for attention mechanisms. *Crucially, SSA consistently outperforms DA across all performance metrics.* This advantage stems from SSA's respect for the intrinsic cross-scale spatiotemporal structure of neural activity, effectively capturing meaningful dependencies without indiscriminately associating all tokens. *We will briefly add this analysis to L293-305 of the main paper and provide the full details in Supp. Sec. E.*

---

> > ### Comment · Reviewer_5sTW · 2025-08-06
> >
> > I thank the authors for their detailed and comprehensive rebuttal. They have provided clear explanations and new experimental results that address several of the concerns raised in my initial review.
> >
> > Specifically:
> >
> > - The clarification on how the CST module handles varying spatiotemporal settings through standardized preprocessing and brain-region partitioning is now clear and well-justified.
> > - The added complexity and runtime analysis for the SSA module is also a welcome addition that strengthens the paper's claims of efficiency.
> >
> > Regarding the pretraining data diversity, I thank the authors for conducting the additional ablation study comparing Single Pretraining (SP) and Multiple Pretraining (MP). However, I have a remaining concern about the experimental design of this comparison. Both SP and MP were constrained to a total of 2000 hours of data. The MP setting introduces significantly more complexity by incorporating diverse temporal scales and data sources. By keeping the total data volume fixed, the amount of data for any single scale or format within the MP setting is reduced, which could increase training difficulty and hinder the model's ability to learn from the more complex data distribution. Therefore, the conclusion that "variations in data format has a minimal impact" may be a direct consequence of this experimental design, rather than a generalizable finding. While the study provides some insight, it doesn't entirely resolve the question of whether a more data-rich, multi-scale pretraining schemes could yield further improvements.
> >
> > The paper is now a more complete work, as the authors have resolved the main points of uncertainty. Therefore, I am maintaining my score of **4: Borderline accept**.

---

> > > ### Author Response · Authors · 2025-08-07
> > >
> > > Dear Reviewer 5sTW,
> > >
> > > Thank you for your insightful and constructive feedback. We are glad to hear that our detailed and comprehensive responses, along with the additional experiments, have resolved the main points of uncertainty in your initial review.
> > >
> > > We agree that further improvements can be achieved by constructing richer and more diverse high-quality databases for pre-training, and we believe this is a key direction for future research. *This point was discussed in the limitations and future work (Supp. G.2&3) sections of our initial version.* Although we were constrained by time and computational resources during the rebuttal period, we are committed to collaborating with the community on more larger-scale studies to address this challenge in our future work.
> > >
> > > We are very grateful for your constructive feedback. *We will integrate your insights into final versions of the paper to inspire further and broader research, pushing for more generalizable EEG modeling and facilitating real-world applications.*

---

### Decision · Program_Chairs · 2025-09-17

**Decision:**

Accept (spotlight)

**Comment:**

The authors present CSBrain, a foundation model for EEG decoding that introduces cross-scale spatiotemporal tokenization (CST) and structured sparse attention (SSA) to capture the multi-scale, anatomically grounded dynamics of neural activity. CST aggregates temporal and spatial features across multiple scales into compact tokens, while SSA efficiently models long-range dependencies across time windows and brain regions. By stacking these modules, CSBrain progressively integrates diverse cross-scale structures, addressing a key limitation of existing models that rely on scale-agnostic tokenization and dense attention. Pretrained on ~9,000 hours of EEG and extensively evaluated across several public datasets, CSBrain consistently outperforms task-specific architectures and foundation baselines. Extensive ablations further validate the effectiveness of cross-scale modeling and structured sparse attention, establishing them as crucial inductive biases for generalized EEG decoding. The findings are consistent with prior works.

The work features clear and well-structured writing, presenting a logical flow of ideas alongside detailed methodological descriptions that enhance reproducibility. It is strongly motivated by the need for cross-scale modeling, effectively addressing a major limitation in existing EEG analysis approaches. The paper introduces novel architectural contributions, specifically the CST and SSA modules, which embed natural inductive biases to better capture EEG characteristics. Its performance is validated through a comprehensive evaluation across 11 tasks and 16 datasets, consistently outperforming baseline methods. Overall, the work demonstrates strong generalizability, computational efficiency, and holds substantial potential to impact future EEG research.

There are some concerns regarding the design of the analysis for model pretraining, which could be further refined to strengthen the validity of the conclusions. Additionally, minor issues were noted related to the feature fusion approach, suggesting that alternative strategies might improve integration. There are also minor concerns about the interpretability of the model, indicating a need for clearer explanations.

The discussion and interaction between the authors and reviewers were very productive. This process enabled several improvements to the work, including: (i) an additional ablation study, (ii) a runtime analysis of the SSA module, (iii) clarification of the CST method, and (iv) more comprehensive literature references.